# An extended substrate screening strategy enabling a low lattice mismatch for highly reversible zinc anodes

Zhiyang Zheng [1,2], Xiongwei Zhong[1,2], Qi Zhang[1,2], Mengtian Zhang[1], Lixin Dai [1], Xiao Xiao[1], Jiahe Xu[1], Miaolun Jiao [1], Boran Wang[1], Hong Li[1], Yeyang Jia[1], Rui Mao[1] & Guangmin Zhou [1] ✉

Aqueous zinc batteries possess intrinsic safety and cost-effectiveness, but dendrite growth and side reactions of zinc anodes hinder their practical application. Here, we propose the extended substrate screening strategy for stabilizing zinc anodes and verify its availability ($d_{substrate}$: $d_{Zn(002)}$ = 1:1→$d_{substrate}$: $d_{Zn(002)}$=n:1, $n$ = 1, 2). From a series of calculated phyllosilicates satisfying $d_{substrate} \approx 2d_{Zn(002)}$, we select vermiculite, which has the lowest lattice mismatch (0.38%) reported so far, as the model to confirm the effectiveness of "$2d_{Zn(002)}$" substrates for zinc anodes protection. Then, we develop a monolayer porous vermiculite through a large-scale and green preparation as a functional coating for zinc electrodes. Unique "planting Zn(002) seeds" mechanism for "$2d_{Zn(002)}$" substrates is revealed to induce the oriented growth of zinc deposits. Additionally, the coating effectively inhibits side reactions and promotes zinc ion transport. Consequently, the modified symmetric cells operate stably for over 300 h at a high current density of 50 mA cm$^{-2}$. This work extends the substrate screening strategy and advances the understanding of zinc nucleation mechanism, paving the way for realizing high-rate and stable zinc-metal batteries.

Rechargeable aqueous zinc (Zn) metal-based batteries (AZBs) have arisen as a promising complement to organic electrolyte-based lithium batteries[1–4], due to the low working potential (−0.762 V versus the standard hydrogen electrodes) of Zn, high volumetric and specific capacity (5855 Ah L$^{-1}$ and 820 mAh g$^{-1}$), intrinsic safety, low cost and environmentally friendliness[5–8]. However, critical issues existing in Zn anodes heavily impede the practical application of AZBs. The rampant dendrites, caused by nonuniform plating/stripping of Zn, lead to inner short circuits and significantly affect the lifespan of batteries[9,10]. In addition, side reactions including hydrogen evolution reaction (HER) and irreversible byproduct formation are the culprits of low Coulombic efficiency (CE) and fast capacity decay (Fig. 1a)[11]. So far, various strategies have been proposed to address these issues, such as artificial interface construction[12–15], Zn bulk structure engineering[16–18],

electrolyte modification[19–21], and separator design[22,23]. Recent research reports that the preferred orientation of Zn(002) has a vital effect on the cycling durability of Zn anodes[24,25]. This is because: (i) Zn(002) plane with a relatively smooth surface enables an even electric field that guides the subsequent Zn deposition along (002) orientation; (ii) The lower chemical activity of Zn(002) plane alleviates side reactions and corrosion[17,26]. Therefore, inducing the growth of Zn electrodeposits along (002) facet has become an appealing strategy to stabilize Zn anodes.

Essentially, the anisotropy of the substrates significantly influences the orientation of Zn electrodeposits[27]. According to the lattice mismatch (δ) between deposited Zn and substrates, the contact interface can be divided into incoherent (δ > 25%), semi-coherent (25% ≥ δ ≥ 5%) and coherent interfaces (δ < 5%)[28]. The lower lattice

[1]Tsinghua-Berkeley Shenzhen Institute & Tsinghua Shenzhen International Graduate School, Tsinghua University, Shenzhen 518055, China. [2]These authors contributed equally: Zhiyang Zheng, Xiongwei Zhong, Qi Zhang. ✉e-mail: guangminzhou@sz.tsinghua.edu.cn

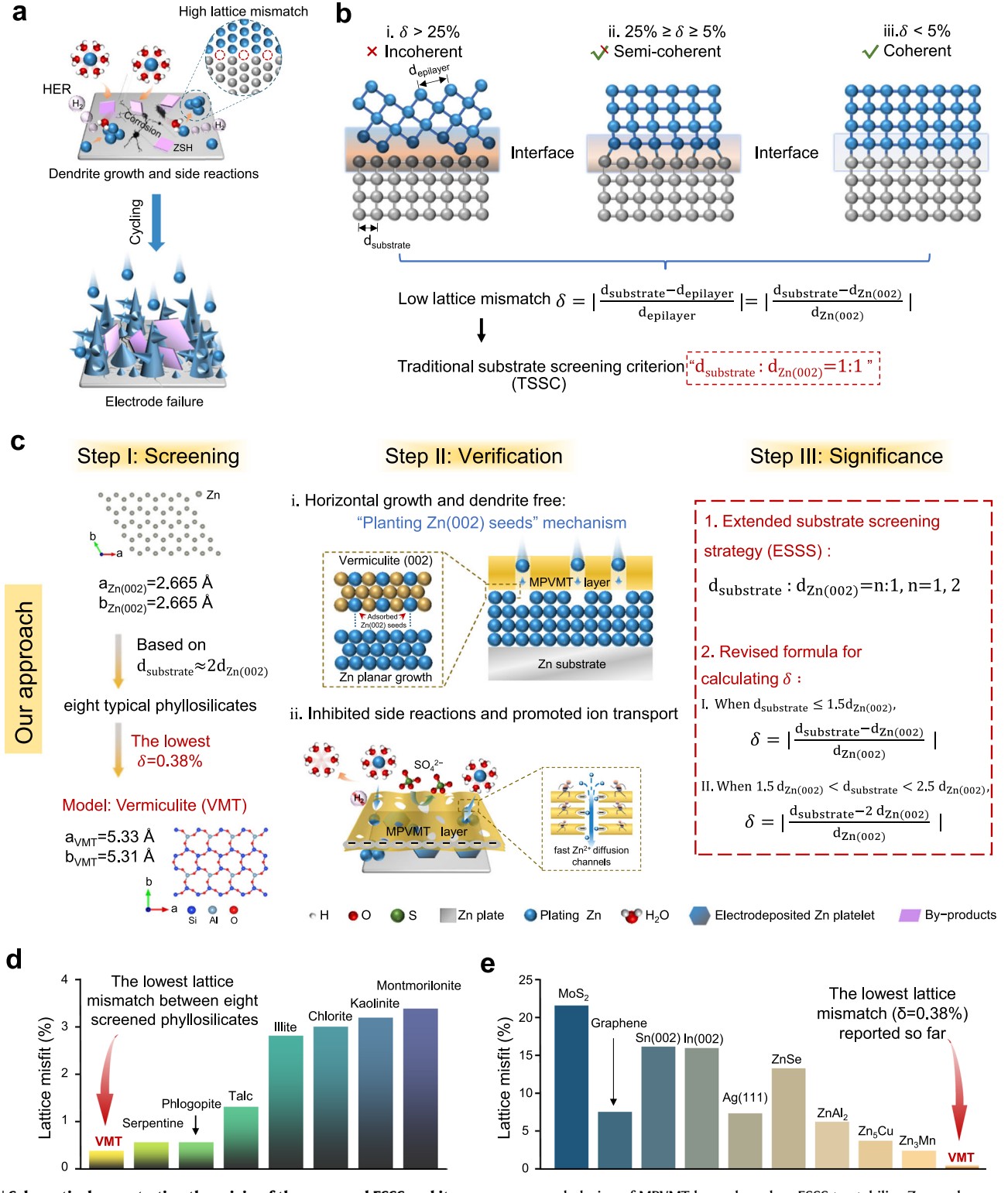

**Fig. 1 | Schematic demonstrating the origin of the proposed ESSS and its verification based on MPVMT coatings. a** Schematic of challenges existing in Zn anodes. **b** Different lattice strain and orientation correlation of Zn electrodeposition formed on substrates with incoherent ($\delta$ > 25%), semi-coherent (25% ≥ $\delta$ ≥ 5%) and coherent ($\delta$ < 5%) interfaces and TSSC used in previous work. **c** Schematic of our work design of MPVMT layers based on ESSS to stabilize Zn anodes. **d** Comparison of lattice mismatch among eight typical phyllosilicates screened based on $d_{substrate}$ : $d_{Zn(002)} = 2$:1. **e** Comparison of lattice mismatch between recently reported substrates and our work.

mismatch, the smaller lattice strain of deposits on substrates, the stronger orientation correlation[29,30]. Hence, it is necessary to search for a suitable substrate with a low lattice mismatch with Zn(002) plane. At present, based on the traditional substrate screening criterion that lattice parameters of substrates should be close to that of Zn(002) (i.e.,

$d_{substrate}$ : $d_{Zn(002)} = 1$:1) (Fig. 1b), several substrates have been reported including two-dimensional (2D) materials (e.g., graphene[24], MoS$_2$[31]) and highly oriented metals (e.g., Sn(200)[32], In(002)[33]). However, most of them form semi-coherent interfaces with Zn(002). Only a few Zn-based alloys[34] or single-crystal Zn(002)[35] anodes enable to form

coherent interfaces. Unfortunately, these alloy or metal substrates are electrically conductive, and thus they cannot isolate Zn electrodes from aqueous electrolytes, resulting in unavoidable side reactions[36,37]. Hence, there is an urgent need to explore possible extended substrate screening strategy (ESSS) to guide the search for substrate materials that not only possess low lattice mismatch with Zn(002), but also inhibit side reactions and promote ion transport, especially at high current densities.

Here, we conjecture and verify the application potential of selecting suitable coatings for Zn anodes based on ESSS, namely $d_{substrate}$: $d_{Zn(002)} = 1 : 1 \rightarrow d_{substrate}$: $d_{Zn(002)} = n : 1$, $n = 1, 2$. First, through calculation, we discover eight typical phyllosilicates that meet "$d_{substrate} \approx 2d_{Zn(002)}$". Among them, vermiculite (VMT) with the lowest lattice mismatch ($\delta = 0.38\%$) reported so far, is selected as our model clay system to study the regulatory mechanism of Zn deposition on "$2d_{Zn(002)}$" substrates and its comprehensive effect on stabilizing Zn anodes protection (Fig. 1c, d). Next, we synthesize monolayer porous vermiculites (MPVMTs) as coatings for Zn anodes (MPVMT@Zn). Notably, the preparation of MPVMT is simple and sustainable, and thus easy to achieve large-scale preparation. Owing to the low lattice misfit and double large lattice of MPVMT than Zn, together with strong interaction with deposited Zn atom, MPVMT layers allow to plant Zn(002) seeds in their planes, thus guiding subsequent horizontal growth of Zn and achieving dendrite-free Zn electrodes. Additionally, the non-conductive MPVMT with hydrophilic groups on the surface effectively mitigates HER and the formation of byproducts. Moreover, the monolayer porous structure with negative charges provides acceleration transport channels for the migration of $Zn^{2+}$, leading to superior cyclability at high current densities. As a result, the MPVMT@Zn symmetric cells exhibit stable cycling for over 800 h at a high current density of 10 mA cm$^{-2}$ and 300 h at a higher current density of 50 mA cm$^{-2}$. Furthermore, practical MPVMT@Zn based 1.25 Ah pouch cell is also demonstrated.

## Results

### Synthesis and characterization

According to the calculation results of adsorption energies of Zn atom on different Zn crystal planes (Supplementary Fig. 1, Supplementary Note 1), it is necessary to search for a substrate highly matched with Zn(002) to induce the planar deposition of Zn and inhibit dendrite growth. Based on the conjecture of ESSS, eight typical phyllosilicate clay materials satisfying "$d_{substrate}$: $d_{Zn(002)} = 2 : 1$" were found by calculation. Then, we computed the lattice mismatches between Zn(002) and eight phyllosilicates and selected VMT as a coating layer for Zn anodes because it exhibits the lowest lattice misfit ($\delta = 0.38\%$) among all clays calculated (Supplementary Fig. 2, Supplementary Table 1). Notably, the lattice mismatch of VMT with the Zn(002) plane is the lowest reported so far, indicating its high potential for stabilizing Zn anodes by inducing the growth of Zn (002) plane (Fig. 1e, Supplementary Table 2).

As shown in Fig. 2a, each layer of VMT consists of one Mg-based octahedral sheet sandwiched between two tetrahedral silicate sheets[38–41]. The cross-sectional scanning transmission electron microscopy (STEM) image of VMT clearly shows that tetrahedral silicate sheets have hollow sites between the atoms. Then, by a one-step ultrasonic and agitating method using only deionized water as the solvent, we easily prepared large quantities of MPVMT dispersion and corresponding powders after freeze-drying with zero pollution (Fig. 2b, Supplementary Fig. 3). The dispersion shows obvious Tyndall effect and a zeta potential of −25.9 mV (Supplementary Fig. 4), indicating negatively charged MPVMT. The negative charges are naturally produced on the layers mainly by $Al^{3+}$ partially substituting $Si^{4+}$ in the tetrahedral sheets, which are balanced by cations (e.g., $K^+$ and $Na^+$) between layers[42]. The AFM image shows typical exfoliated porous vermiculite nanosheets (Fig. 2c), of which the thickness of nanosheets

is all within 2 nm, corresponding to monolayer VMT[43]. MPVMT flakes are also clearly observed by Transmission electron microscopy (TEM) and their main elements (O, Si, Al, Mg and Fe) were characterized by corresponding energy-dispersive X-ray spectroscopy mapping (Supplementary Fig. 5). Furthermore, the enlarged TEM image determines that the pore size of 2D MPVMTs ranges from 10 nm to 30 nm (Fig. 2e) and the image of selected area electron diffraction in the inset indicates their high crystallinity. The unique porous structure establishes the basis for the rapid transport of ions. In contrast, nonporous vermiculites (NVMT) with thicknesses of approximately 2 nm were controllably synthesized by regulating the power of the ultrasonic homogenizer (Supplementary Fig. 6).

The MPVMT layers were coated onto the Zn surface by the spray coating method. The X-ray diffraction (XRD) pattern indicates that the Zn electrode remains unchanged after surface modification (Supplementary Fig. 7). The typical thickness of the coatings is around 1.1 μm measured by scanning electron microscopy (SEM, Supplementary Fig. 8). The hydrophilicity of MPVMT layer was tested through static contact angle measurements. The water contact angle on bare Zn is 95°, while the contact angle on MPVMT@Zn is 27° (Fig. 2f, g), revealing significantly enhanced hydrophilicity due to the formation of strong hydrogen bonds between the silanol groups on the surface of MPVMTs and the oxygen in water molecules[44–46]. Kelvin probe force microscopy (KPFM) reveals that the surface potential of MPVMT@Zn is about 300 mV lower than that of bare Zn (Fig. 2h, i), suggesting that the surface of MPVMT coatings is more electronegative. Additionally, insulating MPVMT layers exhibit an electronic resistivity of approximately $1.4 \times 10^5 \Omega$ cm (Fig. 2j, Supplementary Note 2)[47]. Moreover, owing to the simple preparation process, the MPVMT@Zn electrode can be easily scaled up to 300 cm$^2$ (Fig. 2k).

### Horizontal growth of Zn deposition

The morphology and texture of Zn deposits greatly influence the lifespan of Zn anode and a uniform parallel deposition enables the cell to stably operate for a long time without a short circuit[48]. First, Zn||Ti cells are used to study the deposition morphology of Zn underneath the MPVMT layer. As shown in Fig. 3a, Zn deposits on the bare Ti surface show an anisotropic platelet-like dendritic morphology at 3 mA cm$^{-2}$/3 mAh cm$^{-2}$, accounting for the inhomogeneous nucleation of $Zn^{2+}$. In contrast, for the MPVMT-coated Ti (MPVMT@Ti), the surface is covered with 2D nanosheets which illustrates strong adhesion between MPVMT and substrates, and no Zn platelet is observed which reveals that Zn deposits underneath the insulating coatings (Supplementary Fig. 9a). After removing MPVMTs by high-power ultrasonication, horizontally stacked Zn platelets are shown in Fig. 3b. When further increasing the current density/capacity to 10 mA cm$^{-2}$/ 10 mAh cm$^{-2}$, more randomly distributed vertical zinc platelets are observed on the surface of bare Ti (Fig. 3c). For MPVMT@Ti, the coating remains intact on the surface (Supplementary Fig. 9b) and Zn platelets plating under the coatings are still arranged horizontally and even more closely (Fig. 3d). Then, the 2D XRD results reveal that Zn deposits with MPVMT protection have a stronger intensity of (002) plane and weaker intensity of (100) plane than that of the bare Zn deposits at both 3 mA cm$^{-2}$/3 mAh cm$^{-2}$ and 10 mA cm$^{-2}$/10 mAh cm$^{-2}$ (Fig. 3e−h). Corresponding 1D XRD patterns assure the (002) plane orientation of Zn deposits with MPVMT layers (Supplementary Fig. 10). Notably, with the increase of current density, the effect on inducing the growth of Zn(002) plane by MPVMTs becomes more prominent. The $I_{(002)}/I_{(100)}$ value of Zn deposits with MPVMT protection is 1.7 times that without protection at 3 mA cm$^{-2}$ and increases to 5 times at 10 mA cm$^{-2}$ (Fig. 3i). In the crystal structure of Zn (Fig. 3j), (100) crystal planes arranged in an unsmooth and wavy pattern are the main culprit of dendrite growth, while the crystallographic orientation of (002) crystal planes with relatively smooth surface enable dendrite-free metal anode[17]. This verifies that the growth of Zn(002) plane promoted

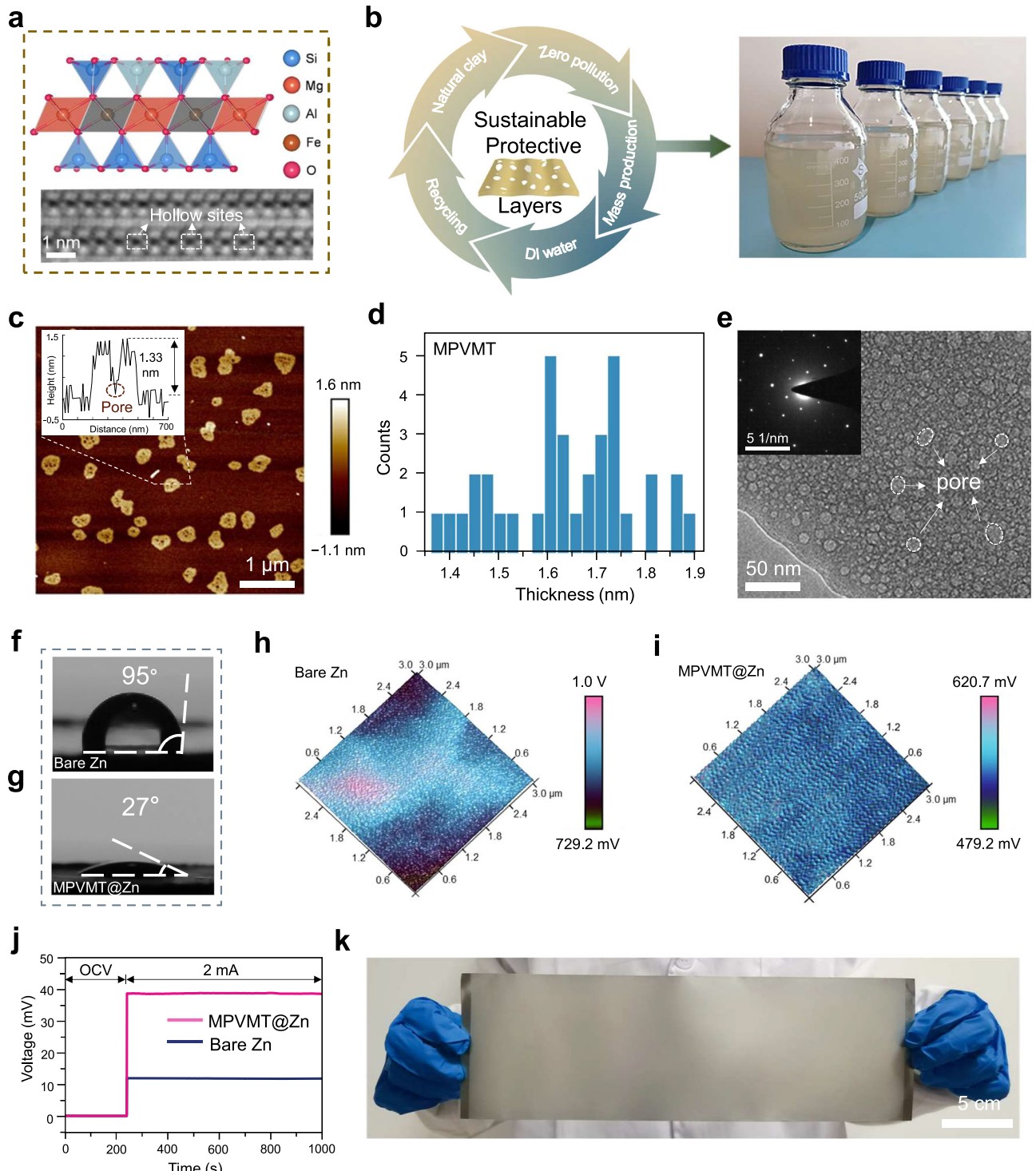

**Fig. 2 | Preparation and characterization of MPVMT@Zn anode. a** Cross-sectional schematic of a monolayer vermiculite and cross-sectional STEM image of unexfoliated vermiculite using the high-angle annular dark-field mode. **b** Schematic of the mass production of MPVMT materials through a sustainable method. **c** Atomic force microscopy (AFM) image of MPVMT nanosheets. The inset is the height profile of the corresponding lines. **d** Statistics for the thickness of MPVMT nanosheets. **e** Typical TEM image and SAED pattern (inset) of MPVMT. Contact angles of bare Zn (**f**) and MPVMT@Zn (**g**). KPFM images of MPVMT@Zn (**h**) and bare Zn (**i**). **j** Voltage response to a current of 2 mA of MPVMT@Zn and bare Zn electrodes. **k** The prepared MPVMT@Zn with an area of 300 cm$^2$ fabricated using a scalable strategy.

by MPVMT layers is conducive to inhibiting dendrite formation. Furthermore, deposition morphology on the Zn surface is shown in Supplementary Fig. 11 and horizontally arranged platelets underneath the MPVMT layer are both observed from the front and cross section, consistent with the morphology on MPVMT@Ti. In addition, in situ monitoring of Zn deposition process with homemade bare Zn and MPVMT@Zn symmetric cells was performed by an optical microscope under a current density of 20 mA cm$^{-2}$. Figure 3k shows that uneven Zn morphology with obvious protrusions and hydrogen bubbles appears on the bare Zn anode after an initial 10 min. As the deposition time

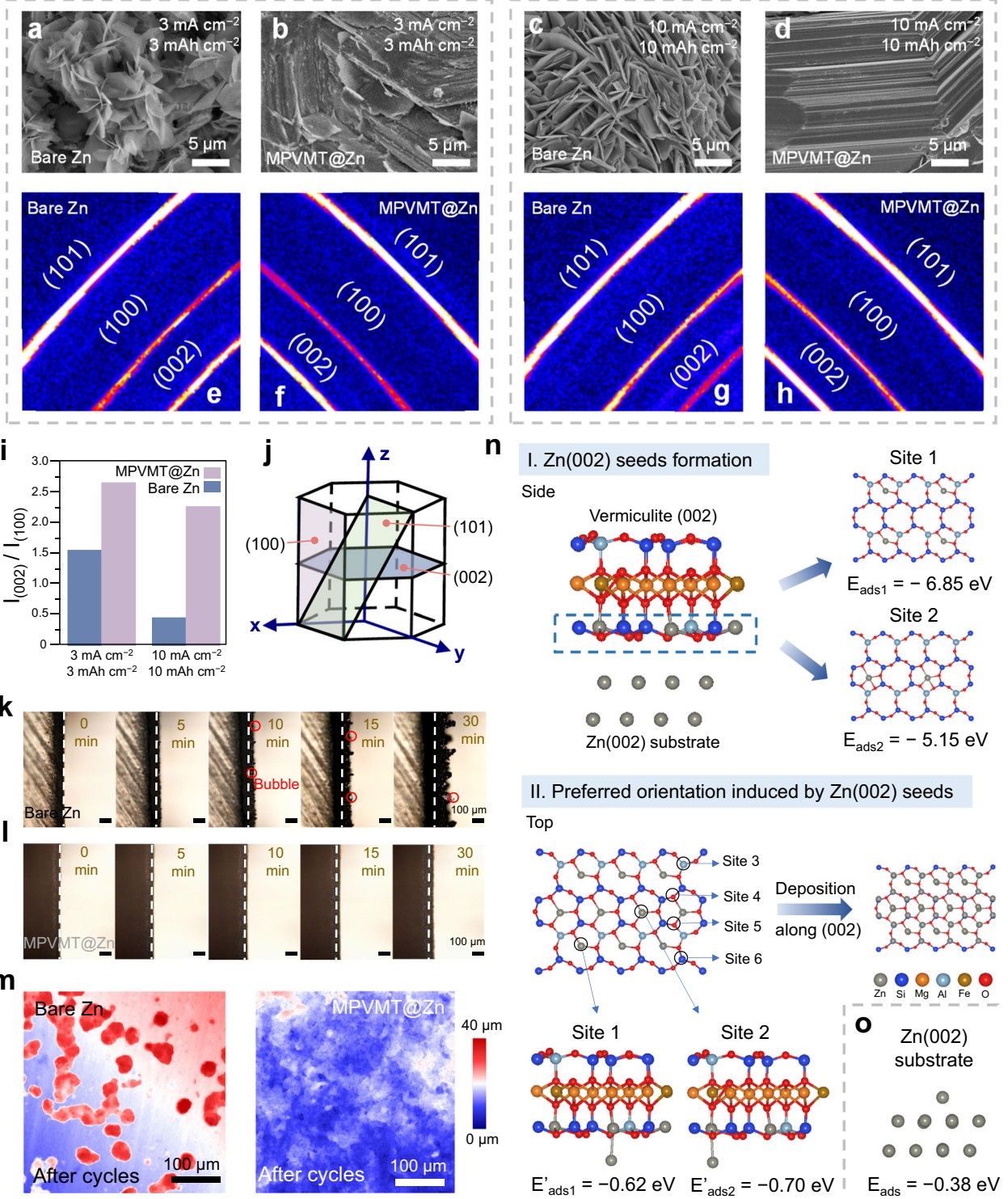

**Fig. 3 | Characterization and theoretical simulation of Zn deposition behavior.** SEM images of Zn electrodeposits on bare Ti (**a**, **c**) and underneath a MPVMT layer (**b**, **d**) at different current densities. Corresponding 2D XRD of Zn electrodeposits on bare Ti (**e**, **g**) and underneath a MPVMT layer (**f**, **h**). **i** Intensity ratio of Zn (002) and Zn (100) in corresponding 2D XRD patterns. **j** The crystalline structure of Zn metal. In situ optical observations of Zn electrodeposition on bare Zn (**k**) and underneath a MPVMT layer (**l**) at 20 mA cm⁻². **m** CLSM 2D height images of bare Zn and MPVMT@Zn after 10 cycles of plating/stripping (10 mA cm⁻²,1 mAh cm⁻²). **n** The process of Zn(002) growth induced by MPVMT layers, including Zn(002) seeds formation and preferred orientation growth based on Zn(002) seeds, and corresponding calculated adsorption energy of Zn atom on the surface of MPVMT or semi-filled Zn(002)/ MPVMT substrate. **o** The adsorption energy of Zn atom on the surface of Zn substrate.

increases to 30 min, the protrusions maintain and grow into dendrites. In comparison, the deposition on MPVMT@Zn is uniform and no obvious dendrites and bubbles are observed during the whole plating process, demonstrating the suppression of dendrite growth and side

reaction (Fig. 3l). The morphology changes of Zn anodes after cycling at 10 mA cm⁻²/1 mAh cm⁻² are also investigated. Confocal laser scanning microscope (CLSM) images show that small "islands" of Zn blocks form on the bare Zn surface after 10 cycles, whereas the surface of

MPVMT@Zn anodes maintains smooth (Fig. 3m). After 200 cycles, randomly distributed Zn dendrites are observed on the surface of bare Zn, while 2D MPVMT nanosheets on MPVMT@Zn anodes are not peeled off showing high adhesion and excellent chemical stability and parallel Zn platelets are consistently sustained under nanosheets (Supplementary Figs. 12, 13).

We further study the interaction between Zn (002) deposits and the MPVMT by DFT calculations. As shown in Fig. 3n and Supplementary Fig. 14, the plating Zn atoms prefer to adsorb in hollow sites of the vermiculite layer instead of other sites, owing to their strong adsorption energy ($E_{ads1}$ = −6.85 eV, $E_{ads2}$ = −5.15 eV). Benefitting from low lattice misfit (δ = 0.38%) between MPVMT layers and the Zn(002) crystal plane, the deposited Zn are arranged in the manner of Zn(002) to form semi-filled Zn(002)/ MPVMT composite substrate. Subsequently, Zn atoms deposited later continue to preferentially grow on the Zn(002) seeds of semi-filled Zn(002)/ MPVMT composite substrate ($E'_{ads1}$ = −0.62 eV, $E'_{ads2}$ = −0.70 eV, other possible adsorption sites see Supplementary Fig. 15 for details), rather than adsorbing on the Zn substrate ($E_{ads}$ = −0.38 eV, Fig. 3o), which contributes to the parallel growth of Zn along the (002) plane. It is concluded that the strong adsorption of MPVMT layers to Zn atoms and the super match allow for "planting Zn(002) seeds" on the "$2d_{Zn(002)}$" substrate, thereby achieving the subsequent preferred deposition and alleviating dendrite growth even at high current densities. The above evidence confirms the validity of ESSS for screening substrates that can induce the orientation of Zn electrodeposits. Based on ESSS, we have modified the formula for calculating the lattice mismatch δ as follows:

$$\delta = \begin{cases} |\frac{d_{substrate} - d_{Zn(002)}}{d_{Zn(002)}}|, d_{substrate} \le 1.5\, d_{Zn(002)} \\ |\frac{d_{substrate} - 2d_{Zn(002)}}{d_{Zn(002)}}|, 1.5\, d_{Zn(002)} < d_{substrate} < 2.5\, d_{Zn(002)} \end{cases} \quad (1)$$

where $d$ is the lattice parameter of the substrate or Zn(002) plane.

## Inhibition of side reactions

The side reactions mainly include the HER process and generation of $Zn_4SO_4(OH)_6 \cdot xH_2O$ by-products formed with the decreased concentration of $H^+$ [26]. As shown in Fig. 4a, in situ gas chromatography (GC) was used to determine HER of Zn symmetric cells during the Zn stripping/plating with a simple configuration at a current density/capacity of 20 mA cm$^{-2}$/2 mAh cm$^{-2}$ for 90 min. The peak intensity of $H_2$ for bare Zn anodes dramatically increases after cycling, whereas very little $H_2$ is captured for the MPVMT@Zn anodes (Fig. 4b). Moreover, many bubble footprints can be seen on the surface of cycled bare Zn (Fig. 4c). Owing to the interfacial turbulence and nonuniform current distribution caused by the generation of $H_2$, uneven deposition of Zn around the bubbles occurs, making it easier to form dendrites. On the contrary, due to the isolation of the electrolyte and Zn electrode, the cycled MPVMT@Zn maintains the smooth surface and demonstrates the effectively suppressed HER (Fig. 4d). Next, linear sweep voltammetry (LSV) measurements in 1 M $Na_2SO_4$ electrolyte were carried out as shown in Fig. 4e. The current density of MPVMT@Zn is always smaller than that of the bare Zn, and the corresponding Tafel slope of MPVMT@Zn (291.3 mV dec$^{-1}$) is much larger compared with that of the bare Zn (219.1 mV dec$^{-1}$), manifesting low kinetics of HER under MPVMTs protection. Furthermore, the corrosion current density of MPVMT@Zn (2.163 mA cm$^{-2}$) is also lower than that of the bare Zn (3.099 mA cm$^{-2}$), revealing prominent corrosion resistance of MPVMT layers (Fig. 4f). The inhibited HER can be ascribed that insulating and hydrophilic MPVMTs effectively absorb water molecules and prevent them from gaining electrons to decompose. Benefitting from the suppressed HER and repulsion of surface negative charge of MPVMT layers on sulfate ($SO_4^{2-}$), by-product formation is restrained. Supplementary Fig. 16 shows that the MPVMT@Zn keeps a flat and smooth surface without any trace of corrosion even after immersing in 2 M $ZnSO_4$ electrolytes for 6 days, implying better chemical stability than the bare Zn. Meanwhile, according to XRD patterns, the peak density of $Zn_4SO_4(OH)_6 \cdot xH_2O$ on the surface of MPVMT@Zn is much weaker than that of bare Zn (Fig. 4g), again confirming the effective restriction of by-products using MPVMT layers.

In addition, to better elucidate Zn deposition behavior under the influence of side reactions, the electric field distribution of the plating process for bare Zn and MPVMT@Zn was simulated, and geometric models are shown in Supplementary Fig. 17. Regarding bare Zn, $H_2$ bubbles generated during plating lead to intense interfacial turbulence, resulting in uneven distribution of current density and $Zn^{2+}$ concentration. As time increases from 0 s to 3600 s, the evolution of severe and messy dendrite growth process is shown in Fig. 4h. As a control, the electric field distribution at the interface without $H_2$ bubbles was also simulated, as shown in Supplementary Fig. 18. Although the disappearance of bubbles alleviates the interfacial disturbance to a certain extent, the inability of bare Zn to homogenize the $Zn^{2+}$ flux still leads to dendritic Zn growth. Thanks to the suppressed HER and negative charge layer, MPVMT@Zn anodes without interfacial disturbance endow even $Zn^{2+}$ distribution and dendrite-free deposition (Fig. 4i).

## Fast ion transfer and rapid de-solvation

Rapid $Zn^{2+}$ transport through protective coatings is urgently needed for Zn anodes, particularly at high current densities[49]. To investigate the $Zn^{2+}$ electrochemical behavior with MPVMT layers, COMSOL simulations were conducted and the comparison models for MPVMT@Zn and NVMT@Zn anodes were established to better uncover the role of pore structures in facilitating ion transport (Supplementary Fig. 19). As shown in Fig. 5a, the edges of NVMTs, served as transport channels of ions, exhibit stronger electric fields than bare Zn surface resulting from the accumulation of negative charge from NVMT layers in the edge. Meanwhile, the surface of MPVMTs shows more and stronger accelerating electric fields around pores than that of NVMTs. As a result, when $Zn^{2+}$ fluxes penetrate the MPVMT layers (Fig. 5b), their transport pathway is greatly shortened than NVMT layers because of abundant channels established by pore structures, and the migration speed simultaneously increases under the acceleration of the local electric fields around the pores, thereby achieving fast $Zn^{2+}$ transport.

To experimentally certify the ion transport acceleration of MPVMT layers, electrochemical impedance spectroscopy (EIS) results are shown in Fig. 5c. Apparently, the MPVMT@Zn anode exhibits the lowest charge transfer resistance ($R_{ct}$) among three anodes, while the $R_{ct}$ of NVMT@Zn is lower than the bare Zn, verifying the improving transport of $Zn^{2+}$ by designs of pores and local acceleration fields. Furthermore, the plating/stripping voltage curves of bare Zn, NVMT@Zn and MPVMT@Zn based on their rate performance testing were measured with different current densities. As shown in Fig. 5d, the voltage hysteresis of MPVMT@Zn is always smaller than that of bare Zn. When the current increases to 10 mA cm$^{-2}$, the polarization of bare Zn soars to 162 mV owing to poor hydrophilicity and sluggish ion transfer, while that of MPVMT@Zn is only 114 mV. Regarding the NVMT@Zn anode, it displays a narrow voltage gap, almost the same as MPVMT@Zn, under small current densities (0.5 and 1 mA cm$^{-2}$), but exhibits even larger voltage hysteresis (167 mV) than bare Zn at 10 mA cm$^{-2}$ (Supplementary Fig. 20a, b). The "anomalous" behavior of the NVMT@Zn under different current densities might be derived from the shortcomings of its limited acceleration effect and tortuous migration path for $Zn^{2+}$ which are fatal particularly under high current densities (Supplementary Fig. 20c, d, Supplementary Note 3), again proving the advantage of MPVMT with numerous acceleration channels to significantly reduce the overpotential of the batteries.

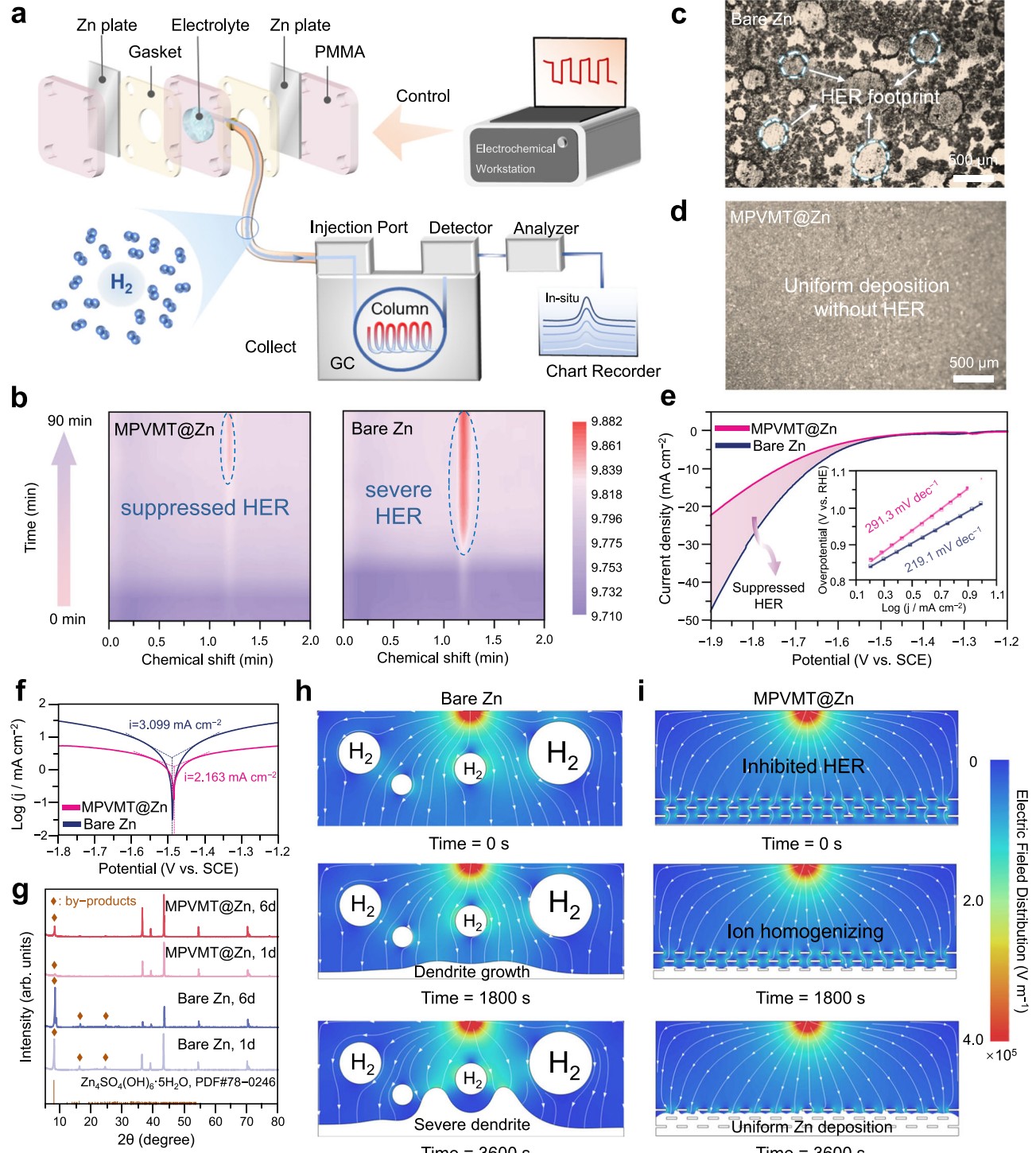

**Fig. 4 | Characterization and theoretical simulation of H$_2$ evolution behavior during Zn plating/stripping. a** Schematic of the configuration used for observing H$_2$ evolution in situ. **b** In situ electrochemical GC profile for bare Zn and MPVMT@Zn symmetric cells at 20 mA cm$^{-2}$ and 2 mAh cm$^{-2}$. Corresponding optical microscopy images of bare Zn (**c**) and MPVMT@Zn (**d**) after operando characterization. **e** LSV curves using 1 M aqueous Na$_2$SO$_4$ solution as the electrolyte with the scan speed of 2 mV s$^{-1}$ and corresponding Tafel slopes (inset). **f** Tafel plots demonstrating corrosion of Zn anodes in 2 M aqueous ZnSO$_4$ electrolyte. **g** XRD patterns of bare Zn and MPVMT@Zn after immersing in 2 M ZnSO$_4$ electrolyte for 1 and 6 days. **h**, **i** Electric field simulation for Zn plating process at the electrolyte/electrode interface. Bare Zn with severe HER and interfacial turbulence at 0 s, 1800 s and 3600 s (**h**). MPVMT@Zn with suppressed HER and interfacial turbulence at 0 s, 1800 s and 3600 s (**i**).

Ion transport behavior is further determined experimentally. Zn$^{2+}$ transference number (ZTN) is evaluated based on chronoamperometry measurements and EIS results before and after polarization. The MPVMT layers display a ZTN of 0.67, three times higher than the bare Zn (0.20) (Supplementary Fig. 21 and Supplementary Note 4). Moreover, as shown in Supplementary Fig. 22, the ionic conductivity of MPVMT@Zn is calculated to be 29.79 mS cm$^{-1}$, that is, 2.8 times greater than that of bare Zn (10.50 mS cm$^{-1}$). These results again confirm that MPMVT accelerates Zn$^{2+}$ transport and thus promotes ionic conductivity[50]. Specifically, when hydrated Zn$^{2+}$ transport

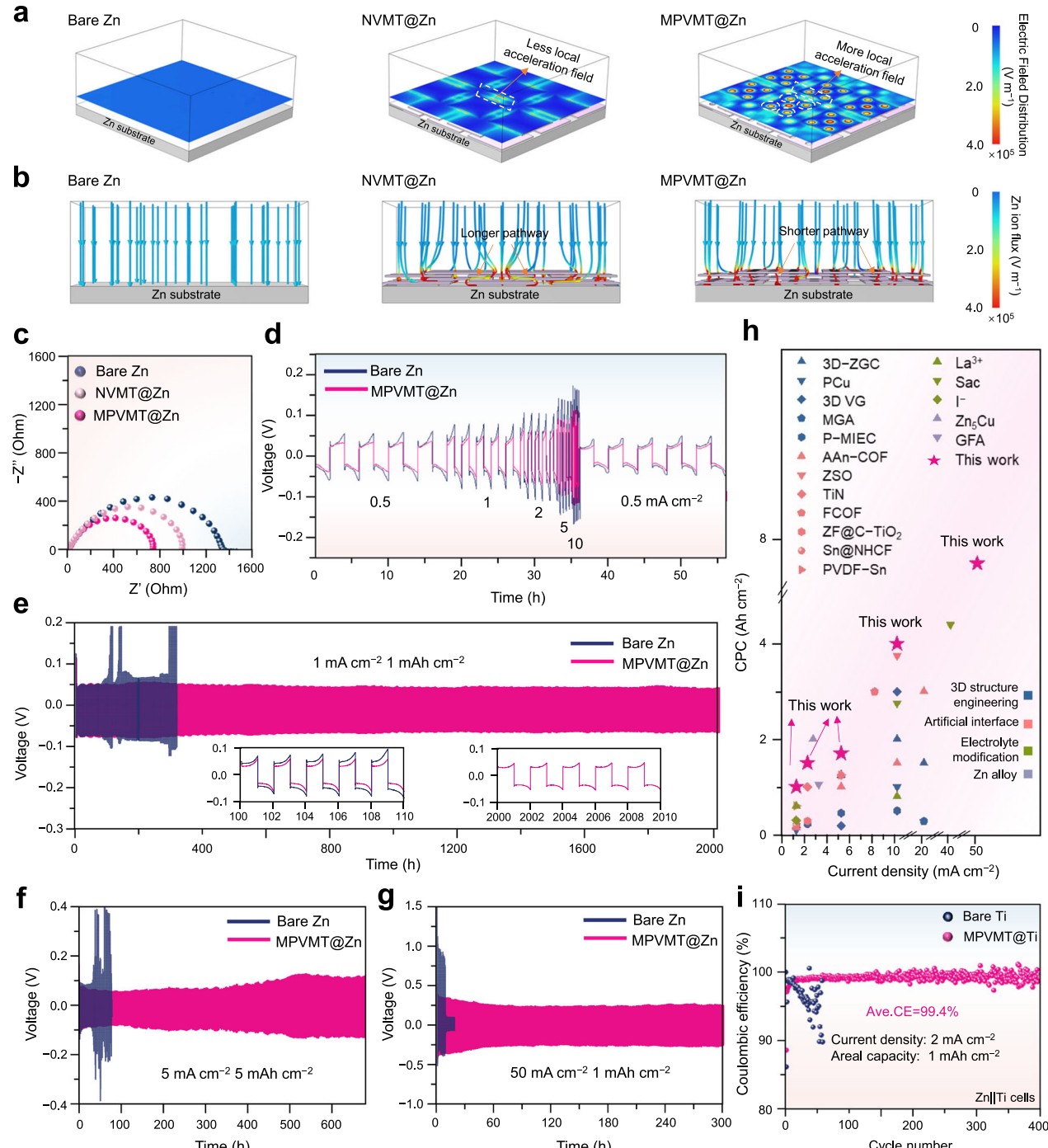

**Fig. 5 | COMSOL simulations of ion transfer and the electrochemical performance of Zn anodes.** Electric field distribution (**a**) and Zn ion flux (**b**) of bare Zn, NVMT@Zn and MPVMT@Zn at 10 mA cm⁻². **c** Nyquist plots of the bare Zn, NVMT@Zn and MPVMT@Zn symmetric cell at initial state. **d** Rate performance of the bare Zn and MPVMT@Zn electrodes at different current densities from 0.5 to 10 mA cm⁻² with the same capacity of 1 mAh cm⁻². Cycling performance of symmetric cells with or without MPVMT layer protection at 1 mA cm⁻² (**e**), 5 mA cm⁻² (**f**) and 50 mA cm⁻² (**g**). **h** Performance comparison between MPVMT@Zn and recently reported Zn anodes using different strategies. **i** CE of Zn||Ti cells with a cut-off charging voltage of 0.5 V at 2 mA cm⁻² and 1 mAh cm⁻².

through MPVMT nanochannels is driven by electric field force from the battery and the coating, water molecules coordinated with Zn²⁺ will be captured because of the strong hydrogen bonding between silanol groups on MPVMT surface and the oxygen in water molecules[20]. Consequently, the de-solvation of Zn²⁺ is effectively promoted by silanol groups. To further assure the effect of MPVMT layers to facilitate de-solvation, the activation energy ($E_a$) which represents the de-solvation barrier for Zn²⁺ transport is identified by temperature-dependent EIS (Supplementary Fig. 23) and calculated according to the

Arrhenius equation[14,18]:

$$\frac{1}{R_{ct}} = A \exp\left(-\frac{E_a}{RT}\right) \tag{2}$$

where $R_{ct}$, $A$, $R$ and $T$ represent the charge transfer resistance, pre-exponential factor, molar gas constant (8.3145 J mol⁻¹ K⁻¹) and absolute temperature, respectively. Based on the fitting equivalent circuit, $R_{ct}$ at different temperatures is obtained (Supplementary Table 4). The

resultant $E_a$ of MPVMT@Zn (59.9 kJ mol$^{-1}$) is lower than that of the bare Zn (64.9 kJ mol$^{-1}$), verifying the faster de-solvation of hydrated Zn$^{2+}$. These results have suggested that MPVMTs are remarkable ionic conductors due to their inherent ordered channels and facilitated electric fields to promote Zn$^{2+}$ transfer.

## Electrochemical performance

The parallel growth of Zn deposits, suppressed side reactions and fast ion transport achieved by MPVMT layers are expected to significantly improve the electrochemical performance of Zn anodes. First, the galvanostatic cyclic performance of symmetric cells was performed at various current densities and areal capacities. As shown in Fig. 5e, the Zn||Zn symmetric cell exhibits a sudden and irreversible voltage increase only after cycling for 112 h at a current density/capacity of 1 mA cm$^{-2}$/1 mAh cm$^{-2}$. In contrast, MPVMT@Zn symmetric cells show prolonged cycle life for over 2000 h, approximately 18 times longer than the bare Zn. The MPVMT@Zn electrode also exhibits excellent cycling stability at increasing current capacity of 2 mA cm$^{-2}$/2 mAh cm$^{-2}$ for 1500 h, 5 mA cm$^{-2}$/5 mAh cm$^{-2}$ for 680 h (Fig. 5f), 0.6 mA cm$^{-2}$/10 mAh cm$^{-2}$ for 500 h and 10 mA cm$^{-2}$/10 mAh cm$^{-2}$ for 400 h (Supplementary Fig. 24), much better than the bare Zn. Furthermore, at a higher current density of 10 mA cm$^{-2}$/1 mAh cm$^{-2}$ and even 50 mA cm$^{-2}$/1 mAh cm$^{-2}$, the MPVMT@Zn||MPVMT@Zn cell still stably operates for over 800 h with the cumulative plating capacity (CPC) of 4000 mAh cm$^{-2}$ (Supplementary Fig. 24b) and 300 h with the CPC of 7500 mAh cm$^{-2}$ (Fig. 5g), respectively. It is proved that MPVMT layers demonstrate an excellent ability to control dendrite growth and stabilize Zn anodes even under the superhigh current and deposition capacity. Notably, the current density and CPC enabled by the MPVMT@Zn symmetric cell are much higher than most of the previously reported values from Zn electrodes based on different modification strategies (Fig. 5h, see Supplementary Table 5 for details).

To reflect the plating/stripping reversibility of Zn under MPVMT protection, CE measurements were performed by Zn||Ti asymmetric cells at a current density/capacity of 2 mA cm$^{-2}$/1 mAh cm$^{-2}$. As shown in Fig. 5i, the MPVMT@Ti||Zn cells present stable operation for 400 cycles with an average CE of 99.4%, manifesting favorable durability and superior reversibility. However, bare Ti||Zn cells merely run for 30 cycles with evident fluctuation of the CE, indicating serious side reactions and dendrite growth. Because the bare Ti cannot regulate the zinc deposition and hinder by-product formation, its voltage hysteresis (123 mV at 5th cycles) is much larger than MPVMT@Ti (58 mV at 5th cycles), as shown in Supplementary Fig. 25. Hence, it can be concluded that the multifunctional MPVMT layers greatly regulate Zn deposition behavior, restrain corrosion and promote Zn$^{2+}$ kinetics, leading to high-rate and stable MPVMT@Zn anodes.

## Full cell performance

To explore the practical application of MPVMT@Zn anode, Zn-ion full cells are assembled by coupling with a MnO$_2$ cathode (MPVMT@Zn||MnO$_2$). The XRD pattern shows the successful synthesis of β-MnO$_2$ (Supplementary Fig. 26). The cyclic voltammetry (CV) profile of MPVMT@Zn||MnO$_2$ full cell displays predominant smaller voltage polarization (33 mV, difference in the cathodic peaks) than the bare Zn||MnO$_2$ cell, which indicates its advantageous Zn$^{2+}$/Zn reaction kinetics (Fig. 6a). The EIS results of MPVMT@Zn||MnO$_2$ cells also confirm smaller charge transfer resistance (Supplementary Fig. 27). The small polarization and impedance can be ascribed to the restrained side reactions and ion acceleration effect achieved by electronegative MPVMT interface. Next, the rate performance at different current densities was investigated and the capacities of both batteries are almost the same at around 250 mAh g$^{-1}$ at 1 C (1 C = 308 mA g$^{-1}$, based on the mass of MnO$_2$) (Fig. 6b). When the high current density (10 C) is used, MPVMT@Zn-based full cells

exhibit more than twice the capacity of bare Zn-based full cells, indicating their superior capacity retention. Moreover, the charging and discharging curves at various current densities manifest a smaller voltage gap of MPVMT@Zn||MnO$_2$ than bare Zn||MnO$_2$ (Supplementary Fig. 28), consistent with CV curves. Furthermore, the long cycling performance in Fig. 6c shows that the capacity of bare Zn-based full cells rapidly declines to 50 mAh g$^{-1}$ after 500 cycles, while MPVMT-based full cells sustain a high capacity of 123 mAh g$^{-1}$, which can also be evidenced by polarization curves (Fig. 6d, e). To gain insight into the performance improvement of full cells, the 3D surface morphologies of MPVMT@Zn and bare Zn electrodes after 500 cycles were analyzed (Fig. 6f, g). The cycled MPVMT@Zn anode exhibits a smooth and even surface, while harsh island-like Zn dendrites are observed on bare Zn, implying that the MPVMT overlayer on the Zn foil enables more uniform plating/stripping of Zn. These results again validate the virtue of MPVMT design (Supplementary Table 6). We also assembled zinc-iodine (Zn||I$_2$) practical full cells using iodine-containing electrolytes and activated carbon (AC) as the host material with the high mass loading of about 20 mg cm$^{-2}$. As shown in Fig. 6h, under harsh conditions of high Zn utilization (51%, the thickness of Zn: 10 μm) and low N/P ratio (1.9), the MPVMT coatings enable steady cycling of the Zn anodes and maintain a capacity of almost 100% after 200 cycles, which is over four times higher than the bare Zn||I$_2$ cells (capacity retention: 24%). To further verify the device's feasibility, MPVMT@Zn anodes were assembled into pouch cells. Supplementary Fig. 29 shows that 2 × 3 cm$^2$ MPVMT@Zn||I$_2$ pouch cells show stable cycling without capacity decay and high CE (99.93%) after 1000 cycles. Additionally, we demonstrate a 13 × 15 cm$^2$ MPVMT@Zn||MnO$_2$ pouch cell that delivers cell capacity of 1.25 Ah and energy density of 75 Wh L$^{-1}$, which outperforms most of the previously reported ZIBs (Fig. 6i–k, Supplementary Tables 7 and 8, Supplementary Note 5)[10,36,51–57].

## Discussion

In summary, we develop and confirm the applicability of ESSS ($d_{substrate}$: $d_{Zn(002)} = 1:1 \rightarrow d_{substrate}$: $d_{Zn(002)} = n:1$, $n = 1, 2$). Taking the vermiculite ($d_{vermiculite} \approx 2d_{Zn(002)}$) as a model, we prepare 2D hydrophilic and insulating MPVMTs by a simple and eco-friendly method as protective coatings for Zn anode. Due to MPVMT's low lattice mismatch (δ = 0.38%), "$2d_{Zn(002)}$" structure cell and strong adsorption to Zn deposits, the coatings induce (002) oriented growth of Zn by "planting Zn(002) seeds" mechanism, realizing dendrite-free anodes. Additionally, the hydrophilic functional groups and negative charges on the surface of MPVMTs greatly promote the de-solvation of Zn$^{2+}$ and homogenize the electric field, synergistically suppressing side reactions. Furthermore, the constructed ion acceleration channels significantly facilitate Zn$^{2+}$ transport, achieving high-rate and long-life MPVMT@Zn anodes. Consequently, the MPVMT@Zn anodes deliver a superhigh cumulative capacity of 7500 mAh cm$^{-2}$ at a current density of 50 mA cm$^{-2}$, outperforming most reported modified Zn anodes. The proposed ESSS and nucleation mechanism provide unique insights into highly reversible Zn metal batteries and may apply to other metal-based batteries.

## Methods

### Preparation of monolayer porous/nonporous two-dimensional VMT

First, 12 g vermiculite (VMT, sizes: 2–3 mm, Sigma-Aldrich) was mixed with 500 mL deionized (DI) water (18.2 MΩ, Millipore system) at room temperature in a 500 mL beaker to prepare the suspension liquid, followed by transferring to the ultrasonic homogenizer (1200 W, Φ 20 mm, Biosafer 1200–98). Then, the suspension under vigorous stirring conditions was dispersed with the ultrasonic power of 360 W and pulse mode of 5 s on and 5 s off in a circulating water bath (6 °C) for 6 h to obtain the crude dispersion. Subsequently, 20 mL dispersion was

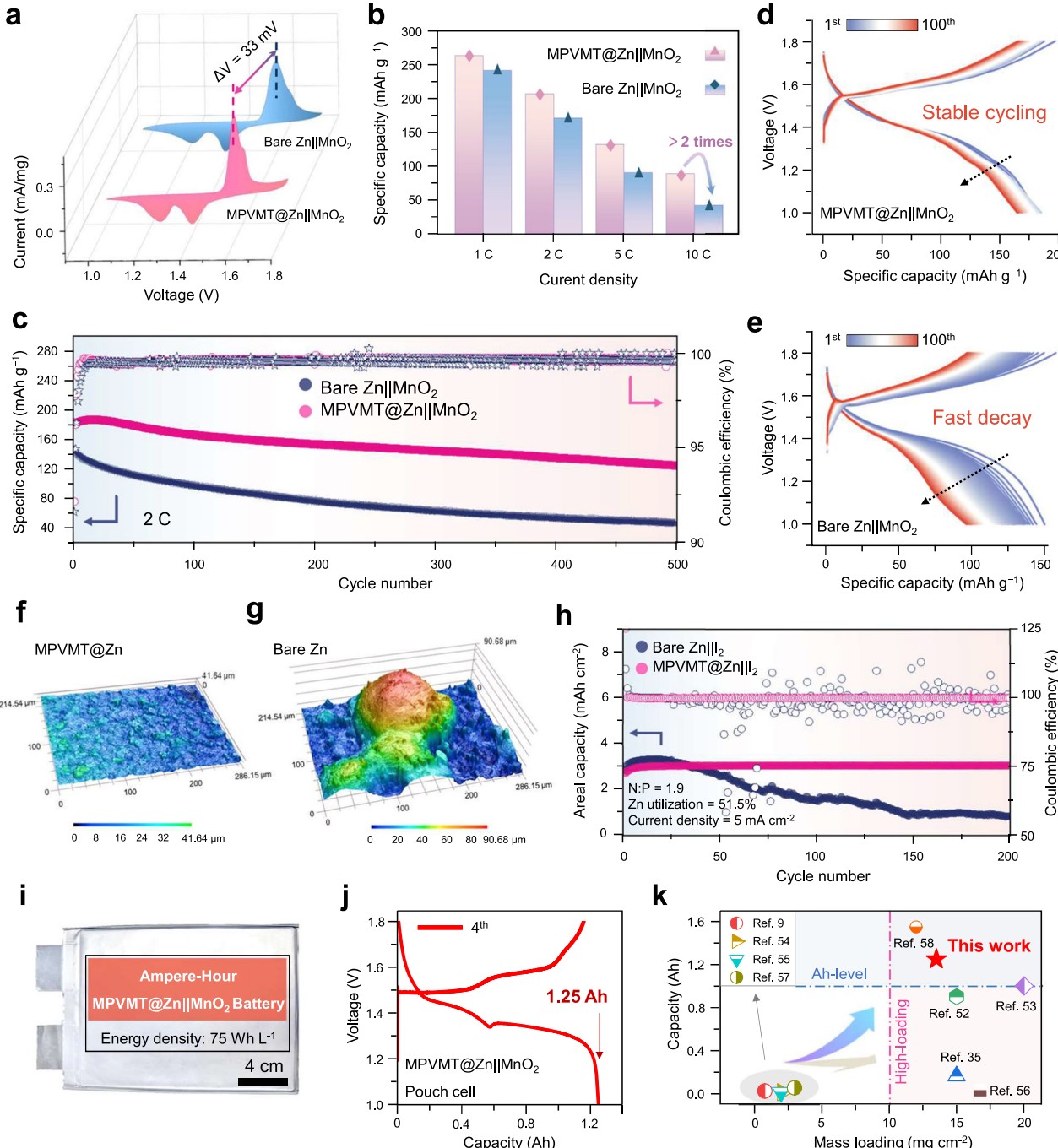

**Fig. 6 | Electrochemical performance of full cells and pouch cells. a** CV profiles of Zn | |MnO$_2$ cells based on bare Zn and MPVMT@Zn anodes with the scan speed of 0.1 mV s$^{-1}$. **b** Rate performance of the liquid full batteries at current densities from 1 C to 10 C. Cycling performance of liquid full cells with 2 M ZnSO$_4$ and 0.1 M MnSO$_4$ electrolyte at 2 C (**c**) and corresponding voltage profiles of 100 cycles for MPVMT@Zn ||MnO$_2$ cells (**d**) and bare Zn | |MnO$_2$ cells (**e**). CLSM 3D height images of MPVMT@Zn (**f**) and bare Zn (**g**) electrodes after 500 cycles. **h** Cycling performance of Zn | |I$_2$ full cells based on bare Zn and MPVMT@Zn anodes with 2 M ZnSO$_4$ and 0.5 M KI electrolyte at 5 mA cm$^{-2}$. Optical image of an Ah-level MPVMT@Zn | | MnO$_2$ pouch cell (**i**) and corresponding voltage profile of 4th cycle at 0.1 C (**j**). **k** Comparison of cell capacity of MPVMT@Zn ||MnO$_2$ pouch cell with previous works.

centrifuged at 424 × $g$ for 10 min to retain the supernatant slurry, followed by freeze drying to obtain the porous VMT (MPVMT) powder. Nonporous VMT (NVMT) was prepared according to the same method, except that the ultrasonic power was changed to 120 W.

### Preparation of MPVMT@Zn and MPVMT@Ti electrodes

The MPVMT@Zn electrode was prepared by spray coating. First, 54 mg VMT and 6 mg polyvinylidene fluoride (PVDF) were added to 20 mL of

N-methyl-2-pyrrolidine (NMP). Then, the mixture was ultrasonically dispersed for 1 h, followed by magnetically stirring at 500 r.p.m. for 1 h to obtain the slurry. Next, the slurry was coated on the polished and washed Zn foil (99.9%, Sike) by spraying method at 80 °C and dried at the same temperature for 12 h under vacuum to obtain the MPVMT@Zn electrode. The MPVMT@Ti electrode was prepared in almost the same way as the MPVMT@Zn electrode, except that the Zn foil was replaced with the Ti foil.

## Preparation of NVMT@Zn electrode

The preparation of NVMT@Zn electrode was similar to that of MPVMT@Zn electrode, except that the MPVMT layers were replaced with NVMT layers.

## Preparation of $MnO_2$ cathodes

The $MnO_2$ cathodes were synthesized by hydrothermal method[58]. First, 0.507 g manganese sulfate monohydrate ($MnSO_4 \cdot H_2O$, 98%) and 2 mL 0.5 mol·$L^{-1}$ sulfuric acid ($H_2SO_4$, A.R.) were added into 90 ml DI water and magnetically stirred until the mixed solution became transparent. Then, 20 mL 0.1 mol·$L^{-1}$ potassium permanganate ($KMnO_4$, A.R.) was dropwise added into the above solution. The mixture was stirred at room temperature for 2 h, followed by transferring to a Teflon-lined autoclave. Subsequently, the autoclave was kept at 120 °C for 12 h. After cooling to room temperature, the precipitated product was collected by vacuum filtration, washed repeatedly with DI water, and dried in a vacuum oven at 60 °C for 12 h to obtain $MnO_2$. Next, $MnO_2$ powder, acetylene black (AB) and PVDF were mixed at a weight ratio of 8:1:1 in NMP solvent to form a homogeneous slurry, which was then coated onto carbon cloth. After drying at 60 °C for 12 h, the $MnO_2$ cathode was cut into 12 mm circular disks with a mass loading of 1 mg $cm^{-2}$.

## Preparation of MPVMT@Zn||$MnO_2$ pouch cells

An ampere-hour MPVMT@Zn||$MnO_2$ pouch cells consisted of two $13 \times 15$ $cm^2$ $MnO_2$ cathodes (mass loading: 13.5 mg $cm^{-2}$) clamped in the middle sharing one Ti foil current collector and two $13 \times 15$ $cm^2$ Zn foils (50 µm) on both sides separated by $14 \times 16$ $cm^2$ glass fiber (GF) separators (GF/D from Whatman was used in this work unless otherwise specified). The shell of the pouch cell was an aluminum-plastic film and the electrolyte was 2 M zinc sulfate ($ZnSO_4$, A.R.) and 0.1 M $MnSO_4$.

## Preparation of $I_2$ cathodes

Activated carbon (AC, YP80F) was coated on the carbon cloth (mass loading: 20 mg $cm^{-2}$) as the host materials by mixing well with AB and PVDF in NMP solvent with weight ratios of AC:AB:PVDF of 8:1:1. After drying at 60 °C for 12 h, the cathode was cut into 12 mm circular disks with a mass loading of 20 mg $cm^{-2}$.

## Preparation of MPVMT@Zn||$I_2$ pouch cells

Typical MPVMT@Zn||$I_2$ pouch cells were constructed with $2 \times 3$ $cm^2$ Zn foils (80 µm), $2 \times 3$ $cm^2$ $I_2$ cathodes (mass loading of AC: 16 mg $cm^{-2}$), $3 \times 4$ $cm^2$ glass fiber separators and Ti foil current collectors on both sides. The shell of the pouch cell was an aluminum-plastic film and the electrolyte was 2 M $ZnSO_4$ and 0.5 M potassium iodide (KI, 99.5%).

## Electrochemical measurement

All the electrochemical performance was tested by assembling coin cells (CR2032), except for Zn||$MnO_2$ pouch cells, using glass fiber as the separator. Normally, the thickness of Zn foil is 80 µm, if not specifically noted. Cycling tests for Zn||Zn symmetric cells and Zn||Ti asymmetric cells of bare Zn or VMT@Zn were conducted with 150 µL 2 M $ZnSO_4$ as the electrolyte to study the Zn plating/stripping behavior, nucleation process and CE. The cutoff voltage of Zn||Ti cells was set to 0.5 V (vs. $Zn^{2+}$/Zn). For Zn||$MnO_2$ full cells, 2 M $ZnSO_4$ with 0.1 M $MnSO_4$ was used as the electrolyte and the voltage range was set to 1.0–1.8 V. For Zn||$I_2$ full cells, 2 M $ZnSO_4$ with 0.5 M KI were used as the anode and electrolyte, and the voltage range was set to 0.6–1.6 V. All galvanostatic charge-discharge measurements were evaluated using a NEWARE battery testing system at different current densities at 30 °C. The electrochemical impedance spectroscopy (EIS, from 100 kHz to 0.01 Hz), cyclic voltammetry (CV) and chronoamperometry (CA) were measured using an electrochemical work-station (BioLogic SP-150e). The electronic resistivity measurement, LSV and Tafel plot were performed by a CHI760E electrochemical workstation. Corrosion property was tested by Tafel plot from -1.2 V to -1.8 V with the scan speed of 5 mV $s^{-1}$. HER was tested by in situ gas chromatography (GC) with homemade bare Zn and MPVMT@Zn symmetric cells at 20 mA $cm^{-2}$/2 mAh $cm^{-2}$ for 90 min, and LSV from -1.2 V to -1.9 V with the scan speed of 2 mV $s^{-1}$.

The volumetric energy density of MPVMT@Zn||$MnO_2$ pouch cells are calculated as follows:

$$E = \frac{\int IUdt}{V} \tag{3}$$

where $E$, $I$, $U$, $dt$ and $V$ represent the volumetric energy density, discharging current, discharging voltage, time differential and total volume of the MPVMT@Zn||$MnO_2$ pouch cell.

## Materials characterization

Transmission electron microscopy (TEM, FEI Tecnai G2 F30) was used to observe the morphology and microstructure of MPVMT. Cold-field-emission spherical aberration corrected transmission electron microscope (STEM, Thermo Fisher Scientific, Spectra 300) operated at 200 kV. The number of atomic layers of samples and surface roughness were examined by atomic force microscope (AFM) and their surface potential was determined by KPFM, both of which were performed on Bruker Dimension Icon. The zeta potential of MPVMT suspension was analyzed by Malvern Zetasizer Nano S90. The surface morphology and composition were characterized by scanning electron microscopy (SEM, 5 kV, Hitachi SU8010) with a super energy-dispersive X-ray spectroscopy detector. One-dimensional and two-dimensional XRD was performed on Bruker D8 Advance with Cu Kα radiation at a scan rate of 10° $min^{-1}$ to characterize the phase composition and preferred orientation. The contact angles of MPVMT@Zn and bare Zn were measured on the Ossila contact angle system. In situ optical observation was conducted by an optical microscope (BX53M, OLYMPUS). A confocal laser scanning microscope (CLSM, VK-X1000, KEYENCE) was used to analyze the roughness of the surface after cycling at a large area. XPS measurements were conducted by a PHI 5000 Versa Probe II In-Situ XPS. In situ HER observation was performed by GC (ZhongJiaoJinYuan GC7920). All electrochemical tests were performed in the atmosphere at room temperature (-25 °C).

## Density functional theory (DFT) calculations

First-principles DFT calculations were carried out to reveal the growth mechanism of Zn metal on MPVMT. All of the calculations were performed using the projector-augmented wave method as implemented in the Vienna Ab initio Simulation Package (VASP 5.4.4)[17]. The energy cutoff for the plane-wave basis expansion was set to 500 eV. And the generalized gradient approximation with the Perdew-Burke-Ernzerhof exchange-correlation functional was used. The self-consistent electron density was determined using iterative diagonalization of the Kohn−Sham Hamiltonian, with the occupation of the Kohn−Sham states being smeared according to a Fermi−Dirac distribution with a smearing parameter of $k_{BT} = 0.1$ eV. For all of the calculations, the convergence criteria were set as $10^{-5}$ eV for electronic loops and 0.02 eV $Å^{-1}$ for ionic loops. The CIF data of bulk Zn and eight typical 2D phyllosilicate clay materials were obtained from Crystallography Open Database. Zn(100), Zn(101) and Zn(002) surfaces, in accordance with the three strong peaks of Zn metal, were employed to evaluate the adsorption strength of Zn atom on bare Zn plates using a slab model consisting of three atomic layers with the thickness of the vacuum set at 15 Å along the z-axis to avoid the interaction between periodic structures. Based on the average atomic fractions of MPVMT obtained from EDS maps (Si: Al ≈ 3:1, Mg: Fe ≈ 5:1, see Supplementary Table 3 for details), Zn(002)/MPVMT(002) was constructed as Fig. 3n. The composite model consists of two layers of Zn (002) atoms and a single layer

of MPVMT (002) with five atom layers. The Brillouin zone of the supercell was sampled using a $3 \times 4 \times 1$ uniform k-point mesh. All of the adsorption sites for Zn on MPVMT(002) were considered. Van der Waals interaction was taken into account at DFT-D3[59]. The adsorption energy was calculated as follows:

$$E_{ads} = E_{total} - E_{sub} - E_{Zn} \tag{4}$$

where $E_{total}$ is the total energy of Zn adsorbed system, $E_{sub}$ and $E_{Zn}$ are the energies of the pure substrate structure and the isolated Zn atom, respectively.

### Finite element simulation

Utilizing the COMSOL Multiphysics 6.0 software, the finite element analysis was conducted to scrutinize the electrochemical behavior of $Zn^{2+}$. The underlying simulation equations are as follows:

The diffusion and migration of $Zn^{2+}$ within the electrolyte domain adhere to the Nernst-Planck equation:

$$N_{Zn^{2+}} = - D_{Zn^{2+}} \left( \nabla c_0 - \frac{zFc_0}{RT} \nabla \Phi \right) \tag{5}$$

where $N_{Zn}^{2+}$ represents $Zn^{2+}$ flux, $D_{Zn}^{2+}$ denotes the diffusion coefficient, $z$ is the number of electron transferred, and $c_O$ is the concentration of $Zn^{2+}$. The constants $F$ and $R$ represent Faraday's constant and the ideal gas constant, respectively, while $T$ is the absolute temperature in Kelvin and $\Phi$ is the potential of the electrolyte.

In maintaining mass and charge conservation within the electrolyte, each species including $Zn^{2+}$ adheres to the respective equations:

$$\frac{\partial c_{Zn^{2+}}}{\partial t} + \nabla \times N_{Zn^{2+}} = 0 \tag{6}$$

$$\sum_i z_i c_i = 0 \tag{7}$$

where $c_i$ refers to the concentration and $z_i$ to the valence of each species.

The deposition of $Zn^{2+}$ at the electrode-electrolyte interface can be described by the following simplified representation:

$$Zn^{2+} + 2e^- \leftrightarrow Zn$$

The local deposition rate of $Zn^{2+}$ is quantified through the local current density which follows the Butler-Volmer equation:

$$i = i_0 \left[ \exp \left( \frac{\alpha_a F \eta}{RT} \right) - \frac{c_{Zn^{2+}}}{c_0} \exp \left( \frac{\alpha_c F \eta}{RT} \right) \right] \tag{8}$$

with $i_0$ being the exchange current density, $\eta$ representing the overpotential, $\alpha_a$ and $\alpha_c$ denoting the anodic and cathodic charge transfer coefficients, and $c_{Zn}^{2+}$ signifying the $Zn^{2+}$ concentration proximal to the anode surface.

Consequently, the boundary conditions adjacent to the substrate can be defined as:

$$N_{Zn^{2+}} \cdot \mathbf{n} = - \frac{i_0}{2F} \left[ \exp \left( \frac{\alpha_a F \eta}{RT} \right) - \frac{c_{Zn^{2+}}}{c_0} \exp \left( \frac{\alpha_c F \eta}{RT} \right) \right] \right] \tag{9}$$

where $\mathbf{n}$ signifies the normal vector of the boundary.

In the two-dimensional geometric simulations model for $Zn^{2+}$ deposition, to simulate the growth of dendrite, we set an inhomogeneity of the current density at the top of the electrode surface and assumed that the current density presents a Gaussian distribution:

$$i_{loc} = - i_0 * \exp \left( \frac{-x^2}{2\sigma^2} \right) \tag{10}$$

where $\sigma$ is the standard deviation, and $x$ is the lateral distance.

The two-dimensional and three-dimensional geometric simulations model and scales are in Supplementary Fig. 17 and Supplementary Fig. 19, respectively. Both the anodic and cathodic charge transfer coefficients are assigned as 0.5, the exchange current density is set as $100 \, \text{mA cm}^{-2}$, the applied current density is $1 \, \text{mA cm}^{-2}$ and $10 \, \text{mA cm}^{-2}$, and the temperature is 298 K. The diffusion coefficient of $Zn^{2+}$ in the electrolyte is defined as $1e^{-9} \, \text{m}^2 \, \text{s}^{-1}$. The equilibrium potential of Zn is set to 0 V and the equilibrium potential of MPVMT is set to $-300$ mV.

### Reporting summary

Further information on research design is available in the Nature Portfolio Reporting Summary linked to this article.

## Data availability

All data that support the findings of this study are presented in the Manuscript and Supplementary Information, or are available from the corresponding author upon reasonable request. Source data are provided with this paper.

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

## Acknowledgements

This work was supported by the National Key Research and Development Program of China (2019YFA0705703) [G.Z.], Joint Funds of the National Natural Science Foundation of China (U21A20174) [G.Z.], Guangdong Innovative and Entrepreneurial Research Team Program (2021ZT09L197) [G.Z.], Guangdong Basic and Applied Basic Research Foundation (Grant No. 2022A1515110117) [X.Z.], Shenzhen Science and Technology Program (KQTD20210811090112002) [G.Z.], Shenzhen Stabilization Support Program (WDZC20200824091903001) [G.Z.], Start-up Fund [G.Z.], and the Overseas Research Cooperation Fund of Tsinghua Shenzhen International Graduate School [G.Z.]. This work made use of the TEM facilities at the Institute of Materials Research, Tsinghua Shenzhen International Graduate School (Tsinghua SIGS). The first author would like to thank Jianyu Xie from Southern University of Science and Technology, Wei Zhang from University College London, Qingjin Fu, Zhexuan Liu and Jiarong Liu from Tsinghua University for their useful discussion or manuscript revision. The authors thank Dr. Yuyuan Jiang for performing the TEM measurement and helpful discussion, and thank Zhejiang Vastech Co. Ltd. for testing ampere-hour batteries. They also thank the Testing Technology Center of Materials and Devices of Tsinghua Shenzhen International Graduate School for its help with materials characterization.

## Author contributions

G.Z. conceived the project. Z.Z., L.D. and H.L. synthesized the materials. Z.Z. and X.Z. carried out the materials characterization and analyzed the data. Q.Z. and M.Z. conducted theoretical simulations. X.X., J.X., M.J., B.W., Y.J. and R.M. provided important experimental insights. G.Z. supervised the research and revised the manuscript. Z.Z., X.Z. and Q.Z. wrote the manuscript with comments and revisions from all the authors.

## Competing interests

The authors declare no competing interests.
