## [Peer Review File · Nature Communications]

An extended substrate screening strategy enabling a low lattice mismatch for highly reversible zinc anodesReviewers' comments:

Reviewer #1 (Remarks to the Author):

The article "Growing Zn(002) seeds in highly-compatible monolayer porous phyllosilicate to realize high-rate and stable Zn-metal batteries" is a work that deserves to be published with minimal changes. In this paper, the authors have designed a vermiculite-based porous material that they support on a Zn plate. Thus, they are creating a porous film that allows the Zn²⁺ to be deposited with Zn(002) geometry, which is the most favorable for horizontal growth, reducing the formation of dendrites. In addition, this film avoids secondary reactions, such as the evolution of H₂, very common during recharging of Zn batteries.

In this work, a large number of experimental techniques are used and an exhaustive theoretical treatment, using COMSOL and DFT, was carried out.

The results obtained for cyclability and capacity in this work are very promising and improve those previously obtained.

The text is very well structured, the figures are very well chosen and the discussion is coherent and very complete.

The only point that I think the authors should clarify is what mass they mean when they give Specific Capacity values (mAh/g). Is it mass of MnO₂, MPVMT, MPVMT@Zn, ...?

In conclusion, I think this article can be published in Nature Communications with minimal changes.

Reviewer #2 (Remarks to the Author):

The manuscript "Growing Zn (002) seeds in highly compatible monolayer porous phyllosilicate to realize high-rate and stable Zn-metal batteries" proposes the use of a monolayer porous vermiculite (MPVMT) as a coating layer for the Zn anode in Zn-metal batteries. The coating layer promotes planar Zn deposition from the Zn (002) face, inhibits side reactions, and facilitates ion diffusion on the anode surface. Symmetric cells incorporating MPVMT@Zn exhibit stability, operating for over 300 hours at a high current density of 50 mA cm⁻². However, the manuscript lacks novelty as vermiculite has already been studied extensively for electrode protection. Additionally, further investigation is required to confirm the monolayer coating and chemical stability of vermiculite during the battery process. Therefore, we recommend the manuscript should be rejected. See attached comments for further details.

1. The Introduction section of the manuscript provides a comprehensive overview of previously investigated coating materials, both organic and inorganic, for protecting the Zn anode. However, in light of this existing body of research, it is important to clearly articulate the novelty and significance of the present work. Additionally, it should be noted that vermiculite has been extensively studied as a protective layer for electrodes, as demonstrated by previous publications (Angew. Chem. Int. Ed. 2019, 58, 6200; Adv. Funct. Mater. 2019, 29, 1900648). Therefore, it is essential to demonstrate how the proposed approach differs from previous research and contributes to the field.

2. Fig. 5c and 5h depict the current density at a range of 0.6-1.2 mA cm⁻² for MnO₂|MPVMT@Zn full cells and pouch cells, respectively. While this provides valuable information, it is important to investigate the performance of the MPVMT@Zn symmetric cell at lower current densities, specifically below 1 mA cm⁻², to obtain a comprehensive understanding of its Zinc deposition/stripping behavior.

3. Fig. 1b shows porous vermiculite nanosheets with a thickness of approximately 2 nm in the exfoliated liquid, while Supplementary Fig. 9 depicts a VMT coating layer with a thickness of approximately 4.3 μm. However, it is crucial to determine if there is any force between the VMT nanosheets in the coating layer. Additionally, it is challenging to definitively conclude that the deposited VMT on the Zn anode is strictly a monolayer. Further investigation is required to determine if the coated VMT layer is multilayered.

4. Supplementary Fig. 8 marks peaks below 10 degrees as VMT, but no other peaks from CMT are observed. While this provides some evidence for the presence of VMT on the surface of the Zn anode,

further characterization techniques and analysis are necessary to confirm its presence conclusively. 5. While the proposed approach shows promise for protecting the Zn anode, further evidence is needed to determine the chemical stability of vermiculite during the aqueous zinc ion battery process. This will be essential for evaluating the long-term efficacy of the proposed approach.

Reviewer #3 (Remarks to the Author):

This work presents a novel zinc anode protective coating based on a monolayer porous vermiculite (MPVMT) with the lowest lattice mismatch of Zn (002) plane. The MPVMT protective coating possesses natural negative charge insulation properties and has demonstrated effective inhibition of unwanted side reactions. Additionally, it has been designed with vertical transmission channels to enhance ion transportation.

The coating strategy reported in this work is a common approach (Nat Commun 2021, 12 (1), 6606.). The proposed strategy of inducing zinc ion deposition on the (002) crystal plane by designing a protective layer with the lowest lattice mismatch has been previously reported (Adv. Mater. 2021, 33 (21), e2100187., ACS Energy Lett. 2021, 7 (1), 197.), indicating that the work lacks innovation. Moreover, while insulating coatings can effectively suppress side reactions, it does not provide a novel solution to solve the important issue of hindering ion transport. The vertical transport channel mentioned has been already demonstrated (Adv. Funct. Mater. 2020, 30 (21).), and this work does not provide a robust characterization of the formation mechanism and ion migration behavior of this vertical channel. Additionally, the electrochemical performance of the MPVMT@Zn symmetric cell and full cell is not outstanding. The low testing capacities of 50 mA cm⁻²-1 mAh cm⁻² and 10 mA cm⁻²-1 mAh cm⁻² do not demonstrate the ability of accommodating high-capacity zinc deposition, more high zinc deposition capacity data need to be provided. Based on these reasons, I do not recommend publishing this manuscript in Nature Communication.

Nonetheless, further investigation is also required to address the following issues.

- 1) While insulating protective coatings may hinder side reactions, they simultaneously reduce the transfer rate of ions and increase internal resistance. Regarding this issue, the vertical transport channels mentioned in the article have not been fully characterized and proven. Theoretical calculations require support from relevant experimental data.
- 2) The assertion that natural clay minerals mentioned in the article are more easily cleaved into 2D nanosheets than other materials, is doubtful.
- 3) How can the strong hydrogen bonding between the surface silanol groups and oxygen functional groups of water molecules in the coating be demonstrated? Please provide more favorable characterizations to fully demonstrate.
- 4) Explain why the XRD of original pure zinc and MPVMT@Zn, as shown in Supplementary Fig. 8, does not exhibit the advantage of MPVMT@Zn's prominent (002) crystal plane?
- 5) The section on characterizing the coating's rapid ion transport capabilities and fast solvation mechanism does not provide sufficient experimental data, and it is not sufficient to rely on simulations alone for interpretation.

Point-by-point response to the reviewers' comments

Dear Reviewers,

Thank you very much for reviewing our manuscript. We greatly appreciate your professional comments and suggestions, which have helped us improve the scientific quality of this work. According to your valuable comments and suggestions, we have carefully revised the manuscript and supplementary information (SI). All details of the revision and our responses to the comments can be found in the following text. We marked the revised parts of the manuscript and SI in red so that the corresponding changes can be clearly seen.

Sincerely yours,

Guangmin Zhou

~~~~~

Guangmin Zhou, Ph.D, Associate Professor;

Tsinghua-Berkeley Shenzhen Institute (TBSI) & Tsinghua Shenzhen International Graduate School, Tsinghua University;

Low-Dimensional Materials and Devices Laboratory, Shenzhen Geim Graphene Center;

Associate Editor/Scientific Managing Editor, Energy Storage Materials (EnSM);

E-mail: [guangminzhou@sz.tsinghua.edu.cn](mailto:guangminzhou@sz.tsinghua.edu.cn)

**Reviewer #1 (Revision marked with red in the revised manuscript and SI)**

**General comments:** The article "Growing Zn(002) seeds in highly-compatible monolayer porous phyllosilicate to realize high-rate and stable Zn-metal batteries" is a work that deserves to be published with minimal changes. In this paper, the authors have designed a vermiculite-based porous material that they support on a Zn plate. Thus, they creating a porous film that allows the  $Zn^{2+}$  to be deposited with Zn(002) geometry, which is the most favorable for horizontal growth, reducing the formation of dendrites. In addition, this film avoids secondary reactions, such as the evolution of  $H_2$ , very common during recharging of Zn batteries. In this work, a large number of experimental techniques are used and an exhaustive theoretical treatment, using COMSOL and DFT, was carried out. The results obtained for cyclability and capacity in this work are very promising and improve those previously obtained. The text is very well structured, the figures are very well chosen and the discussion is coherent and very complete. The only point that I think the authors should clarify is what mass they mean when they give Specific Capacity values ( $mAh\ g^{-1}$ ). Is it mass of  $MnO_2$ , MPVMT, MPVMT@Zn, ...? In conclusion, I think this article can be published in Nature Communications with minimal changes.

**Response:** We are so grateful for the reviewer's encouraging comments and strong recommendation of the acceptance of our work by *Nature Communications*. Here, from a series of calculated phyllosilicates matching Zn(002), we select vermiculite, which has the lowest lattice mismatch ( $\delta=0.38\%$ ) as the model to study the regulatory mechanism of Zn deposition on " $2d_{Zn(002)}$ " substrates and its comprehensive effect on stabilizing Zn anodes protection. Then, we develop a novel monolayer porous vermiculite (MPVMT) through a large-scale and green preparation as a functional coating for Zn electrodes. Unique "planting Zn(002) seeds" mechanism for " $2d_{Zn(002)}$ " substrates is revealed to induce the oriented growth of Zn deposits. Additionally, MPVMT coatings effectively inhibit side reactions and promote  $Zn^{2+}$  transport. Consequently, the MPVMT@Zn symmetric cells exhibit stable cycling for over 800 h at a high current density of  $10\ mA\ cm^{-2}$  and 300 h at an ultrahigh current density of  $50\ mA\ cm^{-2}$ . Furthermore, the assembled MPVMT@Zn|| $MnO_2$  full cells show an improvement rate of capacity retention of 217% after 500 cycles. This work extends traditional coating screening strategy and advances the understanding of Zn nucleation

mechanism, paving the way for realizing high-rate and stable Zn-metal batteries and probably applying them to other metal-based batteries.

And we are sorry that we haven't clarified the mass in the Specific Capacity values ( $\text{mAh g}^{-1}$ ) clearly before. The mass in the Specific Capacity values ( $\text{mAh g}^{-1}$ ) we give is based on the mass of  $\text{MnO}_2$ .

**We have revised our manuscript accordingly.**

**Revised manuscript, Page 23, a sentence has been added:**

“Next, the rate performance at different current densities was investigated and the capacities of both batteries are almost the same at around  $250 \text{ mAh g}^{-1}$  at 1 C (1 C =  $308 \text{ mA g}^{-1}$ , **based on the mass of  $\text{MnO}_2$** ) (**Fig. 6b**).”

## **Reviewer #2 (Revision marked with red in the revised manuscript and SI)**

**General comments:** The manuscript "Growing Zn (002) seeds in highly compatible monolayer porous phyllosilicate to realize high-rate and stable Zn-metal batteries" proposes the use of a monolayer porous vermiculite (MPVMT) as a coating layer for the Zn anode in Zn-metal batteries. The coating layer promotes planar Zn deposition from the Zn (002) face, inhibits side reactions, and facilitates ion diffusion on the anode surface. Symmetric cells incorporating MPVMT@Zn exhibit stability, operating for over 300 hours at a high current density of  $50 \text{ mA cm}^{-2}$ . However, the manuscript lacks novelty as vermiculite has already been studied extensively for electrode protection. Additionally, further investigation is required to confirm the monolayer coating and chemical stability of vermiculite during the battery process. Therefore, we recommend the manuscript should be rejected. See attached comments for further details.

**Response:** Thanks for your valuable comments on our manuscript, which greatly help us improve the depth and integrity of our work.

1. **Novelty:** Considering your concerns and suggestions, we have revised our manuscript to highlight the novelty of our work, revision including Abstract, Introduction, Discussion and Fig. 1, etc. The novelty of this work can be summarized in the following three points. i) our work proposes an extended substrate screening strategy (ESSS) based on the traditional substrate/coating screening criterion (TSSC), namely  $d_{\text{substrate}}: d_{\text{Zn}(002)}=1:1 \rightarrow d_{\text{substrate}}: d_{\text{Zn}(002)}=n:1$  ( $n=1, 2$ ), and discovers the " $2d_{\text{Zn}(002)}$ " vermiculite (VMT) substrate with the record-low lattice mismatch with Zn(002) and unique "planting Zn(002) seeds" mechanism to realize dendrite-free Zn anode. ii) Our MPVMT coatings not only inhibit side reactions through dual effects of surface functional groups and charge repulsion, but also promote  $\text{Zn}^{2+}$  transport by designed acceleration channels. iii) We first prepare monolayer porous vermiculites by a simple, large-scale and sustainable method. Please see details in Response to Comment 1.
2. **Monolayer coating confirmation:** we have added characterizations to investigate the existence form of the coating formed by stacking monolayer nanosheet. Please see details in Response to Comment 3.
3. **Chemical stability of vermiculite:** we have added characterizations to verify the chemical stability of MPVMT coatings during cycling. Please see details in Response to Comment 5.

In conclusion, as you suggested, more details are provided in the revised manuscript and SI. Based on all these efforts, we believe the quality of this manuscript has been

improved greatly. We hope that you can reconsider the publication of the present manuscript in *Nature Communications*.

**Comment 1.** The Introduction section of the manuscript provides a comprehensive overview of previously investigated coating materials, both organic and inorganic, for protecting the Zn anode. However, in light of this existing body of research, it is important to clearly articulate the novelty and significance of the present work. Additionally, it should be noted that vermiculite has been extensively studied as a protective layer for electrodes, as demonstrated by previous publications (*Angew. Chem. Int. Ed.* 2019, 58, 6200; *Adv. Funct. Mater.* 2019, 29, 1900648). Therefore, it is essential to demonstrate how the proposed approach differs from previous research and contributes to the field.

**Response:** We acknowledge the reviewer's thoughtful comments and are sorry that the novelty and significance of our work may not be clearly shown in the previous manuscript. Now we have reorganized Abstract, Introduction, Discussion and Fig.1&2 to highlight the novelty and significance, and elucidated the difference between our work and vermiculite-based previous work.

1. Novelty and significance of the present work

First, our work proposes an ESSS based on the TSSC, namely  $d_{\text{substrate}}: d_{\text{Zn}(002)}=1:1 \rightarrow d_{\text{substrate}}: d_{\text{Zn}(002)}=n:1$  ( $n=1, 2$ ), and discovers the substrate material (VMT) with the lowest lattice mismatch ( $\delta=0.38\%$ ) with Zn(002) reported so far, together with unique “planting Zn(002) seeds” mechanism for “ $2d_{\text{Zn}(002)}$ ” substrates. It is generally believed that the preferred orientation of Zn(002) has a vital effect on the cycling durability of Zn anodes, because of the relatively smooth surface and lower chemical activity of Zn(002) plane (*Science* 2019, 366, 645-648; *Adv. Mater.* 2021, 33, e2100187; *Energy Environ. Sci.* 2022, 15, 5017-5038). Essentially, the anisotropy of the substrates significantly influences the orientation of Zn electrodeposits (*Adv. Mater.* 2023, 35, e2211961; *Adv. Mater.* 2022, 34, e2202552). As shown in **Response Fig. 1**, due to the limit of TSSC, there is an urgent need to explore new substrate materials that not only possess low lattice mismatch with Zn(002), but also inhibit side reactions and promote ion transport. Here, as shown in **Response Fig. 2**, we conjecture and verify the feasibility of selecting suitable coatings for Zn anodes based on the ESSS. From a series of calculated phyllosilicates satisfying  $d_{\text{substrate}} \approx 2d_{\text{Zn}(002)}$ , we select VMT, which has the lowest lattice mismatch ( $\delta=0.38\%$ ) as the model to confirm the effectiveness of “ $2d_{\text{Zn}(002)}$ ” substrates for Zn anodes protection. As a result, we find that VMT layers allow to “plant Zn(002) seeds” in their planes, thus guiding subsequent horizontal

growth of Zn and achieving dendrite-free Zn electrodes. This work extends SSS and advances the understanding of Zn nucleation mechanism, paving the way for realizing high-rate and stable Zn-metal batteries and probably applying them to other metal-based batteries.

**Response Fig. 1 a** Schematic of challenges existing in Zn anodes. **b** Different lattice strain and orientation correlation of Zn electrodeposition formed on substrates with incoherent ( $\delta > 25\%$ ), semi-coherent ( $25\% \geq \delta \geq 5\%$ ) and coherent ( $\delta < 5\%$ ) interfaces and TSSC used in previous work.

**Response Fig. 2 a** Schematic of our work design of MPVMT layers based on ESSS to stabilize Zn anodes. **b** Comparison of lattice mismatch among eight typical phyllosilicates screened based on  $d_{\text{substrate}} : d_{\text{Zn}(002)}=2:1$ . **c** Comparison of lattice mismatch between recently reported substrates and our work.

Second, the proposed MPVMT coatings not only inhibit side reactions through dual effects of surface functional groups and charge repulsion, but also promote  $\text{Zn}^{2+}$  transport by designed acceleration channels. Based on TSSC, only a few Zn-based alloy/metal substrates enable to form coherent interfaces with Zn(002) (*Nat. Commun.* 2021, 12, 237; *Adv. Mater.* 2022, 34, e2202552). Unfortunately, these conductive substrates cannot isolate Zn electrodes from aqueous electrolytes, resulting in unavoidable side reactions (*Energy Environ. Sci.* 2022, 15, 1086-1096; *Angew. Chem. Int. Ed.* 2023, 62, e202212695). In contrast, based on ESSS, we find insulating vermiculite coatings not only form coherent interfaces but also effectively inhibit side reactions (**Response Fig. 3**). Meanwhile, insulating coatings usually face the problem of slow ion transport. Here, to address the issue, we design a monolayer porous vermiculite with acceleration channels for  $\text{Zn}^{2+}$ , the ion transport behavior visualized by COMSOL simulations (**Response Fig. 4**), resulting in superior cyclability even at ultrahigh current densities of  $50 \text{ mA cm}^{-2}$ . Therefore, our design develops a promising solution to the contradiction between side reactions and ion transport, providing a new strategy for high-rate and long-cycle Zn-metal batteries.

**Response Fig. 3 a** In situ electrochemical GC profile for bare Zn (left) and MPVMT@Zn (right) symmetric cells at  $20 \text{ mA cm}^{-2}$  and  $2 \text{ mAh cm}^{-2}$ . **b** XRD patterns of bare Zn and MPVMT@Zn after immersing in 2 M  $\text{ZnSO}_4$  electrolyte for 1 and 6 days.

**Response Fig. 4** Electric field distribution (a) and Zn ion flux (b) of bare Zn and MPVMT@Zn at  $10 \text{ mA cm}^{-2}$ .

Third, we first prepare monolayer porous vermiculites by a simple, large-scale and sustainable method. At present, the only reported method for preparing MPVMT materials involves obtaining exfoliated vermiculite nanosheets by an ion exchange method, and then preparing porous vermiculite nanosheets by chemical etching (*J. Mater. Chem. A* 2021, 9, 14576-14581), resulting in a complex and unsustainable preparation process. To overcome these issues, we develop a one-step ultrasonic and agitating method using only deionized water as the solvent to easily produce large quantities of MPVMT materials (**Response Fig. 5**). Hence, our simple and sustainable preparation makes it a viable option to fulfill practical demands.

**Response Fig. 5** Schematic of simple, sustainable and scalable preparation of MPVMTs.

## 2. Differences between previous researches and our work

First, these two works (*Angew. Chem. Int. Ed.*2019,58,6200; *Adv. Funct. Mater.* 2019,29,1900648) mainly took advantage of the negative charge and high mechanical strength of vermiculites. One (*Angew. Chem. Int. Ed.*2019,58,6200) used it as an electrolyte additive to realize a dendrites-free Li anode through co-deposition; The

other (*Adv. Funct. Mater.* 2019,29,1900648) used it as a filler to improve the ionic conductivity and mechanical properties of solid polymer electrolytes in Li-ion batteries. Additionally, both works used monolayer but non-porous vermiculites, while porous structures in vermiculites as a protective layer are essential for ion transport. Here, we develop a large-scale and sustainable method to create porous structures in the atomic layer structure of vermiculites (MPVMT), which has never been reported, and use MPVMT as a coating layer to realize dendrite-free, inhibited side reactions and fast ion transport of Zn anodes.

Essentially, Li-metal batteries in organic electrolytes and Zn-metal batteries in aqueous electrolytes are completely different, so the requirements for protective materials of anodes are different. In our work, we find that the lattice mismatch between vermiculite and Zn(002) is the lowest reported so far and unique “planting Zn(002) seeds” mechanism on MPVMT substrates. Additionally, the non-conductive MPVMT with hydrophilic groups on the surface effectively mitigates HER and the formation of byproducts, the problems particularly existing in aqueous Zn batteries. The above results indicate the uniqueness of vermiculite as Zn anode protection. Therefore, our study on MPVMT coating for Zn anodes significantly differs from the previous works.

**We have revised our manuscript accordingly.**

**Revised manuscript, Page 1, the title has been modified:**

“An extended substrate screening strategy enabling a record-low lattice mismatch for highly reversible Zn anodes”

**Revised manuscript, Page 2, the abstract has been modified:**

“Aqueous zinc (Zn) batteries possess intrinsic safety and cost-effectiveness, but dendrite growth and side reactions of Zn anodes hinder their practical application. Here, we propose the extended substrate screening strategy (SSS) for stabilizing Zn anodes and verify its availability ( $d_{\text{substrate}}: d_{\text{Zn}(002)}=1:1 \rightarrow d_{\text{substrate}}: d_{\text{Zn}(002)}=n:1, n=1, 2$ ). From a series of calculated phyllosilicates satisfying  $d_{\text{substrate}} \approx 2d_{\text{Zn}(002)}$ , we select vermiculite, which has the lowest lattice mismatch ( $\delta=0.38\%$ ) reported so far, as the model to confirm the effectiveness of “ $2d_{\text{Zn}(002)}$ ” substrates for Zn anodes protection. Then, we develop a monolayer porous vermiculite (MPVMT) through a large-scale and green preparation as a functional coating for Zn electrodes. Unique “planting Zn(002) seeds” mechanism for “ $2d_{\text{Zn}(002)}$ ” substrates is revealed to induce the oriented growth of Zn deposits. Additionally, MPVMT coatings effectively inhibit side reactions and promote  $\text{Zn}^{2+}$  transport. Consequently, MPVMT@Zn symmetric cells operate stably for over

300 h at an ultrahigh current density of  $50 \text{ mA cm}^{-2}$ . This work extends traditional SSS and advances the understanding of Zn nucleation mechanism, paving the way for realizing high-rate and stable Zn-metal batteries.”

**Revised manuscript, Pages 3 and 4, the 2nd and 3rd paragraphs of the introduction have been modified:**

“Essentially, the anisotropy of the substrates significantly influences the orientation of Zn electrodeposits. According to the lattice mismatch ( $\delta$ ) between deposited Zn and substrates, the contact interface can be divided into incoherent ( $\delta > 25\%$ ), semi-coherent ( $25\% \geq \delta \geq 5\%$ ) and coherent interfaces ( $\delta < 5\%$ ). The lower lattice mismatch, the smaller lattice strain of deposits on substrates, the stronger orientation correlation. Hence, it is necessary to search for a suitable substrate with a low lattice mismatch with Zn(002) plane. At present, based on the traditional substrate screening criterion (TSSC) that lattice parameters of substrates should be close to that of Zn(002) (i.e.  $d_{\text{substrate}}: d_{\text{Zn}(002)}=1:1$ ) (**Fig. 1b**), several substrates have been reported including two-dimensional (2D) materials (e.g., graphene,  $\text{MoS}_2$ ) and highly oriented metals (e.g., Sn(200), In(002)). However, most of them form semi-coherent interfaces with Zn(002). Only a few Zn-based alloys or single-crystal Zn(002) anodes enable to form coherent interface. Unfortunately, these alloy or metal substrates are electrically conductive, and thus they cannot isolate Zn electrodes from aqueous electrolytes, resulting in unavoidable side reactions. Hence, there is an urgent need to explore possible extended substrate screening strategy (ESSS) to guide the search for new substrate materials that not only possess low lattice mismatch with Zn(002), but also inhibit side reactions and promote ion transport, especially at high current densities.

Here, we conjecture and verify the application potential of selecting suitable coatings for Zn anodes based on ESSS, namely  $d_{\text{substrate}}: d_{\text{Zn}(002)}=1:1 \rightarrow d_{\text{substrate}}: d_{\text{Zn}(002)}=n:1, n=1, 2$ . First, through calculation, we discover eight typical phyllosilicates that meet “ $d_{\text{substrate}} \approx 2d_{\text{Zn}(002)}$ ”. Among them, vermiculite (VMT) with the lowest lattice mismatch ( $\delta=0.38\%$ ) reported so far, is selected as our model clay system to study the regulatory mechanism of Zn deposition on “ $2d_{\text{Zn}(002)}$ ” substrates and its comprehensive effect on stabilizing Zn anodes protection (**Figs. 1c and 1d**). Next, we synthesize monolayer porous vermiculites (MPVMTs) as coatings for Zn anodes (MPVMT@Zn). Notably, the preparation of MPVMT is simple and sustainable, and thus easy to achieve large-scale preparation. Owing to the ultralow lattice misfit and **double large lattice of MPVMT than Zn**, together with strong interaction with deposited Zn atom, **MPVMT layers allow to plant Zn(002) seeds** in their planes, thus guiding subsequent horizontal growth of Zn and achieving dendrite-free Zn electrodes. **Additionally, the non-conductive MPVMT with hydrophilic groups on the surface** effectively mitigates HER

and the formation of byproducts. Moreover, the monolayer porous structure with negative charges provides acceleration transport channels for the migration of  $Zn^{2+}$ , leading to superior cyclability at high current densities. As a result, the MPVMT@Zn symmetric cells exhibit stable cycling for over 800 h at a high current density of  $10 \text{ mA cm}^{-2}$  and 300 h at an ultrahigh current density of  $50 \text{ mA cm}^{-2}$ . Furthermore, the assembled MPVMT@Zn|| $MnO_2$  full cells show a stable lifespan of 500 cycles.”

**Revised manuscript, Page 5, a figure has been added:**

**Fig. 1 Schematic demonstrating the origin of the proposed ESSS and its verification based on MPVMT coatings. a** Schematic of challenges existing in Zn

anodes. **b** Different lattice strain and orientation correlation of Zn electrodeposition formed on substrates with incoherent ( $\delta > 25\%$ ), semi-coherent ( $25\% \geq \delta \geq 5\%$ ) and coherent ( $\delta < 5\%$ ) interfaces and TSSC used in previous work. **c** Schematic of our work design of MPVMT layers based on ESSS to stabilize Zn anodes. **d** Comparison of lattice mismatch among eight typical phyllosilicates screened based on  $d_{\text{substrate}}: d_{\text{Zn}(002)}=2:1$ . **e** Comparison of lattice mismatch between recently reported substrates and our work.

**Revised manuscript, Page 6, several sentences have been added:**

“According to the calculation results of adsorption energies of Zn atom on different Zn crystal planes (**Supplementary Fig. 1, Supplementary Note 1**), it is necessary to search for a substrate highly matched with Zn(002) to induce the planar deposition of Zn and inhibit dendrite growth. **Based on the conjecture of ESSS, eight typical phyllosilicate clay materials satisfying “ $d_{\text{substrate}}: d_{\text{Zn}(002)}=2:1$ ” were found by calculation.**”

“As shown in **Fig. 2a**, each layer of VMT consists of one Mg-based octahedral sheet sandwiched between two tetrahedral silicate sheets. **The cross-sectional scanning transmission electron microscopy (STEM) image of VMT clearly shows that tetrahedral silicate sheets have hollow sites between the atoms.**”

**Revised manuscript, Page 8, the Fig.2 has been modified:**

**Fig. 2 Preparation and characterization of MPVMT@Zn anode. a** Cross-sectional schematic of a monolayer vermiculite and cross-sectional STEM image of an unexfoliated vermiculite using the high-angle annular dark-field mode. **b** Schematic of the mass production of MPVMT materials through a sustainable method. **c** Atomic force microscopy (AFM) image of MPVMT nanosheets. The inset is the height profile of the corresponding lines. **d** Statistics for the thickness of MPVMT nanosheets. **e** Typical TEM image and SAED pattern (inset) of MPVMT. **f, g** Contact angles of bare Zn (**f**) and MPVMT@Zn (**g**). **h, i** KPFM images of MPVMT@Zn (**h**) and bare Zn (**i**). **j** Voltage response to a current of 2 mA of MPVMT@Zn and bare Zn electrodes. **k** The prepared MPVMT@Zn with an area of 300 cm2 fabricated using a scalable strategy.

Revised manuscript, Page 11, several sentences have been added:

“It is concluded that the strong adsorption of MPVMT layers to Zn atoms and the super match allow for “planting Zn(002) seeds” on the “ $2d_{\text{Zn}(002)}$ ” substrate, thereby achieving the subsequent preferred deposition and alleviating dendrite growth even at high current densities. The above evidence confirms the validity of ESSS for screening substrates that can induce the orientation of Zn electrodeposits. Based on ESSS, we have modified the formula for calculating the lattice mismatch  $\delta$  as follows:

$$\delta = \begin{cases} \left| \frac{d_{\text{substrate}} - d_{\text{Zn}(002)}}{d_{\text{Zn}(002)}} \right|, & d_{\text{substrate}} \leq 1.5 d_{\text{Zn}(002)} \\ \left| \frac{d_{\text{substrate}} - 2 d_{\text{Zn}(002)}}{d_{\text{Zn}(002)}} \right|, & 1.5 d_{\text{Zn}(002)} < d_{\text{substrate}} < 2.5 d_{\text{Zn}(002)} \end{cases}$$

where  $d$  is the lattice parameter of the substrate or Zn(002) plane.”

**Revised manuscript, Page 26, the discussion has been added:**

“In summary, we develop and confirm the applicability of ESSS ( $d_{\text{substrate}}: d_{\text{Zn}(002)}=1:1 \rightarrow d_{\text{substrate}}: d_{\text{Zn}(002)}=n:1, n=1, 2$ ). Taking the vermiculite ( $d_{\text{vermiculite}} \approx 2d_{\text{Zn}(002)}$ ) as a model, we prepare 2D hydrophilic and insulating MPVMTs by a simple and eco-friendly method as protective coatings for Zn anode. Due to MPVMT’s ultralow lattice mismatch ( $\delta=0.38\%$ ), “ $2d_{\text{Zn}(002)}$ ” structure cell and strong adsorption to Zn deposits, the coatings induce (002) oriented growth of Zn by “planting Zn(002) seeds” mechanism, realizing dendrite-free anodes. Additionally, the hydrophilic functional groups and negative charges on the surface of MPVMTs greatly promote the de-solvation of  $\text{Zn}^{2+}$  and homogenize the electric field, synergistically suppressing side reactions. Furthermore, the constructed ion acceleration channels significantly facilitate  $\text{Zn}^{2+}$  transport, achieving high-rate and long-life MPVMT@Zn anodes. Consequently, the MPVMT@Zn anodes deliver a superhigh cumulative capacity of  $7500 \text{ mAh cm}^{-2}$  at a current density of  $50 \text{ mA cm}^{-2}$ , outperforming most reported modified Zn anodes. The proposed ESSS and new nucleation mechanism provide new insights into highly reversible Zn metal batteries and may apply to other metal-based batteries.”

**Revised manuscript, Page 28, two references have been added:**

39. Ma, Q. *et al.* Bio-Inspired Stable Lithium-Metal Anodes by Co-depositing Lithium with a 2D Vermiculite Shuttle. *Angew. Chem. Int. Ed.* **58**, 6200-6206 (2019).

41. Tang, W. *et al.* High-Performance Solid Polymer Electrolytes Filled with Vertically Aligned 2D Materials. *Adv. Funct. Mater.* **29**, 1900648 (2019).

**Comment 2.** Fig. 5c and 5h depict the current density at a range of 0.6-1.2 mA cm-2 for MnO2|MPVMT@Zn full cells and pouch cells, respectively. While this provides valuable information, it is important to investigate the performance of the MPVMT@Zn symmetric cell at lower current densities, specifically below 1 mA cm-2, to obtain a comprehensive understanding of its Zinc deposition/stripping behavior.

**Response:** Thanks for your valuable suggestions. First, we have exhibited the rate performance of MPVMT@Zn and bare Zn symmetric cells at different current densities from 0.5 to 10 mA cm-2 in the initial manuscript. As shown in **Response Fig. 6**, the overpotential of MPVMT@Zn symmetric cell is 35 mV at 0.5 mA cm-2, much smaller than that of bare Zn symmetric cell (58 mV), demonstrating the superior fast kinetics of Zn plating/stripping. Additionally, we have added the cycle performance of MPVMT@Zn and bare Zn symmetric cells at 0.6 mA cm-2/10 mAh cm-2. As shown in **Response Fig. 7**, MPVMT@Zn symmetric cell operates stably for over 500 h, almost four times than bare Zn symmetric cell (132 h), indicating the superiority of MPVMT@Zn at low current densities.

**Response Fig. 6** Rate performance of the bare Zn and MPVMT@Zn electrodes at different current densities from 0.5 to 10 mA cm-2 with the same capacity of 1 mAh cm-2 (a) and corresponding voltage hysteresis (b)

**Response Fig. 7** Cycling performance of bare Zn and MPVMT@Zn symmetric cells at  $0.6 \text{ mA cm}^{-2}/10 \text{ mAh cm}^{-2}$

**We have revised our manuscript accordingly.**

**Revised SI, Page 32, a figure has been added:**

**Supplementary Fig. 24** Cycling performance of bare Zn and MPVMT@Zn symmetric cells at  $2 \text{ mA cm}^{-2}/2 \text{ mAh cm}^{-2}$  (a),  $10 \text{ mA cm}^{-2}/1 \text{ mAh cm}^{-2}$  (b),  $0.6 \text{ mA cm}^{-2}/10 \text{ mAh cm}^{-2}$  (c) and  $10 \text{ mA cm}^{-2}/10 \text{ mAh cm}^{-2}$  (d).

**Revised manuscript, Page 20, a sentence has been added:**

“The MPVMT@Zn electrode also exhibits excellent cycling stability at increasing current capacity of  $2 \text{ mA cm}^{-2}/2 \text{ mAh cm}^{-2}$  for 1500 h,  $5 \text{ mA cm}^{-2}/5 \text{ mAh cm}^{-2}$  for almost 700 h (Fig. 5f),  $0.6 \text{ mA cm}^{-2}/10 \text{ mAh cm}^{-2}$  for 500 h and  $10 \text{ mA cm}^{-2}/10 \text{ mAh cm}^{-2}$  for 400 h (Supplementary Fig. 25), much better than the bare Zn.”

**Comment 3.** Fig. 1b shows porous vermiculite nanosheets with a thickness of approximately 2nm in the exfoliated liquid, while Supplementary Fig. 9 depicts a VMT coating layer with a thickness of approximately 4.3  $\mu\text{m}$ . However, it is crucial to determine if there is any force between the VMT nanosheets in the coating layer. Additionally, it is challenging to definitively conclude that the deposited VMT on the Zn anode is strictly a monolayer. Further investigation is required to determine if the coated VMT layer is multilayered.

**Response:** We sincerely appreciate the reviewer's thoughtful comments.

First, negatively charged sheets of unexfoliated VMTs are held together by the electrostatic forces between alternate layers of bridging cations (*Science* 1967, 156, 385-387; *Adv. Colloid Interface Sci.* 1999, 82, 43-92). After the bulk VMTs are exfoliated into monolayer nanosheets, the VMTs lost the cation between the layers (*Desalination* 2007, 215, 133-142; *J. Mater. Sci.* 2019, 54, 5528-5535). Due to the negative repulsion of two nanosheets, it may be difficult for the exfoliated monolayers to spontaneously reassemble into the original tight multilayer structure.

Additionally, we have further investigated the interaction between two monolayers by density functional theory (DFT) calculations. As shown in **Response Fig. 8**, the assembly of two monolayer nanosheets into multilayers with two nanosheets is an endothermic process, which means that the process is not thermodynamically spontaneous. Therefore, from the perspective of theoretical calculations, the exfoliated single-layer nanosheet is thermodynamically inclined to maintain its monolayer property.

Moreover, we have characterized the morphology and layer spacing of unexfoliated and exfoliated coatings by STEM. As shown in **Response Fig. 9**, for the unexfoliated VMT coating, its layers are tightly and neatly arranged with an average layer spacing of 0.95 nm. In contrast, the layers of MPVMT coating are loosely and irregularly piled. Although the morphology of MPVMT coating appears to be “nearly multilayered”, formed by the stack of monolayer nanosheets under the action of binders, the average layer spacing of MPVMT coating is increased to 1.20 nm, which is 26% more than the layer spacing of unexfoliated VMT coating. Therefore, MPVMT coating with a stack of monolayers is significantly different from natural multilayer VMT coating.

In summary, we consider that the MPVMT coating is a “multilayered structure” stacked by monolayer nanosheets with the help of binders, but each layer of vermiculite nanosheets in the “multilayered structure” maintains the property of monolayer.

**Response Fig. 8** The calculated adsorption energy of the assembly process from two monolayer nanosheets to multilayers with two nanosheets.

**Response Fig. 9 a,b** Cross-sectional STEM image of unexfoliated vermiculite using the high-angle annular dark-field (HAADF) mode and corresponding distance of the five nanosheets. **c,d** Cross-sectional STEM image of the MPVMT using the HAADF mode and corresponding distance of the five nanosheets.

**Comment 4.** Supplementary Fig. 8 marks peaks below 10 degrees as VMT, but no other peaks from VMT are observed. While this provides some evidence for the

presence of VMT on the surface of the Zn anode, further characterization techniques and analysis are necessary to confirm its presence conclusively.

**Response:** Thanks for your valuable suggestions. We are sorry for the unclear marks in the XRD pattern of MPVMT@Zn. In the original XRD pattern, peaks from 10 to 38 degrees are also observed, just because the intensity of peaks is too small to be marked. Now we have magnified the peaks from 5 to 38 degrees shown in **Response Fig. 10**, and the peak of MPVMT marked in the figure is consistent with the peak of vermiculites previously reported (*Appl. Clay Sci.* 2009, 42, 368-378; *Nanoscale* 2018, 10, 23182-23190), indicating the presence of VMT on the surface of the Zn anode.

Additionally, since VMT consists of one Mg-based octahedral sheet sandwiched between two tetrahedral silicate sheets (*Nat. Commun.* 2015, 6, 7602; *Nat. Commun.* 2021, 12, 1124), main elements (Si, Mg and O) were characterized by energy-dispersive X-ray spectroscopy mapping to further verify the presence of VMT on the surface of the Zn anode (**Response Fig. 11**)

**Response Fig. 10 XRD patterns of bare Zn and MPVMT@Zn.**

**Response Fig. 11** SEM image and corresponding elemental maps of MPVMT@Zn.

We have revised our manuscript accordingly.

Revised SI, Page 13, the figure has been modified:

**Supplementary Fig. 7** XRD patterns of bare Zn and MPVMT@Zn.

**Comment 5.** While the proposed approach shows promise for protecting the Zn

anode, further evidence is needed to determine the chemical stability of vermiculite during the aqueous zinc ion battery process. This will be essential for evaluating the long-term efficacy of the proposed approach.

**Response:** We are so grateful for your recognition of our proposed approach to protecting Zn anodes and sincerely appreciate your helpful advice. We have added the characterization of X-ray photoelectron spectroscopy (XPS) to confirm the chemical stability of vermiculites during battery cycling. As shown in **Response Fig. 12**, binding energies of main elements (Si and Mg) remain unchanged before and after cycling, demonstrating excellent chemical stability of vermiculites.

**Response Fig. 12** Si 2*p* and Mg 2*p* XPS spectra of MPVMT@Zn before and after 200 cycles at 10 mA cm-2/1 mAh cm-2.

**We have revised our manuscript accordingly.**

**Revised SI, Page 20, a figure has been added:**

**Supplementary Fig. 13** Si 2p (a) and Mg 2p (b) XPS spectra of MPVMT@Zn before and after 200 cycles at  $10 \text{ mA cm}^{-2}/1 \text{ mAh cm}^{-2}$ , demonstrating the chemical stability of MPVMT coatings during battery cycling.

**Revised SI, Page 4, a sentence has been added in Materials characterization:**

“A confocal laser scanning microscope (CLSM, VK-X1000, KEYENCE) was used to analyze the roughness of the surface after cycling at a large area. XPS measurements were conducted by a PHI 5000 Versa Probe II In-Situ XPS.”

**Revised manuscript, Page 10, a sentence has been added:**

“while 2D MPVMT nanosheets on MPVMT@Zn anodes are not peeled off showing high adhesion and excellent chemical stability and parallel Zn platelets are consistently sustained under nanosheets (Supplementary Figs. 12 and 13).”

### **Reviewer #3 (Revision marked with red in the revised manuscript and SI)**

**General comments:** This work presents a novel zinc anode protective coating based on a monolayer porous vermiculite (MPVMT) with the lowest lattice mismatch of Zn (002) plane. The MPVMT protective coating possesses natural negative charge insulation properties and has demonstrated effective inhibition of unwanted side reactions. Additionally, it has been designed with vertical transmission channels to enhance ion transportation.

The coating strategy reported in this work is a common approach (*Nat. Commun.* 2021, 12 (1), 6606.).

**Response:** We are so grateful for your recognition of the novelty of MPVMT coating materials and thanks for your comment on coating strategy. Although the coating strategy is a common approach, there are still many reports using this method because of its effectiveness, simplicity and feasibility (*Nat. Energy* 2023, 8, 340-350; *Nat. Commun.* 2023, 14, 641; *Sci. Adv.* 2022, 8, eabm5766). Rather than the coating method, it is more important to seek suitable coating/substrate materials to solve the problems existing in specific systems.

The difference between Work 1 (*Nat. Commun.* 2021, 12 (1), 6606.) and our work:

- i) Mechanism of regulating Zn oriented growth: Work 1 developed a fluorinated covalent organic framework (FCOF) as a protective layer for Zn anodes. From the perspective of regulation of the surface energy of Zn crystal, the introduced F atoms exhibit strong interaction with the underlying Zn atoms, leading to a low surface energy for Zn (002) to induce the horizontal growth of Zn deposits. In contrast, our strategy is to find a substrate with ultra-low lattice mismatch with Zn(002) based on the proposed extended substrate screening strategy (ESSS), so that the lattice strain of deposits on substrates is small and the orientation correlation is strong, achieving oriented growth of Zn(002) for dendrite-free anodes.
- ii) Inhibiting side reactions: Work 1 utilized the hydrophobic and electronegative properties and of F atoms to suppress the water-related side reactions. In comparison, the hydrophilic group and negative charges on the surface of MPVMT coatings can grab water molecules and repel  $\text{SO}_4^{2-}$  to restrain side reactions, respectively.
- iii) Promoting ion transport: Negative fluorinated nanochannels in FCOF of Work 1 can facilitate ion transport. In our work, we improve ion conduction by combining constructed pore structures and local acceleration fields.

- iv) Synthesis of material: Work 1 used several organic reagents and acidic solvent and synthesized FCOF coatings under complex conditions, including heating, liquid nitrogen freezing and vacuum, indicating that the process of FCOF film is uneconomic and unsustainable. In contrast, the MPVMT material was synthesized using only natural mineral raw materials and distilled water solvent, and at normal temperature and pressure, demonstrating the simplicity and sustainability of preparing MPVMT.

Therefore, our work is significantly different from the previous work. More importantly, we propose ESSS and advance the understanding of Zn nucleation mechanism, paving the way for realizing high-rate and stable Zn-metal batteries and probably applying them to other metal-based batteries. Besides, our simple and sustainable preparation makes MPVMT material a viable option to fulfill practical demands.

**Revised manuscript, Page 27, a reference has been cited as follows:**

13. Zhao, Z. *et al.* Horizontally arranged zinc platelet electrodeposits modulated by fluorinated covalent organic framework film for high-rate and durable aqueous zinc ion batteries. *Nat. Commun.* **12**, 6606 (2021).

The proposed strategy of inducing zinc ion deposition on the (002) crystal plane by designing a protective layer with the lowest lattice mismatch has been previously reported (*Adv. Mater.* 2021, 33 (21), e2100187., *ACS Energy Lett.* 2021, 7 (1), 197.), indicating that the work lacks innovation.

**Response:** We sincerely appreciate the valuable comments from the reviewer and are sorry that the innovation of our work may not be clearly shown in the previous manuscript. Now we have reorganized Abstract, Introduction, Discussion and Fig.1&2 to highlight the innovation of our work. First of all, please let us emphasize the innovation of our work.

#### 1. The innovation of our work

First, our work proposes an ESSS based on the traditional substrate/coating screening criterion (TSSC), namely  $d_{\text{substrate}}: d_{\text{Zn}(002)}=1:1 \rightarrow d_{\text{substrate}}: d_{\text{Zn}(002)}=n:1$  ( $n=1, 2$ ), and discovers the substrate material (VMT) with the lowest lattice mismatch ( $\delta=0.38\%$ ) with  $\text{Zn}(002)$  reported so far, together with unique “planting  $\text{Zn}(002)$  seeds” mechanism for “ $2d_{\text{Zn}(002)}$ ” substrates. It is generally believed that the preferred orientation of  $\text{Zn}(002)$  has a vital effect on the cycling durability of Zn anodes, because of the relatively smooth surface and lower chemical activity of  $\text{Zn}(002)$  plane (*Science* 2019,

366, 645-648; *Adv. Mater.* 2021, 33, e2100187; *Energy Environ. Sci.* 2022, 15, 5017-5038). Essentially, the anisotropy of the substrates significantly influences the orientation of Zn electrodeposits (*Adv. Mater.* 2023, 35, e2211961; *Adv. Mater.* 2022, 34, e2202552). As shown in **Response Fig. 13**, due to the limit of TSSC, there is an urgent need to explore new substrate materials that not only possess low lattice mismatch with Zn(002), but also inhibit side reactions and promote ion transport. Here, as shown in **Response Fig. 14**, we conjecture and verify the feasibility of selecting suitable coatings for Zn anodes based on the ESSS. From a series of calculated phyllosilicates satisfying  $d_{\text{substrate}} \approx 2d_{\text{Zn}(002)}$ , we select VMT, which has the lowest lattice mismatch ( $\delta=0.38\%$ ) as the model to confirm the effectiveness of “ $2d_{\text{Zn}(002)}$ ” substrates for Zn anodes protection. As a result, we find that VMT layers allow to “plant Zn(002) seeds” in their planes, thus guiding subsequent horizontal growth of Zn and achieving dendrite-free Zn electrodes. This work extends SSS and advances the understanding of Zn nucleation mechanism, paving the way for realizing high-rate and stable Zn-metal batteries and probably applying them to other metal-based batteries.

**Response Fig. 13** **a** Schematic of challenges existing in Zn anodes. **b** Different lattice strain and orientation correlation of Zn electrodeposition formed on substrates with incoherent ( $\delta > 25\%$ ), semi-coherent ( $25\% \geq \delta \geq 5\%$ ) and coherent ( $\delta < 5\%$ ) interfaces and TSSC used in previous work.

**Response Fig. 14 a** Schematic of our work design of MPVMT layers based on ESSS to stabilize Zn anodes. **b** Comparison of lattice mismatch among eight typical phyllosilicates screened based on  $d_{\text{substrate}} : d_{\text{Zn}(002)} = 2:1$ . **c** Comparison of lattice mismatch between recently reported substrates and our work.

Second, the proposed MPVMT coatings not only inhibit side reactions through dual effects of surface functional groups and charge repulsion, but also promote  $\text{Zn}^{2+}$  transport by designed acceleration channels. Based on TSSC, only a few Zn-based alloy/metal substrates enable to form coherent interfaces with Zn(002) (*Nat. Commun.* 2021, 12, 237; *Adv. Mater.* 2022, 34, e2202552). Unfortunately, these conductive substrates cannot isolate Zn electrodes from aqueous electrolytes, resulting in unavoidable side reactions (*Energy Environ. Sci.* 2022, 15, 1086-1096; *Angew. Chem. Int. Ed.* 2023, 62, e202212695). In contrast, based on ESSS, we find insulating vermiculite coatings not only form coherent interfaces but also effectively inhibit side reactions (**Response Fig. 15**). Meanwhile, insulating coatings usually face the problem of slow ion transport. Here, to address the issue, we design a monolayer porous vermiculite with acceleration channels for  $\text{Zn}^{2+}$ , the ion transport behavior visualized by COMSOL simulations (**Response Fig. 16**), resulting in superior cyclability even at ultrahigh current densities of  $50 \text{ mA cm}^{-2}$ . Therefore, our design develops a promising solution to the contradiction between side reactions and ion transport, providing a new

strategy for high-rate and long-cycle Zn-metal batteries.

**Response Fig. 15** **a** In situ electrochemical GC profile for bare Zn (left) and MPVMT@Zn (right) symmetric cells at  $20 \text{ mA cm}^{-2}$  and  $2 \text{ mAh cm}^{-2}$ . **b** XRD patterns of bare Zn and MPVMT@Zn after immersing in 2 M  $\text{ZnSO}_4$  electrolyte for 1 and 6 days.

**Response Fig. 16** Electric field distribution (a) and Zn ion flux (b) of bare Zn and MPVMT@Zn at  $10 \text{ mA cm}^{-2}$ .

Third, we first prepare monolayer porous vermiculites by a simple, large-scale and sustainable method. At present, the only reported method for preparing MPVMT materials involves obtaining exfoliated vermiculite nanosheets by an ion exchange method, and then preparing porous vermiculite nanosheets by chemical etching (*J. Mater. Chem. A* 2021, 9, 14576-14581), resulting in a complex and unsustainable preparation process. To overcome these issues, we develop a one-step ultrasonic and agitating method using only deionized water as the solvent to easily produce large quantities of MPVMT materials (**Response Fig. 17**). Hence, our simple and sustainable preparation makes it a viable option to fulfill practical demands.

**Response Fig. 17** Schematic of simple, sustainable and scalable preparation of MPVMTs.

## 2. Differences between previous researches and our work

Work 2 (*Adv. Mater.* 2021, 33 (21), e2100187) directly uses Zn metal with 002 preferential orientation prepared by rolling and annealing as anodes, not involving the design of coating materials with low lattice mismatch with Zn(002). Although Zn(002) anode can induce Zn deposition behavior to a certain extent, it cannot effectively inhibit side reactions and corrosion, leading to insufficient Coulombic efficiency (97.7%) and cycle life (500 h at  $1 \text{ mA cm}^{-2}/1 \text{ mAh cm}^{-2}$ ). Besides, slow ion transport and large interface impedance still exist in the Zn(002) anode, resulting in high overpotential (80 mV at  $1 \text{ mA cm}^{-2}$ ). In contrast, our MPVMT coating with the lowest lattice mismatch not only induces the horizontal growth of Zn deposition, but also suppresses side reactions and promotes ion transport, thus stably cycling for 2000 h at  $1 \text{ mA cm}^{-2}/1 \text{ mAh cm}^{-2}$  with a small overpotential of 50 mV.

For Work 3 (*ACS Energy Lett.* 2021, 7 (1), 197), it should be noted that the lattice parameter of Zn(002) used to calculate the lattice mismatch ( $\delta=0.81\%$ ) is  $2.47 \text{ \AA}$ , while we used  $2.665 \text{ \AA}$  which is in agreement with the previous works (*Science* 2019, 366, 645-648; *Adv. Mater.* 2021, 33, e2105951; *Adv. Funct. Mater.* 2023, 2305204; *ACS Nano* 2021, 15, 14631-14642; *J. Chem. Phys.* 1935, 3, 605-616; *Small* 2021, 18, e2104148). Using the lattice parameter of  $2.665 \text{ \AA}$ , the lattice mismatch reported in Work 3 should be 8.2% (rather than 0.81%), which is much larger than ours (0.38%). Hence, to the best of our knowledge, the lattice mismatch between vermiculite and Zn(002) found in our work is the lowest reported so far. Besides, TiN coating in Work 3 conforms to TSSC ( $d_{\text{TiN}}: d_{\text{Zn(002)}} \approx 1:1$ ), while our MPVMT coating is selected based on the ESSS ( $d_{\text{MPVMT}}: d_{\text{Zn(002)}} \approx 2:1$ ) with new nucleation mechanism (“planting Zn(002) seeds”).

Therefore, our work significantly differs from the previous works, both in terms of material design strategy and Zn nucleation mechanism.

**We have revised our manuscript accordingly.**

**Revised manuscript, Page 1, the title has been modified:**

“An extended substrate screening strategy enabling a record-low lattice mismatch for highly reversible Zn anodes”

**Revised manuscript, Page 2, the abstract has been modified:**

“Aqueous zinc (Zn) batteries possess intrinsic safety and cost-effectiveness, but dendrite growth and side reactions of Zn anodes hinder their practical application. Here, we propose the extended substrate screening strategy (SSS) for stabilizing Zn anodes and verify its availability ( $d_{\text{substrate}}: d_{\text{Zn}(002)}=1:1 \rightarrow d_{\text{substrate}}: d_{\text{Zn}(002)}=n:1, n=1, 2$ ). From a series of calculated phyllosilicates satisfying  $d_{\text{substrate}} \approx 2d_{\text{Zn}(002)}$ , we select vermiculite, which has the lowest lattice mismatch ( $\delta=0.38\%$ ) reported so far, as the model to confirm the effectiveness of “ $2d_{\text{Zn}(002)}$ ” substrates for Zn anodes protection. Then, we develop a monolayer porous vermiculite (MPVMT) through a large-scale and green preparation as a functional coating for Zn electrodes. Unique “planting Zn(002) seeds” mechanism for “ $2d_{\text{Zn}(002)}$ ” substrates is revealed to induce the oriented growth of Zn deposits. Additionally, MPVMT coatings effectively inhibit side reactions and promote  $\text{Zn}^{2+}$  transport. Consequently, MPVMT@Zn symmetric cells operate stably for over 300 h at an ultrahigh current density of  $50 \text{ mA cm}^{-2}$ . This work extends traditional SSS and advances the understanding of Zn nucleation mechanism, paving the way for realizing high-rate and stable Zn-metal batteries.”

**Revised manuscript, Pages 3 and 4, the 2nd and 3rd paragraphs of the introduction have been modified:**

“Essentially, the anisotropy of the substrates significantly influences the orientation of Zn electrodeposits. According to the lattice mismatch ( $\delta$ ) between deposited Zn and substrates, the contact interface can be divided into incoherent ( $\delta > 25\%$ ), semi-coherent ( $25\% \geq \delta \geq 5\%$ ) and coherent interfaces ( $\delta < 5\%$ ). The lower lattice mismatch, the smaller lattice strain of deposits on substrates, the stronger orientation correlation. Hence, it is necessary to search for a suitable substrate with a low lattice mismatch with Zn(002) plane. At present, based on the traditional substrates screening criterion (TSSC) that lattice parameters of substrates should be close to that of Zn(002) (i.e.  $d_{\text{substrate}}: d_{\text{Zn}(002)}=1:1$ ) (**Fig. 1b**), several substrates have been reported including two-dimensional (2D) materials (e.g., graphene,  $\text{MoS}_2$ ) and highly oriented metals (e.g., Sn(200), In(002)). However, most of them form semi-coherent interfaces with Zn(002). Only a few Zn-based alloys or single-crystal Zn(002) anodes enable to

form coherent interface. Unfortunately, these alloy or metal substrates are electrically conductive, and thus they cannot isolate Zn electrodes from aqueous electrolytes, resulting in unavoidable side reactions. Hence, there is an urgent need to explore possible extended substrate screening strategy (ESSS) to guide the search for new substrate materials that not only possess low lattice mismatch with Zn(002), but also inhibit side reactions and promote ion transport, especially at high current densities.

Here, we conjecture and verify the application potential of selecting suitable coatings for Zn anodes based on ESSS, namely  $d_{\text{substrate}}:d_{\text{Zn}(002)}=1:1 \rightarrow d_{\text{substrate}}:d_{\text{Zn}(002)}=n:1, n=1, 2$ . First, through calculation, we discover eight typical phyllosilicates that meet “ $d_{\text{substrate}} \approx 2d_{\text{Zn}(002)}$ ”. Among them, vermiculite (VMT) with the lowest lattice mismatch ( $\delta=0.38\%$ ) reported so far, is selected as our model clay system to study the regulatory mechanism of Zn deposition on “ $2d_{\text{Zn}(002)}$ ” substrates and its comprehensive effect on stabilizing Zn anodes protection (**Figs. 1c and 1d**). Next, we synthesize monolayer porous vermiculites (MPVMTs) as coatings for Zn anodes (MPVMT@Zn). Notably, the preparation of MPVMT is simple and sustainable, and thus easy to achieve large-scale preparation. Owing to the ultralow lattice misfit and **double large lattice of MPVMT than Zn**, together with strong interaction with deposited Zn atom, **MPVMT layers allow to plant Zn(002) seeds** in their planes, thus guiding subsequent horizontal growth of Zn and achieving dendrite-free Zn electrodes. **Additionally, the non-conductive MPVMT with hydrophilic groups on the surface** effectively mitigates HER and the formation of byproducts. Moreover, the monolayer porous structure with negative charges provides acceleration transport channels for the migration of  $\text{Zn}^{2+}$ , leading to superior cyclability at high current densities. As a result, the MPVMT@Zn symmetric cells exhibit stable cycling for over 800 h at a high current density of  $10 \text{ mA cm}^{-2}$  and 300 h at an ultrahigh current density of  $50 \text{ mA cm}^{-2}$ . Furthermore, the assembled MPVMT@Zn|| $\text{MnO}_2$  full cells show a stable lifespan of 500 cycles.”

**Revised manuscript, Page 5, a figure has been added:**

**Fig. 1 Schematic demonstrating the origin of the proposed ESSS and its verification based on MPVMT coatings.** **a** Schematic of challenges existing in Zn anodes. **b** Different lattice strain and orientation correlation of Zn electrodeposition formed on substrates with incoherent ( $\delta > 25\%$ ), semi-coherent ( $25\% \geq \delta \geq 5\%$ ) and coherent ( $\delta < 5\%$ ) interfaces and TSSC used in previous work. **c** Schematic of our work design of MPVMT layers based on ESSS to stabilize Zn anodes. **d** Comparison of lattice mismatch among eight typical phyllosilicates screened based on  $d_{\text{substrate}} : d_{\text{Zn}(002)} = 2:1$ . **e** Comparison of lattice mismatch between recently reported substrates and our work.

Revised manuscript, Page 6, several sentences have been added:

“According to the calculation results of adsorption energies of Zn atom on different Zn crystal planes (**Supplementary Fig. 1, Supplementary Note 1**), it is necessary to search for a substrate highly matched with Zn(002) to induce the planar deposition of Zn and inhibit dendrite growth. **Based on the conjecture of ESSS, eight typical phyllosilicate clay materials satisfying “ $d_{\text{substrate}}: d_{\text{Zn}(002)}=2:1$ ” were found by calculation.**”

“As shown in **Fig. 2a**, each layer of VMT consists of one Mg-based octahedral sheet sandwiched between two tetrahedral silicate sheets. **The cross-sectional scanning transmission electron microscopy (STEM) image of VMT clearly shows that tetrahedral silicate sheets have hollow sites between the atoms.**”

Revised manuscript, Page 8, the Fig.2 has been modified:

**Fig. 2 Preparation and characterization of MPVMT@Zn anode.** **a** Cross-sectional schematic of a monolayer vermiculite and cross-sectional STEM image of an unexfoliated vermiculite using the high-angle annular dark-field mode. **b** Schematic of the mass production of MPVMT materials through a sustainable method. **c** Atomic force microscopy (AFM) image of MPVMT nanosheets. The inset is the height profile of the corresponding lines. **d** Statistics for the thickness of MPVMT nanosheets. **e** Typical TEM image and SAED pattern (inset) of MPVMT. **f, g** Contact angles of bare Zn (**f**) and MPVMT@Zn (**g**). **h, i** KPFM images of MPVMT@Zn (**h**) and bare Zn (**i**). **j** Voltage response to a current of 2 mA of MPVMT@Zn and bare Zn electrodes. **k** The prepared MPVMT@Zn with an area of 300 cm2 fabricated using a scalable strategy.

**Revised manuscript, Page 11, several sentences have been added:**

“It is concluded that the strong adsorption of MPVMT layers to Zn atoms and the super match allow for “planting Zn(002) seeds” on the “2dZn(002)” substrate, thereby achieving the subsequent preferred deposition and alleviating dendrite growth even at high current densities. The above evidence confirms the validity of ESSS for screening substrates that can induce the orientation of Zn electrodeposits. Based on ESSS, we have modified the formula for calculating the lattice mismatch  $\delta$  as follows:

$$\delta = \begin{cases} \left| \frac{d_{\text{substrate}} - d_{\text{Zn}(002)}}{d_{\text{Zn}(002)}} \right|, & d_{\text{substrate}} \leq 1.5 d_{\text{Zn}(002)} \\ \left| \frac{d_{\text{substrate}} - 2 d_{\text{Zn}(002)}}{d_{\text{Zn}(002)}} \right|, & 1.5 d_{\text{Zn}(002)} < d_{\text{substrate}} < 2.5 d_{\text{Zn}(002)} \end{cases}$$

where d is the lattice parameter of the substrate or Zn(002) plane.”

**Revised manuscript, Page 26, the discussion has been added:**

“In summary, we develop and confirm the applicability of ESSS ( $d_{\text{substrate}}: d_{\text{Zn}(002)}=1:1 \rightarrow d_{\text{substrate}}: d_{\text{Zn}(002)}=n:1, n=1, 2$ ). Taking the vermiculite ( $d_{\text{vermiculite}} \approx 2d_{\text{Zn}(002)}$ ) as a model, we prepare 2D hydrophilic and insulating MPVMTs by a simple and eco-friendly method as protective coatings for Zn anode. Due to MPVMT’s ultralow lattice mismatch ( $\delta=0.38\%$ ), “2dZn(002)” structure cell and strong adsorption to Zn deposits, the coatings induce (002) oriented growth of Zn by “planting Zn(002) seeds” mechanism, realizing dendrite-free anodes. Additionally, the hydrophilic functional groups and negative charges on the surface of MPVMTs greatly promote the de-solvation of Zn2+ and homogenize the electric field, synergistically suppressing side reactions. Furthermore, the constructed ion acceleration channels significantly facilitate Zn2+ transport, achieving high-rate and long-life MPVMT@Zn anodes. Consequently, the MPVMT@Zn anodes deliver a superhigh cumulative capacity of 7500 mAh cm-2 at a

current density of  $50 \text{ mA cm}^{-2}$ , outperforming most reported modified Zn anodes. The proposed ESSS and new nucleation mechanism provide new insights into highly reversible Zn metal batteries and may apply to other metal-based batteries.”

**Revised manuscript, Page 27, two references have been added:**

14. Zheng, J. *et al.* Preferred Orientation of TiN Coatings Enables Stable Zinc Anodes. *ACS Energy Lett.* 7, 197-203 (2021).

16. Zhou, M. *et al.* Surface-Preferred Crystal Plane for a Stable and Reversible Zinc Anode. *Adv. Mater.* 33, e2100187 (2021).

Moreover, while insulating coatings can effectively suppress side reactions, it does not provide a novel solution to solve the important issue of hindering ion transport. The vertical transport channel mentioned has been already demonstrated (*Adv. Funct. Mater.* 2020, 30 (21).), and this work does not provide a robust characterization of the formation mechanism and ion migration behavior of this vertical channel.

**Response:** Thanks for your comments. In Work 4 (*Adv. Funct. Mater.* 2020, 30 (21).), the authors reported that the interlayer of kaolin nanosheets acted as a vertical transport channel for  $\text{Zn}^{2+}$ . In contrast, the transport channels proposed in our work are composed of porous structures of vermiculite materials. Moreover, the channels we constructed utilize the negative charge of vermiculite itself and have the effect of accelerating ion transport, which is different from the above reference. The force diagram of  $\text{Zn}^{2+}$  in two kinds of channels is shown in **Response Fig. 18**.

**Response Fig. 18** The force diagram of  $\text{Zn}^{2+}$  in two kinds of transport channels: Work 4 (a) and our work (b).

Besides, characterizations of the formation mechanism and ion migration behavior of

ion transport channels are shown as follows.

The acceleration transport channels of MPVMT coating are formed by its porous structures and local acceleration electric field (LAEF) resulted from negatively charged characteristics. The MPVMT's porous structure is characterized by AFM and TEM (Response Fig. 19), and its negative charge property is proven by KPFM and Zeta potential (Response Fig. 20).

**Response Fig. 19** a Atomic force microscopy (AFM) image of MPVMT nanosheets. The inset is the height profile of the corresponding lines. b Typical TEM image and SAED pattern (inset) of MPVMT.

**Response Fig. 20** a,b KPFM images of MPVMT@Zn (a) and bare Zn (b). c Zeta potential ( $-25.9$  mV) of a MPVMT dispersion.

To verify the advantages of porous structures and LAEF for ion transport, we compared the kinetics of MPVMT, non-porous vermiculite and bare Zn anodes by EIS and rate performance testing. Additionally, based on the experimental data of MPVMT material, we carried out COMSOL simulations to further elucidate the transport behavior of  $Zn^{2+}$  in the acceleration channels. To be honest, ion transport channels are difficult to be strictly vertical and we can also see that ion transport channels are not strictly vertical paths through COMSOL simulations (Response Fig. 21). We are so sorry if the

reviewer feels confused by this misunderstanding expression and have revised “vertical acceleration transport channel” to “acceleration transport channel” in the revised manuscript and SI, making it more appropriate.

**Response Fig. 21** Electric field distribution (a) and Zn ion flux (b) of MPVMT@Zn at  $10 \text{ mA cm}^{-2}$  by COMSOL simulations.

Moreover, we conducted chronoamperometry (CA) to further demonstrate the superiority of MPVMT coatings experimentally (**Response Fig. 22**). The charge transfer resistances ( $R_{ct}$ ) of MPVMT@Zn before and after polarization both are much smaller than that of bare Zn, and MPVMT@Zn shows transference number of 0.67, over 3 times than bare Zn (0.20). Besides, we have added the ionic conductivity of MPVMT@Zn ( $29.79 \text{ mS cm}^{-1}$ ) and bare Zn ( $10.50 \text{ mS cm}^{-1}$ ) to evaluate the transfer rate of ions, as shown in **Response Fig. 23**. These results firmly verify the significantly improved transport of  $\text{Zn}^{2+}$  by acceleration channels of MPVMT coatings designed by porous structures and negative local acceleration fields.

**Response Fig. 22 a, b** CA curves of bare Zn (a) and MPVMT@Zn (b) symmetric cells. **c, d** Nyquist plots for bare Zn (c) and MPVMT@Zn (d) symmetric cells before and

after polarization. The insets are their fitting equivalent circuit. **e** The calculated transference number of bare Zn and MPVMT@Zn.

**Response Fig. 23** **a** EIS results of the symmetric cells in SS/MPVMT@Zn/GF or SS/bare Zn/GF configurations (immersed in 2 M  $\text{ZnSO}_4$ ). **b** Calculated ionic conductivity based on the EIS results.

We have revised our manuscript accordingly.

Revised manuscript, Page 29, a reference has been added:

49. Deng, C. *et al.* A Sieve-Functional and Uniform-Porous Kaolin Layer toward Stable Zinc Metal Anode. *Adv. Funct. Mater.* **30**, 2000599 (2020).

Revised SI, Page 30, a figure has been added:

**Supplementary Fig. 22** **a** EIS results of the symmetric cells in SS/MPVMT@Zn/GF or SS/bare Zn/GF configurations (immersed in 2 M  $\text{ZnSO}_4$ ). **b** Calculated ionic conductivity based on the EIS results.

Revised manuscript, Page 19, several sentences have been added:

“Ion transport behavior is further determined experimentally.  $Zn^{2+}$  transference number (ZTN) is evaluated based on chronoamperometry measurements and EIS results before and after polarization. The MPVMT layers display a ZTN of 0.67, three times higher than the bare Zn (0.20) (**Supplementary Fig. 21 and Supplementary Note 4**). Moreover, as shown in **Supplementary Fig. 22**, the ionic conductivity of MPVMT@Zn is calculated to be  $29.79 \text{ mS cm}^{-1}$ , that is, 2.8 times greater than that of bare Zn ( $10.50 \text{ mS cm}^{-1}$ ). These results again confirm that MPMVT accelerates  $Zn^{2+}$  transport and thus promotes ionic conductivity.”

Additionally, the electrochemical performance of the MPVMT@Zn symmetric cell and full cell is not outstanding. The low testing capacities of  $50 \text{ mA cm}^{-2}$ - $1 \text{ mAh cm}^{-2}$  and  $10 \text{ mA cm}^{-2}$ - $1 \text{ mAh cm}^{-2}$  do not demonstrate the ability of accommodating high-capacity zinc deposition, more high zinc deposition capacity data need to be provided. Based on these reasons, I do not recommend publishing this manuscript in Nature Communication.

**Response:** Thank you so much for the suggestions.

For symmetric cells, benefitting from regulated Zn deposition, inhibited side reactions and significantly improved ion transport, our MPVMT@Zn anodes not only show long cycle stability at wide current densities ( $1\sim 50 \text{ mA cm}^{-2}$ ), but also stably operate at high current capacities ( $5 \text{ mAh cm}^{-2}$ , **Response Fig. 24**). Additionally, we have added the cycle performance of MPVMT@Zn anodes at higher current capacities ( $10 \text{ mAh cm}^{-2}$ , **Response Fig. 25**) to demonstrate the promising ability of accommodating high-capacity Zn deposition. Notably, the current density and cumulative plating capacity enabled by the MPVMT@Zn symmetric cell are much higher than most of previously reported values from Zn electrodes based on different modification strategies (**Response Fig. 26**).

**Response Fig. 24** Cycling performance of symmetric cells with or without MPVMT layer protection at  $5 \text{ mA cm}^{-2} / 5 \text{ mAh cm}^{-2}$ .

**Response Fig. 25** Cycling performance of symmetric cells with or without MPVMT layer protection at  $0.6 \text{ mA cm}^{-2} / 10 \text{ mAh cm}^{-2}$  with the same current capacity of  $10 \text{ mAh cm}^{-2}$ .

**Response Fig. 26** Performance comparison between MPVMT@Zn and recently reported Zn anodes using different strategies.

For Zn||MnO2 full cells, we have added the performance comparison between previous researches with different modified Zn anodes and this work. As shown in **Response Fig. 27 (see details in Response Table 1)**, MPVMT@Zn||MnO2 batteries exhibit superior improvement rate of capacity retention than most previous reports, demonstrating the advantages of MPVMT@Zn anodes.

**Response Fig. 27 Performance comparison of Zn||MnO2 full cells from previous researches with different modified Zn anodes and this work.**

| Anode                 | Current Density (A g -1 ) | Mass loading (mg cm -2 ) | Capacity retention improvement rate | Cell type  | Ref.                               | Year |
|-----------------------|--------------------------------------|-------------------------------------|-------------------------------------|------------|------------------------------------|------|
| MPVMT                 | 0.616                                | 1~2                                 | 217% (500 cycles)                   | Coin cell  | This work                          |      |
|                       | 0.308                                |                                     | 254% (200 cycles)                   | Pouch cell |                                    |      |
| PS-Zn                 | 0.5                                  | 1.4                                 | 104% (200 cycles)                   | Coin cell  | Nat. Commun. 1          | 2022 |
| Zn-P-MIEC             | 0.5                                  | 1                                   | 125% (500 cycles)                   | Coin cell  | Adv. Mater. 2           | 2022 |
| Zn@In                 | 1                                    | /                                   | 239% (500 cycles)                   | Coin cell  | Energy Environ. Sci. 3  | 2022 |
| GBL additive          | 0.5                                  | /                                   | 134% (400 cycles)                   | Coin cell  | Adv. Energy Mater. 4    | 2022 |
| β-CD additive         | 1                                    | /                                   | 165% (500 cycles)                   | Coin cell  | Angew. Chem. Int. Ed. 5 | 2022 |
| ZF@F-TiO 2 | 1                                    | /                                   | 112% (300 cycles)                   | Coin cell  | Nat. Commun. 6          | 2020 |
| Zn@PFSA               | 0.616                                | /                                   | 230% (500 cycles)                   | Coin cell  | ACS Nano 7              | 2022 |
| PVDF-Sn@Zn            | 2                                    | 0.8                                 | 171% (500 cycles)                   | Coin cell  | Nat. Commun. 8          | 2023 |

|                           |       |         |                      |            |                                    |      |
|---------------------------|-------|---------|----------------------|------------|------------------------------------|------|
|                           | 2     |         | 177%(250 cycles) | Pouch cell |                                    |      |
| Zn-AAAn-COF               | 1     | 1.3~1.5 | 151%(500 cycles) | Coin cell  | Angew. Chem. Int. Ed. 9 | 2022 |
| TiN(200)@Zn               | 0.370 | ~2      | 141%(500 cycles) | Coin cell  | ACS Energy Lett. 10     | 2021 |
| PVA@SR-ZnMoO 4 | 1     | 1.5     | 127%(500 cycles) | Coin cell  | Energy Environ. Sci. 11 | 2022 |

**Response Table 1. Performance comparison of Zn||MnO2 full cells from previous researches with different modified Zn anodes and this work.**

### References

- Li, Q. *et al.* Tailoring the metal electrode morphology via electrochemical protocol optimization for long-lasting aqueous zinc batteries. *Nat. Commun.* **13**, 3699 (2022).
- Zhang, M. *et al.* Construction of mixed ionic-electronic conducting scaffolds in Zn powder: A scalable route to dendrite-free and flexible Zn anodes. *Adv. Mater.* **34**, e2200860 (2022).
- Xiao, P. *et al.* An anticorrosive zinc metal anode with ultra-long cycle life over one year. *Energy Environ. Sci.* **15**, 1638-1646 (2022).
- Huang, H. *et al.* Boosting Reversibility and Stability of Zn Anodes via Manipulation of Electrolyte Structure and Interface with Addition of Trace Organic Molecules. *Adv. Energy Mater.* **12**, 2202419 (2022).
- Qiu, M. *et al.* Anion-Trap Engineering toward Remarkable Crystallographic Reorientation and Efficient Cation Migration of Zn Ion Batteries. *Angew. Chem. Int. Ed.* **61**, e202210979 (2022).
- Zhang, Q. *et al.* Revealing the role of crystal orientation of protective layers for stable zinc anode. *Nat. Commun.* **11**, 3961 (2020).
- Hong, L. *et al.* Highly Reversible Zinc Anode Enabled by a Cation-Exchange Coating with Zn-Ion Selective Channels. *ACS Nano* **16**, 6906-6915 (2022).
- Cao, Q. *et al.* Gradient design of imprinted anode for stable Zn-ion batteries. *Nat. Commun.* **14**, 641 (2023).
- Guo, C. *et al.* Synergistic Manipulation of Hydrogen Evolution and Zinc Ion Flux in Metal-Covalent Organic Frameworks for Dendrite-free Zn-based Aqueous Batteries. *Angew. Chem. Int. Ed.* **61**, e202210871 (2022).
- Zheng, J. *et al.* Preferred Orientation of TiN Coatings Enables Stable Zinc Anodes. *ACS Energy Lett.* **7**, 197-203 (2021).
- Chen, A. S. *et al.* Multifunctional SEI-like structure coating stabilizing Zn anodes at a large current and capacity. *Energy Environ. Sci.* **16**, 275-284 (2023).

**We have revised our manuscript accordingly.**

Revised SI, Page 32, two figures have been added:

**Supplementary Fig. 24** Cycling performance of bare Zn and MPVMT@Zn symmetric cells at  $2 \text{ mA cm}^{-2}/2 \text{ mAh cm}^{-2}$  (a),  $10 \text{ mA cm}^{-2}/1 \text{ mAh cm}^{-2}$  (b),  $0.6 \text{ mA cm}^{-2}/10 \text{ mAh cm}^{-2}$  (c) and  $10 \text{ mA cm}^{-2}/10 \text{ mAh cm}^{-2}$  (d).

Revised SI, Page 44, a table has been added:

|       | Current               | Mass                    | Capacity    |           |           |      |
|-------|-----------------------|-------------------------|-------------|-----------|-----------|------|
| Anode | Density               | loading                 | retention   | Cell type | Ref.      | Year |
|       | ( $\text{A g}^{-1}$ ) | ( $\text{mg cm}^{-2}$ ) | improvement |           |           |      |
|       |                       |                         | rate        |           |           |      |
| MPVMT | 0.616                 | 1~2                     | 217%        | Coin cell | This work |      |

|                           |       |         |                      |            |                                    |      |
|---------------------------|-------|---------|----------------------|------------|------------------------------------|------|
|                           |       |         | (500 cycles)         |            |                                    |      |
|                           | 0.308 |         | 254%(200 cycles) | Pouch cell |                                    |      |
| PS-Zn                     | 0.5   | 1.4     | 104%(200 cycles) | Coin cell  | Nat. Commun. 1          | 2022 |
| Zn-P-MIEC                 | 0.5   | 1       | 125%(500 cycles) | Coin cell  | Adv. Mater. 2           | 2022 |
| Zn@In                     | 1     | /       | 239%(500 cycles) | Coin cell  | Energy Environ. Sci. 3  | 2022 |
| GBL additive              | 0.5   | /       | 134%(400 cycles) | Coin cell  | Adv. Energy Mater. 4    | 2022 |
| $\beta$ -CD additive      | 1     | /       | 165%(500 cycles) | Coin cell  | Angew. Chem. Int. Ed. 5 | 2022 |
| ZF@F-TiO 2     | 1     | /       | 112%(300 cycles) | Coin cell  | Nat. Commun. 6          | 2020 |
| Zn@PFSA                   | 0.616 | /       | 230%(500 cycles) | Coin cell  | ACS Nano 7              | 2022 |
| PVDF-Sn@Zn                | 2     | 0.8     | 171%(500 cycles) | Coin cell  | Nat. Commun. 8          | 2023 |
| Zn-AAAn-COF               | 1     | 1.3~1.5 | 177%(250 cycles) | Pouch cell |                                    |      |
| TiN(200)@Zn               | 0.370 | ~2      | 151%(500 cycles) | Coin cell  | Angew. Chem. Int. Ed. 9 | 2022 |
| PVA@SR-ZnMoO 4 | 1     | 1.5     | 141%(500 cycles) | Coin cell  | ACS Energy Lett. 10     | 2021 |
|                           |       |         | 127%(500 cycles) | Coin cell  | Energy Environ. Sci. 11 | 2022 |

**Supplementary Table 6. Performance comparison of Zn||MnO2 full cells from previous researches with different modified Zn anodes and this work.**

**Revised manuscript, Page 20, a sentence has been added:**

“The MPVMT@Zn electrode also exhibits excellent cycling stability at increasing current capacity of 2 mA cm-2/2 mAh cm-2 for 1500 h, 5 mA cm-2/5 mAh cm-2 for almost 700 h (Fig. 5f), 0.6 mA cm-2/10 mAh cm-2 for 500 h and 10 mA cm-2/10 mAh cm-2 for 400 h (Supplementary Fig. 24), much better than the bare Zn. ”

**Revised manuscript, Page 24, a sentence has been added:**

“Therefore, the superior rate performance and cycling stability of MPVMT@Zn||MnO2 batteries again validate the virtue of MPVMT design (Supplementary Table 6).”

Nonetheless, further investigation is also required to address the following issues.

**Comment 1.** While insulating protective coatings may hinder side reactions, they simultaneously reduce the transfer rate of ions and increase internal resistance. Regarding this issue, the vertical transport channels mentioned in the article have not been fully characterized and proven. Theoretical calculations require support from relevant experimental data.

**Response:** Thanks for your valuable suggestions.

The acceleration transport channels of MPVMT coating are formed by its porous structures and local acceleration electric field (LAEF) resulted from negatively charged characteristics. The MPVMT's porous structure is characterized by AFM and TEM (Response Fig. 19), and its negative charge property is proven by KPFM and Zeta potential (Response Fig. 20).

**Response Fig. 19 a** Atomic force microscopy (AFM) image of MPVMT nanosheets. The inset is the height profile of the corresponding lines. **b** Typical TEM image and SAED pattern (inset) of MPVMT.

**Response Fig. 20 a,b** KPFM images of MPVMT@Zn (a) and bare Zn (b). **c** Zeta potential (-25.9 mV) of a MPVMT dispersion.

To verify the advantages of porous structures and LAEF for ion transport, we compared the kinetics of MPVMT, non-porous vermiculite and bare Zn anodes by EIS and rate performance testing. Additionally, based on the experimental data of MPVMT material, we carried out COMSOL simulations to further elucidate the transport behavior of  $\text{Zn}^{2+}$  in the acceleration channels. To be honest, ion transport channels are difficult to be strictly vertical and we can also see that ion transport channels are not strictly vertical paths through COMSOL simulations (**Response Fig. 21**). We are so sorry if the reviewer feels confused by this misunderstanding expression and **have revised “vertical acceleration transport channel” to “acceleration transport channel” in the revised manuscript and SI, making it more appropriate.**

**Response Fig. 21** Electric field distribution (a) and Zn ion flux (b) of MPVMT@Zn at  $10 \text{ mA cm}^{-2}$  by COMSOL simulations.

Moreover, we conducted chronoamperometry (CA) to further demonstrate the superiority of MPVMT coatings experimentally (**Response Fig. 22**). The charge transfer resistances ( $R_{ct}$ ) of MPVMT@Zn before and after polarization both are much smaller than that of bare Zn, and MPVMT@Zn shows transference number of 0.67, over 3 times than bare Zn (0.20). Besides, we have added the ionic conductivity of MPVMT@Zn ( $29.79 \text{ mS cm}^{-1}$ ) and bare Zn ( $10.50 \text{ mS cm}^{-1}$ ) to evaluate the transfer rate of ions, as shown in **Response Fig. 23**. These results firmly verify the significantly improved transport of  $\text{Zn}^{2+}$  by acceleration channels of MPVMT coatings designed by porous structures and negative local acceleration fields.

**Response Fig. 22 a, b** CA curves of bare Zn (a) and MPVMT@Zn (b) symmetric cells. **c, d** Nyquist plots for bare Zn (c) and MPVMT@Zn (d) symmetric cells before and after polarization. The insets are their fitting equivalent circuit. **e** The calculated transference number of bare Zn and MPVMT@Zn.

**Response Fig. 23 a** EIS results of the symmetric cells in SS/MPVMT@Zn/GF or SS/bare Zn/GF configurations (immersed in 2 M ZnSO4). **b** Calculated ionic conductivity based on the EIS results.

**We have revised our manuscript accordingly.**

**Revised SI, Page 30, a figure has been added:**

**Supplementary Fig. 22 a** EIS results of the symmetric cells in SS/MPVMT@Zn/GF or SS/bare Zn/GF configurations (immersed in 2 M ZnSO4). **b** Calculated ionic conductivity based on the EIS results.

**Revised manuscript, Page 19, several sentences have been added:**

“Ion transport behavior is further determined experimentally. Zn2+ transference number (ZTN) is evaluated based on chronoamperometry measurements and EIS results before and after polarization. The MPVMT layers display a ZTN of 0.67, three times higher than the bare Zn (0.20) (Supplementary Fig. 21 and Supplementary Note 4). Moreover, as shown in Supplementary Fig. 22, the ionic conductivity of MPVMT@Zn is calculated to be 29.79 mS cm-1, that is, 2.8 times greater than that of bare Zn (10.50 mS cm-1). These results again confirm that MPMVT accelerates Zn2+ transport and thus promotes ionic conductivity.”

**Comment 2.** The assertion that natural clay minerals mentioned in the article are more easily cleaved into 2D nanosheets than other materials, is doubtful.

**Response:** We acknowledge the reviewer’s comments and are sorry for the unclear expression. Clays minerals here refer to expansive phyllosilicates (e.g., vermiculite, montmorillonite), not all clay materials. Expansive phyllosilicates, whose interlamellar spacing is generally larger than 1 nm, are composed of phyllosilicate layers with cations both adsorbed on the basal plane surfaces and existing in the space between the layers (*Adv. Colloid Interface Sci.* 1999, 82, 43-92; *Minerals in soil environments* 1989, 1, 789-828.). Because the native cations can be exchanged for others when the material is exposed to electrolytes through an ionic exchange, the crystals are relatively easy to cleave along the basal planes, forming high aspect ratio atomically thin sheets (*Nat. Mater.* 2021, 20, 1677-1682). Compared with other 2D materials such as graphite, its layer spacing is only 0.34 nm and there is no exchangeable cation, so it is relatively difficult to exfoliate into 2D nanosheets (*Chem. Mater.* 2007, 19, 4396–4404).

Therefore, clay minerals can be exfoliated in water by ionic exchange, which is more benign than chemical exfoliation of graphite and other 2D materials (*Science* 2013, 340, 1226419; *Nat. Commun.* 2015, 6, 7602). **Since we have reorganized the introduction to highlight our innovations, the sentence has been deleted in the revised manuscript.**

**Comment 3.** How can the strong hydrogen bonding between the surface silanol groups and oxygen functional groups of water molecules in the coating be demonstrated? Please provide more favorable characterizations to fully demonstrate.

**Response:** Thanks again for your valuable advice.

According to references (*J. Phys. Chem.* 1958, 62, 1168-1178; *J. Colloid Interface Sci.* 1989, 136, 85-94; *J. Colloid Interface Sci.* 1994, 165, 367-385), siloxane groups (Si–O–Si) on the surface of vermiculites are highly strained and may have dangling bonds, which will quickly react with liquid water or the saturated water vapor in the atmosphere to form silanol groups. Due to the ability of silanol groups to form strong hydrogen bonds to the oxygen of water molecules, the resulting surface will be hydrophilic and spread water.

Experimentally, we have added the Fourier Transform Infrared Spectroscopy (FTIR) of MPVMT materials. As shown in **Response Fig. 28**, the peak at  $3404\text{ cm}^{-1}$  arises due to the O–H stretching vibration of silanol groups of MPVMT, indicating the presence of silanol groups on the surface of MPVMT (*Appl. Clay Sci.* 2006, 32, 94-98; *Appl. Clay Sci.* 2015, 109-110, 22-32). By comparing water contact angles of bare Zn ( $95^\circ$ ) and MPVMT@Zn ( $27^\circ$ ) shown in **Response Fig. 29**, the hydrophilicity of Zn electrode is significantly enhanced by MPVMT coatings, because the formation of strong hydrogen bonds between the silanol groups on the surface of MPVMTs and the oxygen of water molecules, helps spread water.

**Response Fig. 28** FTIR spectra of MPVMT materials, suggesting the presence of silanol groups and absorbed water in MPVMT.

**Response Fig. 29** Contact angles of bare Zn (a) and MPVMT@Zn (b).

Furthermore, we have added the DFT calculation to elucidate the formation of strong hydrogen bonding between the surface silanol groups and oxygen of water molecules. As shown in **Response Fig. 30**, the adsorption energy of silanol on the surface of MPVMT to water is  $-0.57$  eV, suggesting that the process by which silanol adsorbs water is energy favorable. Moreover, after the formation of hydrogen bond between silanol and water molecule, the length of O–H bond in silanol is extended from  $0.976$  Å to  $1.147$  Å, and the length of the hydrogen bond is  $1.296$  Å. Consequently, silanol groups on the surface of MPVMT have strong hydrogen bonding with water molecules.

**Response Fig. 30** FTIR spectra of MPVMT materials, suggesting the presence of silanol groups and absorbed water in MPVMT.

**Comment 4.** Explain why the XRD of original pure zinc and MPVMT@Zn, as shown in Supplementary Fig. 8, does not exhibit the advantage of MPVMT@Zn's prominent (002) crystal plane?

**Response:** Thanks for your comment. Uneven plating/stripping and preferential aggregation of  $\text{Zn}^{2+}$  at the tip (“tip effect”) result in dendrite formation (*Energy Environ. Sci.* 2022, 15, 5017-5038). MPVMT coatings enable regulation of the  $\text{Zn}^{2+}$  behavior and induce  $\text{Zn}^{2+}$  to deposit along the Zn(002) orientation, thus achieving dendrite-free Zn anodes. However, the orientation of the original Zn foil depends on their production conditions, and coating MPVMT on Zn anodes without cycling will not change the orientation of the original Zn, since MPVMT coatings haven't had a chance to regulate  $\text{Zn}^{2+}$ .

**Comment 5.** The section on characterizing the coating's rapid ion transport capabilities and fast solvation mechanism does not provide sufficient experimental data, and it is not sufficient to rely on simulations alone for interpretation.

**Response:** We are grateful for your thoughtful comments.

For characterizing coating's rapid ion transport capabilities, we first conducted EIS of initial bare Zn/MPVMT@Zn symmetric cells (**Response Figs. 22c and 22d**). The result showed that  $R_{ct}$  of MPVMT@Zn at the initial state is much smaller than that of bare Zn, and the same trend exists after polarization. Second, CA was used to calculate the transference number of MPVMT@Zn (0.67) and bare Zn (0.20) (**Response Fig. 22e**). Third, we have added the ionic conductivity of MPVMT@Zn ( $29.79 \text{ mS cm}^{-1}$ ) and bare Zn ( $10.50 \text{ mS cm}^{-1}$ ) to further evaluate the transfer rate of ions (**Response Fig. 23**).

Additionally, the smaller polarization voltage of MPVMT@Zn anodes than that of bare Zn anodes at different current densities comprehensively indicates the greatly enhanced kinetics with MPVMT modification (**Response Fig. 31**). These evidences firmly verify the coating's rapid ion transport capabilities.

**Response Fig. 22 a, b** CA curves of bare Zn (**a**) and MPVMT@Zn (**b**) symmetric cells. **c, d** Nyquist plots for bare Zn (**c**) and MPVMT@Zn (**d**) symmetric cells before and after polarization. The insets are their fitting equivalent circuit. **e** The calculated transference number of bare Zn and MPVMT@Zn.

**Response Fig. 23 a** EIS results of the symmetric cells in SS/MPVMT@Zn/GF or SS/bare Zn/GF configurations (immersed in 2 M ZnSO4). **b** Calculated ionic conductivity based on the EIS results.

**Response Fig. 31** Rate performance of the bare Zn and MPVMT@Zn electrodes at different current densities from 0.5 to 10 mA cm-2 with the same capacity of 1 mAh cm-2 (a) and corresponding voltage hysteresis (b)

In a typical zinc ion electrolyte, Zn2+ solvated with six water molecules forms hydrated zinc ions [Zn(H2O)6]2+ instead of existing alone. Hence, solvated [Zn(H2O)6]2+ located in the electric double layer has to overcome the energy barrier to de-solvate and release a large amount of electrochemically reactive free water molecules to form zinc ions, resulting in sluggish kinetics (*Adv. Mater.* 2022, 34, e2206754, *Energy Environ. Sci.* 2021, 14, 3796-3839). Therefore, it is necessary to realize fast de-solvation by modification to improve kinetics of Zn-based batteries. We notice that the reviewer mentioned “fast solvation mechanism”. It might refer to “fast de-solvation mechanism” proposed in this work, so we will elaborate “fast de-solvation mechanism” next. Here, we proposed that the silanol groups of MPVMT’s surface interact with H2O by strong hydrogen bonding (proved in Comment 3), which weakens the interaction between the solvated water and Zn2+, thus facilitating the de-solvation process of hydrated Zn2+ and achieving fast de-solvation. Experimentally, we calculated the activation energy (Ea) which represents the de-solvation barrier for Zn2+ transport by temperature-dependent EIS (*Energy Environ. Sci.* 2020, 13, 503-510; *Adv. Mater.* 2021, 33, e2007388). The resultant Ea of MPVMT@Zn (59.9 kJ mol-1) is lower than that of the bare Zn (64.9 kJ mol-1) (**Response Fig. 32**), verifying the faster de-solvation of hydrated Zn2+.

**Response Fig. 32 Calculating  $E_a$  based on temperature-dependent EIS.** Nyquist plots at different temperatures of bare Zn (a) and MPVMT@Zn (c) symmetrical cells. Corresponding Arrhenius curves and calculated  $E_a$  of Zn (b) and MPVMT@Zn (d).

We have revised our manuscript accordingly.

Revised manuscript, Page 19, several sentences have been added:

“These results again confirm that MPMVT accelerates  $Zn^{2+}$  transport and thus promotes ionic conductivity. Specifically, when hydrated  $Zn^{2+}$  transport through MPVMT nanochannels is driven by electric field force from the battery and the coating, water molecules coordinated with  $Zn^{2+}$  will be captured because of the strong hydrogen bonding between silanol groups on MPVMT surface and the oxygen in water molecules. Consequently, the de-solvation of  $Zn^{2+}$  is effectively promoted by silanol groups. To further assure the effect of MPVMT layers to facilitate de-solvation, the activation energy ( $E_a$ ) which represents the de-solvation barrier for  $Zn^{2+}$  transport is identified by temperature-dependent EIS (Supplementary Fig. 23) and calculated according to the Arrhenius equation.”

## Reviewers' comments:

Reviewer #2 (Remarks to the Author):

I am impressed by the exceptional efforts dedicated by the authors' in addressing reviewers' concerns. The manuscript has been remarkably improved and it is now ready for publishing in NC.

Reviewer #4 (Remarks to the Author):

The authors indeed made improvement in the revised manuscript; however, I would agree with both Reviewer #2 and #3 on challenging the novelty and significance of this work. The authors tried to explain and emphasize the uniquenesses of this work, but these explanations seem not convincing enough. Moreover, the full-cell performance is also not good enough to be promising.

In addition, there are also issues to be addressed, please check following comments:

- 1, Authors use thick separators (Whatman grade D,  $\sim 0.67$  mm thickness) for batteries, and this may hide true behavior of zinc metal during cycling. How about cell performance using a thin separator such as Whatman grade A, are cells still able to cycle at ultrahigh rates up to  $50 \text{ mA cm}^{-2}$ ?
- 2, for practical application, zinc metal batteries need to satisfy following requirements: limited Zn use (high Zn utilization), high-loading cathodes, low N/P ratios, and lean electrolyte. However, this work does not demonstrate relevant performance under this condition. While the symmetric cell performance seems to be one of best, full cell results do not match with this claim. To further demonstrate that, a practical full cell needs to be built to show its application potential.
- 3, There is much difference between coin cells and pouch cells. While there is continuous capacity decay in coin cells (Fig. 6c), pouch cells seem to be more stable (Fig. 6h). And their Coulombic efficiency are quite different in initial cycles, some goes over 100%? Please give explanation.
- 4, Electrochemical performance of this work should be compared with previously reported lattice mismatch substrates, e.g., materials in Supplementary Table 2. The battery performance is not solely determined by little lattice mismatch. Is there solid evidence that the lowest mismatch could lead to highest performance of metal electrodes?
- 5, How to test corrosion property and HER? How to make the pouch cell? What is the total capacity? Experimental details need to be provided.
- 6, The transference number may be unreliable as proton also takes part in the ion storage process.
- 7, Please define capacity retention improvement rate in the Supplementary Table 6.
- 8, EDX Mapping of different elements in Fig. S8 shows nonuniform distribution of elements, how to explain.

## **Point-by-point response to the reviewers' comments**

Dear Reviewers,

Thank you very much for reviewing our manuscript. We greatly appreciate your professional comments and suggestions, which have helped us improve the scientific quality of this work. According to your valuable comments and suggestions, we have carefully revised the manuscript and supplementary information (SI). All details of the revision and our responses to the comments can be found in the following text. We marked the revised parts of the manuscript and SI in red so that the corresponding changes can be clearly seen.

Sincerely yours,

Guangmin Zhou

~~~~~

Guangmin Zhou, Ph.D, Associate Professor;

Tsinghua-Berkeley Shenzhen Institute (TBSI) & Tsinghua Shenzhen International Graduate School, Tsinghua University;

Low-Dimensional Materials and Devices Laboratory, Shenzhen Geim Graphene Center;

Associate Editor/Scientific Managing Editor, Energy Storage Materials (EnSM);

E-mail: guangminzhou@sz.tsinghua.edu.cn

Reviewer #2

General comments: I am impressed by the exceptional efforts dedicated by the authors' in addressing reviewers' concerns. The manuscript has been remarkably improved and it is now ready for publishing in NC.

Response: We greatly appreciate your positive comments on our work and approval of the modifications made to the manuscript.

Reviewer #4 (Revision marked with red in the revised manuscript and SI)

General comments: The authors indeed made improvement in the revised manuscript; however, I would agree with both Reviewer #2 and #3 on challenging the novelty and significance of this work. The authors tried to explain and emphasize the uniquenesses of this work, but these explanations seem not convincing enough. Moreover, the full-cell performance is also not good enough to be promising.

Response: Thanks very much for your thoughtful comments. Although Reviewer #2 had some concerns about the novelty and significance of this work at the beginning, he/she has approved the revised manuscript for publishing in *Nature Communications* after our reorganized innovation and sufficient experiment supplements. At present, Reviewers #1 and #2 both have agreed to accept this work.

1. Novelty and significance of this work

We propose an “extended substrate screening strategy” (ESSS, $d_{\text{substrate}}: d_{\text{Zn}(002)}=n:1$, $n=1, 2$) and discover a unique “planting Zn(002) seeds” mechanism for “ $2d_{\text{Zn}(002)}$ ” substrates, which significantly differs from substrates based on traditional substrate screening criterion (TSSC, $d_{\text{substrate}}: d_{\text{Zn}(002)}=1:1$). The importance of ESSS lies in that it jumps out of TSSC, solves the problems faced by TSSC, and opens up a new pathway to find a suitable substrate material for the Zn anode and even other metal anodes (**Response Fig. 1**).

According to your suggestions from **Comment 4**, we have summarized the electrochemical performance of previously reported lattice mismatch substrates (please see the response to **Comment 4** for details). It is concluded that the performance of substrates with lattice mismatch $\delta < 5\%$ ($\delta = \left| \frac{d_{\text{substrate}} - d_{\text{Zn}(002)}}{d_{\text{Zn}(002)}} \right|$), which can form a coherent interface with Zn(002) (*Nature*, 2017, 544, 460; *Nat. Rev. Mater.*, 2016, 1, 1; *Compos. Pt. B-Eng.*, 2018, 140, 27), is significantly better than substrates with $\delta > 5\%$. Notably, all reported substrates with $\delta < 5\%$ are conductive substrates of Zn-based alloy. Although these Zn-based alloys enable inhibition of dendrite growth and mitigation of corrosion, they cannot deal with side reactions owing to the direct contact between Zn electrodes and aqueous electrolytes. Obviously, using insulating substrates is expected to inhibit side reactions. However, it seems impossible to find an insulating substrate with low δ ($< 5\%$) and inhibition of side reactions by the TSSC. Therefore, we audaciously conjecture the ESSS, namely $d_{\text{substrate}}: d_{\text{Zn}(002)}=1:1 \rightarrow d_{\text{substrate}}: d_{\text{Zn}(002)}=n:1$ ($n=1, 2$). Then, through calculation, we discovered a series of phyllosilicates matching $d_{\text{substrate}}: d_{\text{Zn}(002)}=2:1$ and further selected insulting vermiculite substrates with the lowest lattice mismatch as the model to study characteristics of “ $2d_{\text{Zn}(002)}$ ” substrates and their

effects on Zn anodes. Experimental and computational results show that vermiculites induce the growth of Zn(002) and suppress side reactions. Considering that the insulating coating generally inhibits ion transport, we designed a monolayer porous vermiculite (MPVMT) with acceleration channels for Zn²⁺ by a simple and large-scale method, thus acquiring rapid ion transport and small polarization. Hence, our work verifies the feasibility of ESSS. It is worth mentioning that the ESSS we proposed is not limited to Zn metal batteries, but can also guide other metal-based batteries.

Response Fig. 1 Schematic demonstrating the origin of the proposed ESSS and its verification based on MPVMT coatings. a Schematic of challenges existing in Zn anodes. **b** Different lattice strain and orientation correlation of Zn electrodeposition

formed on substrates with incoherent ($\delta > 25\%$), semi-coherent ($25\% \geq \delta \geq 5\%$) and coherent ($\delta < 5\%$) interfaces and TSSC used in previous work. **c** Schematic of our work design of MPVMT layers based on ESSS to stabilize Zn anodes. **d** Comparison of lattice mismatch among eight typical phyllosilicates screened based on $d_{\text{substrate}}: d_{\text{Zn}(002)}=2:1$. **e** Comparison of lattice mismatch between recently reported substrates and our work.

2. Electrochemical performance of full cells

In the previous manuscript, we assembled Zn||MnO₂ coin cells and pouch cells with a mass loading of 1 mg cm⁻², and demonstrated that MPVMT coating stabilizes Zn anode and significantly slows down capacity decay.

However, as you suggested in **Comment 2**, for practical application, requirements including limited Zn use, high-loading cathodes, low N/P ratios and lean electrolyte, should be considered. Thus, we have tested the electrochemical performance of batteries under harsh conditions including high Zn utilization (>50%), high-loading cathodes (20 mg cm⁻²), low N/P ratios (1.9) and lean electrolyte (10 $\mu\text{L mg}^{-1}$). Additionally, we have assembled ampere-hour-level Zn metal battery pouch cells (1.25 Ah) based on our MPVMT@Zn anodes, which outperforms most of the previously reported ZIBs, further demonstrating the application potential of MPVMT coatings (please see responses to **Comment 2** and **Comment 5** for details).

In addition, there are also issues to be addressed, please check following comments:

Comment 1. Authors use thick separators (Whatman grade D, ~0.67 mm thickness) for batteries, and this may hide true behavior of zinc metal during cycling. How about cell performance using a thin separator such as Whatman grade A, are cells still able to cycle at ultrahigh rates up to 50 mA cm⁻²?

Response: Thanks for your valuable suggestions. We have tested the cycle performance of MPVMT@Zn anodes at 50 mA cm⁻² using Whatman grade A (GF/A) as the separator. As shown in **Response Fig. 2**, the Zn||Zn symmetric cell fails only after cycling for 7.5 h. In contrast, MPVMT@Zn symmetric cells show prolonged cycle life for over 100 h, 13 times longer than the bare Zn, demonstrating that MPVMT coatings effectively stabilize the zinc anode at ultrahigh rates.

Response Fig. 2 Cycling performance of bare Zn and MPVMT@Zn symmetric cells at $50 \text{ mA cm}^{-2}/1 \text{ mAh cm}^{-2}$ using GF/A as the separator.

Comment 2. For practical application, zinc metal batteries need to satisfy following requirements: limited Zn use (high Zn utilization), high-loading cathodes, low N/P ratios, and lean electrolyte. However, this work does not demonstrate relevant performance under this condition. While the symmetric cell performance seems to be one of best, full cell results do not match with this claim. To further demonstrate that, a practical full cell needs to be built to show its application potential.

Response: We sincerely appreciate your recognition of the outstanding symmetric cell performance in this work and thank for your valuable comments on practical full cell demonstration.

First, to fulfill the practical requirements and verify the superiority of MPVMT pairing with different cathodes, we have assembled zinc-iodine ($\text{Zn}||\text{I}_2$) practical full cells using iodine-containing electrolytes and activated carbon (AC) as the host material with the high mass loading of about 20 mg cm^{-2} . As shown in **Response Fig. 3**, under harsh conditions of high Zn utilization (51%, the thickness of Zn: $10 \mu\text{m}$) and low N/P ratio (1.9), the MPVMT coatings enable steady cycling of the Zn anodes and maintain a capacity of almost 100% after 200 cycles, which is over four times higher than the bare $\text{Zn}||\text{I}_2$ cells (capacity retention: 24%).

Response Fig. 3 Cycling performance of Zn||I₂ full cells based on bare Zn and MPVMT@Zn anodes with 2 M ZnSO₄ and 0.5 M KI electrolyte at 5 mA cm⁻².

Additionally, MPVMT@Zn||MnO₂ full cells were tested under lean electrolytes (10 μL mg⁻¹). As shown in **Response Fig. 4**, the battery operates stably at 2 C for 100 cycles and subsequently at 0.5 C for 50 cycles.

Response Fig. 4 Cycling performance of MPVMT@Zn||MnO₂ full cells under lean electrolytes (10 μL mg⁻¹).

In summary, we have tested practical Zn-metal full cells to satisfy commercial requirements, including limited Zn use (51%), high-loading cathodes (20 mg cm⁻²), low N/P ratio (1.9) and lean electrolyte (10 μL mg⁻¹), further demonstrating the application potential of our strategy on anodes.

We have revised our manuscript accordingly.

Revised manuscript, Page 25, the Fig. 6h has been modified:

Fig. 6 Electrochemical performance of full cells and pouch cells. **a** CV profiles of Zn||MnO₂ cells based on bare Zn and MPVMT@Zn anodes with the scan speed of 0.1 mV s⁻¹. **b** Rate performance of the liquid full batteries at current densities from 1 C to 10 C. **c-e** Cycling performance of liquid full cells with 2 M ZnSO₄ and 0.1 M MnSO₄ electrolyte at 2 C (**c**) and corresponding voltage profiles of 100 cycles for MPVMT@Zn||MnO₂ cells (**d**) and bare Zn||MnO₂ cells (**e**). **f, g** CLSM 3D height images of MPVMT@Zn (**f**) and bare Zn (**g**) electrodes after 500 cycles. **h** Cycling performance of Zn||I₂ full cells based on bare Zn and MPVMT@Zn anodes with 2 M ZnSO₄ and 0.5 M KI electrolyte at 5 mA cm⁻². **i, j** Optical image of an Ah-level MPVMT@Zn||MnO₂ pouch cell (**i**) and corresponding voltage profile of 4th cycle at 0.1 C (**j**). **k** Comparison of cell capacity of MPVMT@Zn||MnO₂ pouch cell with previous works.

Revised manuscript, Pages 23-24, several sentences have been added:

“These results again validate the virtue of MPVMT design (**Supplementary Table 6**). We also assembled zinc-iodine (Zn||I₂) practical full cells using iodine-containing electrolytes and activated carbon (AC) as the host material with the high mass loading of about 20 mg cm⁻². As shown in **Fig. 6h**, under harsh conditions of high Zn utilization (51%, the thickness of Zn: 10 μm) and low N/P ratio (1.9), the MPVMT coatings enable steady cycling of the Zn anodes and maintain a capacity of almost 100% after 200 cycles, which is over four times higher than the bare Zn||I₂ cells (capacity retention: 24%).”

Revised SI, Page 3, several sentences have been added:

“**Preparation of I₂ cathodes.** Activated carbon (AC, YP80F) was coated on the carbon cloth (mass loading: 20 mg cm⁻²) as the host materials by mixing well with AB and PVDF in NMP solvent with weight ratios of AC:AB:PVDF of 8:1:1. After drying at 60 °C for 12 h, the cathode was cut into 12 mm circular discs with a mass loading of 20 mg cm⁻².”

Revised SI, Page 4, a sentence has been added:

“For Zn||MnO₂ full cells, 2 M ZnSO₄ with 0.1 M MnSO₄ was used as the electrolyte and the voltage range was set to 1.0~1.8 V. For Zn||I₂ full cells, 2 M ZnSO₄ with 0.5 M KI were used as the anode and electrolyte, and the voltage range was set to 0.6~1.6 V.”

Comment 3. There is much difference between coin cells and pouch cells. While there is continuous capacity decay in coin cells (Fig. 6c), pouch cells seem to be more stable (Fig. 6h). And their Coulombic efficiency are quite different in initial cycles, some goes over 100%? Please give explanation.

Response: We are grateful for your thoughtful comments.

1. Reasons for different capacity decay in coin cells and pouch cells

Because the testing current of coin cells (1 C) and pouch cells (2 C) is different. It is widely accepted that MnO₂ electrodes suffer from serious phase transition and Mn³⁺ disproportionation reaction during the charge/discharge process, resulting in inferior rate capability and continuous capacity decay (*Nat. Energy*, 2016, 1, 16039; *Angew. Chem. Int. Ed.*, 2022, 134, e202212231). Therefore, pouch cells at 1 C are more stable than coin cells at 2 C. Under the same conditions for cathodes, the MPVMT coating stabilizes the Zn anode, significantly slowing down capacity decay. The capacity retention improvement rate (defined as follows) is over 200%

(Response Fig. 5).

$$\text{Capacity retention improvement rate} = \frac{\text{Cycled Capacity}_{\text{modified Zn}} / \text{Initial Capacity}_{\text{modified Zn}}}{\text{Cycled Capacity}_{\text{bare Zn}} / \text{Initial Capacity}_{\text{bare Zn}}}$$

Response Fig. 5 Cycling performance of liquid full cells at 2 C and pouch cells at 1 C with 2 M ZnSO₄ and 0.1 M MnSO₄ electrolyte.

2. Difference between the initial Coulomb efficiency of coin cells and pouch cells

Initial Coulombic efficiency (CE) of coin cells and pouch cells are about 93% and 97%, respectively. The different CE between them may be ascribed to a combination of cell architectures/parameters. Sun et al. (*Joule*, 2023, 7, 1) reported an experimental phenomenon that the smaller the size of Zn anode, the lower CE of the tested Zn||Cu batteries and claimed that electrodeposition of Zn²⁺ is more likely to occur at the edge than inside due to the uneven electric field, resulting in dead Zn formation and reduced CE. This may be one way to explain the difference in CE between coin cells and pouch cells, since the size of Zn anodes in pouch cells is larger than that of coin cells. Moreover, other engineering factors such as external pressure applied on the Zn metal may also play an important role. In short, the reasons for the difference in the performance of coin cells and pouch cells still need further investigation.

3. Reasons why Coulomb efficiency sometimes exceeds 100%

It is not uncommon for Zn||MnO₂ batteries to deliver CE of more than 100% due to abnormal capacity fluctuations (*Nat. Commun.*, 2023, 14, 641; *Adv. Mater.*, 2022, 34, e2200860; *Adv. Energy Mater.*, 2022, 12, 2202419). Recently, a perspective by Chao et al. (*Adv. Mater.*, 2023, 35, 2300053) summarized abnormal capacity phenomena and analyzed the possible reasons. They claimed that due to the evolution of the pH environment during the charge/discharge, abnormal capacity fluctuation is triggered by the competitive capacity contribution of the electrolyte and cathode material. Thus, abnormal capacity fluctuations affect the CE, even

making it appear higher than 100%.

Comment 4. Electrochemical performance of this work should be compared with previously reported lattice mismatch substrates, e.g., materials in Supplementary Table 2. The battery performance is not solely determined by little lattice mismatch. Is there solid evidence that the lowest mismatch could lead to highest performance of metal electrodes?

Response: We sincerely appreciate your valuable suggestions. We strongly agree that the battery performance is not solely determined by little lattice mismatch (δ). To obtain excellent Zn-metal battery performance, it is necessary to solve the problems of dendrites growth, side reactions (HER and by-products formation), corrosion and sluggish ion transport (**Response Fig. 6a**). Meanwhile, the deposition behavior of Zn^{2+} greatly affects the electrochemical performance. As first reported by Archer et al. (*Science*, 2019, 366, 645), Zn(002) lattice matching strategy has been proven to be effective, since Zn(002) deposition induced by substrates can hinder dendrite formation and alleviate corrosion due to its planar morphology and lower chemical activity compared to other lattice planes (*Adv. Mater.*, 2021, 33, e2100187; *Energy Environ. Sci.*, 2022, 15, 5017).

According to your advice, we have added the electrochemical performance comparison of this work with previously reported lattice mismatch substrates (shown in **Response Table 1**). From the table, we can basically conclude that the performance of substrates with $\delta < 5\%$ ($\delta = \left| \frac{d_{\text{substrate}} - d_{\text{Zn}(002)}}{d_{\text{Zn}(002)}} \right|$), which can form a coherent interface with Zn(002) (*Compos. Pt. B-Eng.*, 2018, 140, 27), is significantly better than substrates with $\delta > 5\%$. Notably, the reported substrates with $\delta < 5\%$ are all conductive substrates of Zn-based alloy. Although these Zn-based alloys enable inhibition of dendrite growth and mitigation of corrosion, they cannot deal with side reactions owing to the direct contact between Zn electrodes and aqueous electrolytes. Obviously, using insulating substrates is expected to inhibit side reactions. However, it seems impossible to find an insulating substrate with low δ and inhibition of side reactions by the traditional substrate screening criterion (TSSC, $d_{\text{substrate}}: d_{\text{Zn}(002)}=1:1$, **Response Fig. 6b**). Therefore, we audaciously conjecture extended substrate screening strategy (ESSS, $d_{\text{substrate}}: d_{\text{Zn}(002)}=n:1$, $n=1, 2$). Then, through calculation, we discovered a series of phyllosilicates matching $d_{\text{substrate}}: d_{\text{Zn}(002)}=2:1$ and further selected insulating vermiculite substrates with the lowest lattice mismatch as the model to study (**Response Fig. 7**). Experimental and computational results show that vermiculites induce the growth of Zn(002) and

suppress side reactions. Considering that the insulating coating generally inhibits ion transport, we designed a monolayer porous vermiculite with acceleration channels for Zn^{2+} by a simple and large-scale method, thus acquiring rapid ion transport and small polarization. Hence, our work verifies the feasibility of extending TSSC, namely $d_{\text{substrate}}: d_{\text{Zn}(002)}=1:1 \rightarrow d_{\text{substrate}}: d_{\text{Zn}(002)}=n:1$ ($n=1, 2$). Furthermore, we also find a unique “planting Zn(002) seeds” mechanism for “ $2d_{\text{Zn}(002)}$ ” substrates, which is significantly different from the traditional “ $1d_{\text{Zn}(002)}$ ” substrates, providing new theoretical guidance to reveal Zn^{2+} deposition mechanism. It is worth mentioning that the ESSS we proposed is not limited to Zn metal batteries, but can also guide other metal-based batteries.

Please note that our work is not to prove the opinion that “the lowest mismatch could lead to the highest performance of metal electrodes”. On the contrary, We notice that even using Zn(002) single crystal as the substrate with zero lattice mismatch (*Adv. Mater.*, 2022, 34, e2202552), Zn anodes still are bothered by side reactions. Consequently, our proposed ESSS opens a new pathway for the coating selection of metal electrodes, aiming to control the orientation of metal ion deposition well and overcome the challenges of side reactions. Since the traditional “ $1d_{\text{Zn}(002)}$ ” substrates cannot satisfactorily resolve troubles of Zn anodes, we suggest jumping out of traditional criteria ($d_{\text{substrate}}: d_{\text{Zn}(002)}=1:1$) and exploring new possibilities on the extended strategies ($d_{\text{substrate}}: d_{\text{Zn}(002)}=n:1$, $n=1, 2$).

Response Fig. 6 **a** Schematic of challenges existing in Zn anodes. **b** Different lattice strain and orientation correlation of Zn electrodeposition formed on substrates with incoherent ($\delta > 25\%$), semi-coherent ($25\% \geq \delta \geq 5\%$) and coherent ($\delta < 5\%$) interfaces and TSSC used in previous work.

Response Fig. 7 a Schematic of our work design of MPVMT layers based on ESSS to stabilize Zn anodes. **b** Comparison of lattice mismatch among eight typical phyllosilicates screened based on $d_{\text{substrate}} : d_{\text{Zn}(002)}=2:1$. **c** Comparison of lattice mismatch between recently reported substrates and our work.

Response Table 1. Comparison of lattice mismatch between recently reported substrates and Zn(002) plane, and their corresponding electrochemical performance.

Substrates	Lattice misfit (%)	Zn asymmetric cell		Zn symmetric cell			Ref.
		Current density (mA cm ⁻²)	Coulombic efficiency (%)	Zn Zn Current density (mA cm ⁻²)	Time (h)	Cumulative plating capacity (Ah cm ⁻²)	
Graphene	7.41	4 (Zn SS)	99%	/	/	/	Science ¹
MoS ₂	21.40	/	/	/	/	/	Adv. Mater. ²
Zn ₅ Cu	3.60	1 (Zn Ti)	99.2%	2.5	1600	2	Adv. Mater. ³
ZnAl ₂	6.10	/	/	/	/	/	
Zn ₃ Mn	2.30	10 (Zn Cu)	99.62%	80	750	30	Nat. Commun. ⁴

ZnSe	13.16	2 (Zn Ti)	99.2%	1	1530	0.765	Adv. Mater. ⁵
				30	170	2.55	
Ag(111)	7.21	/	/	/	/	/	Adv. Mater. ⁶
Sn(002)	~16	1 (Zn Cu)	98.9%	1	500	0.25	Adv. Mater. ⁷
				2	330	0.33	
In(002)	15.81	/	/	5	400	1	Adv. Funct. Mater.. ⁸
				1	2000	1	
				2	1500	1.5	
Vermiculite (002)	0.38	2 (Zn Ti)	99.4%	5	680	1.7	This work
				10	800	4	
				50	300	7.5	

We have revised our manuscript accordingly.

Revised manuscript and SI, Page 1, the title has been modified:

“An extended substrate screening strategy enabling an **ultralow** lattice mismatch for highly reversible Zn anodes”

Revised SI, Page 40, Supplementary Table 2 has been modified:

Supplementary Table 2. Comparison of lattice mismatch between recently reported substrates and Zn(002) plane, and their corresponding electrochemical performance.

Substrates	Lattice misfit (%)	Zn asymmetric cell		Zn symmetric cell			Ref.
		Current density (mA cm ⁻²)	Coulombic efficiency (%)	Zn Zn Current density (mA cm ⁻²)	Time (h)	Cumulative plating capacity (Ah cm ⁻²)	
Graphene	7.41	4 (Zn SS)	99%	/	/	/	Science ¹
MoS ₂	21.40	/	/	/	/	/	Adv. Mater. ²

Zn ₅ Cu	3.60	1 (Zn Ti)	99.2%	2.5	1600	2	Adv. Mater. ³
ZnAl ₂	6.10	/	/	/	/	/	
Zn ₃ Mn	2.30	10 (Zn Cu)	99.62%	80	750	30	Nat. Commun. ⁴
ZnSe	13.16	2 (Zn Ti)	99.2%	1	1530	0.765	Adv. Mater. ⁵
				30	170	2.55	
Ag(111)	7.21	/	/	/	/	/	Adv. Mater. ⁶
Sn(002)	~16	1 (Zn Cu)	98.9%	1	500	0.25	Adv. Mater. ⁷
				2	330	0.33	
In(002)	15.81	/	/	5	400	1	Adv. Funct. Mater.. ⁸
Vermiculite (002)	0.38	2 (Zn Ti)	99.4%	1	2000	1	This work
				2	1500	1.5	

5	680	1.7
10	800	4
50	300	7.5

Comment 5. How to test corrosion property and HER? How to make the pouch cell? What is the total capacity? Experimental details need to be provided.

Response: Thanks again for your valuable advice.

1. Corrosion property was tested by Tafel plot from -1.2 V to -1.8 V with the scan speed of 5 mV s^{-1} through a CHI760E electrochemical workstation.
2. HER was tested by in situ gas chromatography (GC, ZhongJiaoJinYuan GC7920) with homemade bare Zn and MPVMT@Zn symmetric cells at $20 \text{ mA cm}^{-2}/2 \text{ mAh cm}^{-2}$ for 90 min, and linear sweep voltammetry (LSV) from -1.2 V to -1.9 V with the scan speed of 2 mV s^{-1} through a CHI760E electrochemical workstation.
3. Total capacity of pouch cells

In the previous manuscript, the capacity of MPVMT@Zn||MnO₂ pouch cells is not enough for practical application (mass loading: 1 mg cm^{-2} , area: 6 cm^2). Therefore, in the revised manuscript, we have assembled $2 \times 3 \text{ cm}^2$ MPVMT@Zn||I₂ pouch cells (mass loading: 16 mg cm^{-2}) with total capacity of 10 mAh (**Response Fig. 8**) and $13 \times 15 \text{ cm}^2$ MPVMT@Zn||MnO₂ pouch cells (mass loading: 13.5 mg cm^{-2}) with total capacity of 1.25 Ah (**Response Fig. 9**), which outperforms most of the previously reported ZIBs (**Response Fig. 9c, Response Table 2**), further demonstrating the application potential of the modified Zn by pairing it with different cathodes.

4. Preparation of pouch cells

Preparation of MPVMT@Zn||I₂ pouch cells. Typical MPVMT@Zn||I₂ pouch cells were constructed with $2 \times 3 \text{ cm}^2$ Zn foils ($80 \mu\text{m}$), $2 \times 3 \text{ cm}^2$ I₂ cathodes (mass loading of AC: 16 mg cm^{-2}), $3 \times 4 \text{ cm}^2$ glass fiber separators and Ti foil current collectors on both sides. The shell of the pouch cell was an aluminum-plastic film and the electrolyte was 2 M ZnSO₄ and 0.5 M potassium iodide (KI, 99.5%).

Preparation of MPVMT@Zn||MnO₂ pouch cells. An ampere-hour MPVMT@Zn||MnO₂ pouch cells consisted of two $13 \times 15 \text{ cm}^2$ MnO₂ cathodes (mass loading: 13.5 mg cm^{-2}) clamped in the middle sharing one Ti foil current collector and two $13 \times 15 \text{ cm}^2$ Zn foils ($50 \mu\text{m}$) on both sides separated by $14 \times 16 \text{ cm}^2$ glass fiber separators (GF/D, Whatman). The shell of the pouch cell was an aluminum-plastic film and the electrolyte was 2 M zinc sulfate (ZnSO₄, A.R.) and 0.1 M MnSO₄.

Response Fig. 8 Cycling performance of MPVMT@Zn||I₂ pouch cells at 1 mA cm⁻².

Response Fig. 9 a, b Optical image of an Ah-level MPVMT@Zn||MnO₂ pouch cell (a) and corresponding voltage profile of 4th cycle at 0.1 C (b). **c** Comparison of cell capacity of MPVMT@Zn||MnO₂ pouch cell with previous works.

Response Table 2. Comparison of the MPVMT@Zn||MnO₂ pouch cell with previously reported Zn metal pouch cells.

Electrodes	Electrolyte	Total capacity (mAh)	Mass loading (mg cm ⁻²)	Ref.	Year
MPVMT@Zn MnO₂	2 M ZnSO₄ +0.1 M MnSO₄	1250	13.5	This work	
PVDF-Sn@Zn MnO ₂ @C	2 M ZnSO ₄ +0.1 M MnSO ₄	20	0.8	Nat. Commun. ⁹	2023
GFA@Zn I ₂	1 M ZnSO ₄	160	15	Energy Environ. Sci. ¹⁰	2022

Zn ZVO	2 M ZnSO ₄ in CarraChi gel	900	15	Nat. Commun. ¹¹	202
Zn Zn _{0.25} V ₂ O ₅ •nH ₂ O	RME	1000	20	Nat. Commun. ¹²	2023
Zn KVOH	TMP-40	9	2.1	Nat. Commun. ¹³	2023
Zn PANI	1 M Zn(PS) ₂ + 0.2 TBATS	7	2	Nat. Commun. ¹⁴	2023
Zn MnO ₂	3 M Zn(OTf) ₂ +0.1 M Mn(OTf) ₂	1550	12	Nat. Commun. ¹⁵	2017
NGO@Zn LiMn ₂ O ₄	2 M Li ₂ SO ₄ +1 M ZnSO ₄	32	16.7	Adv. Mater. ¹⁶	2021
Zn VOPO ₄	4 M Zn(OTf) ₂ +0.5 M Me ₃ EtNOTf	50	3	Nat. Nanotechnol. ¹⁷	2021

We have revised our manuscript accordingly.

Revised manuscript, Page 25, the Figs. 6i-6k have been modified:

Fig. 6 Electrochemical performance of full cells and pouch cells. **a** CV profiles of Zn||MnO₂ cells based on bare Zn and MPVMT@Zn anodes with the scan speed of 0.1 mV s⁻¹. **b** Rate performance of the liquid full batteries at current densities from 1 C to 10 C. **c-e** Cycling performance of liquid full cells with 2 M ZnSO₄ and 0.1 M MnSO₄ electrolyte at 2 C (**c**) and corresponding voltage profiles of 100 cycles for MPVMT@Zn||MnO₂ cells (**d**) and bare Zn||MnO₂ cells (**e**). **f, g** CLSM 3D height images of MPVMT@Zn (**f**) and bare Zn (**g**) electrodes after 500 cycles. **h** Cycling performance of Zn||I₂ full cells based on bare Zn and MPVMT@Zn anodes with 2 M ZnSO₄ and 0.5 M KI electrolyte at 5 mA cm⁻². **i, j** Optical image of an Ah-level MPVMT@Zn||MnO₂ pouch cell (**i**) and corresponding voltage profile of 4th cycle at 0.1 C (**j**). **k** Comparison of cell capacity of MPVMT@Zn||MnO₂ pouch cell with previous works.

Revised manuscript, Page 4, a sentence has been modified:

“As a result, the MPVMT@Zn symmetric cells exhibit stable cycling for over 800 h at a high current density of 10 mA cm^{-2} and 300 h at an ultrahigh current density of 50 mA cm^{-2} . Furthermore, practical MPVMT@Zn based 1.25 Ah pouch cell is also demonstrated.”

Revised manuscript, Page 24, several sentences have been added:

“To further verify the device's feasibility, MPVMT@Zn anodes were assembled into pouch cells. **Supplementary Fig. 29** shows that $2 \times 3 \text{ cm}^2$ MPVMT@Zn||I₂ pouch cells show stable cycling without capacity decay and high CE (99.93%) after 1000 cycles. Additionally, we demonstrate a $13 \times 15 \text{ cm}^2$ MPVMT@Zn||MnO₂ pouch cell that delivers cell capacity of 1.25 Ah, which outperforms most of the previously reported ZIBs (**Figs. 6i-k, Supplementary Table 7**).”

Revised SI, Page 3, several sentences have been added:

“**Preparation of MPVMT@Zn||MnO₂ pouch cells.** An ampere-hour MPVMT@Zn||MnO₂ pouch cells consisted of two $13 \times 15 \text{ cm}^2$ MnO₂ cathodes (mass loading: 13.5 mg cm^{-2}) clamped in the middle sharing one Ti foil current collector and two $13 \times 15 \text{ cm}^2$ Zn foils ($50 \text{ }\mu\text{m}$) on both sides separated by $14 \times 16 \text{ cm}^2$ glass fiber separators (GF/D, Whatman). The shell of the pouch cell was an aluminum-plastic film and the electrolyte was 2 M zinc sulfate (ZnSO₄, A.R.) and 0.1 M MnSO₄.”

“**Preparation of MPVMT@Zn||I₂ pouch cells.** Typical MPVMT@Zn||I₂ pouch cells were constructed with $2 \times 3 \text{ cm}^2$ Zn foils ($80 \text{ }\mu\text{m}$), $2 \times 3 \text{ cm}^2$ I₂ cathodes (mass loading of AC: 16 mg cm^{-2}), $3 \times 4 \text{ cm}^2$ glass fiber separators and Ti foil current collectors on both sides. The shell of the pouch cell was an aluminum-plastic film and the electrolyte was 2 M ZnSO₄ and 0.5 M potassium iodide (KI, 99.5%).”

Revised SI, Page 4, several sentences have been added:

“The electronic resistivity measurement, linear sweep voltammetry (LSV) and Tafel plot were performed by a CHI760E electrochemical workstation. Corrosion property was tested by Tafel plot from -1.2 V to -1.8 V with the scan speed of 5 mV s^{-1} . HER was tested by in situ gas chromatography (GC) with homemade bare Zn and MPVMT@Zn symmetric cells at $20 \text{ mA cm}^{-2}/2 \text{ mAh cm}^{-2}$ for 90 min, and linear sweep voltammetry (LSV) from -1.2 V to -1.9 V with the scan speed of 2 mV s^{-1} .”

Revised SI, Page 5, a sentence has been modified:

“XPS measurements were conducted by a PHI 5000 Versa Probe II In-Situ XPS. In situ

HER observation was performed by GC (ZhongJiaoJinYuan GC7920).”

Revised SI, Page 38, Supplementary Fig. 29 has been modified:

Supplementary Fig. 29 Cycling performance of MPVMT@Zn||I₂ pouch cells at 1 mA cm⁻².

Revised SI, Page 49, Supplementary Table 7 has been modified:

Supplementary Table 7. Comparison of the MPVMT@Zn||MnO₂ pouch cell with previously reported Zn metal pouch cells.

Electrodes	Electrolyte	Total capacity (mAh)	Mass loading (mg cm ⁻²)	Ref.	Year
MPVMT@Zn MnO₂	2 M ZnSO₄ +0.1 M MnSO₄	1250	13.5	This work	
PVDF-Sn@Zn MnO ₂ @C	2 M ZnSO ₄ +0.1 M MnSO ₄	20	0.8	Nat. Commun. ⁹	2023
GFA@Zn I ₂	1 M ZnSO ₄	160	15	Energy Environ. Sci. ¹⁰	2022
Zn ZVO	2 M ZnSO ₄ in CarraChi gel	900	15	Nat. Commun. ¹¹	202

Zn Zn _{0.25} V ₂ O ₅ •nH ₂ O	RME	1000	20	Nat. Commun. ¹²	2023
Zn KVOH	TMP-40	9	2.1	Nat. Commun. ¹³	2023
Zn PANI	1 M Zn(PS) ₂ + 0.2 TBATS	7	2	Nat. Commun. ¹⁴	2023
Zn MnO ₂	3 M Zn(OTf) ₂ +0.1 M Mn(OTf) ₂	1550	12	Nat. Commun. ¹⁵	2017
NGO@Zn Li Mn ₂ O ₄	2 M Li ₂ SO ₄ +1 M ZnSO ₄	32	16.7	Adv. Mater. ¹⁶	2021
Zn VOPO ₄	4 M Zn(OTf) ₂ +0.5 M Me ₃ EtNOTf	50	3	Nat. Nanotechnol. ¹⁷	2021

Comment 6. The transference number may be unreliable as proton also takes part in the ion storage process.

Response: Thank you for your comments.

The transference number (TN) was tested by symmetric cells with 2 M ZnSO₄ electrolytes. The pH value of 2 M ZnSO₄ electrolytes is normally > 4 (*Metallurgical and Materials Transactions B*, 1998, 29, 1157-1166; *Angew. Chem. Int. Ed.*, 2023, 62, e202311988), indicating that the concentration of proton in the electrolyte is less than 10⁻⁴ (0.0001) mol L⁻¹, while the concentration of Zn²⁺ is 2 mol L⁻¹, much higher than that of proton. Therefore, TN is mainly contributed by Zn²⁺. Besides, in previous reports, TN is generally regarded as a reference parameter to evaluate the behavior of Zn²⁺ (*Nat. Sustain.*, 2023, 6, 325-335; *Nat. Commun.*, 2021, 12, 6606; *Nat. Commun.*, 2022, 13, 5348; *Nat. Commun.*, 2023, 14, 4981; *Energy Environ. Sci.*, 2023, 16, 275-284).

In addition to the significant increase of TN (0.67) compared with bare Zn (0.20), we further verified that MPVMT coatings effectively accelerate the transport of Zn²⁺ by experiments including faster ion conductivity (**Response Fig. 10**), lower activation energy (**Response Fig. 11**) and smaller polarization voltage (**Response Fig. 12**), and COMSOL simulations (**Response Fig. 13**).

Response Fig. 10 a EIS results of the symmetric cells in SS/MPVMT@Zn/GF or SS/bare Zn/GF configurations (immersed in 2 M ZnSO₄). **b** Calculated ionic conductivity based on the EIS results.

Response Fig. 11 Calculating E_a based on temperature-dependent EIS. Nyquist plots at different temperatures of bare Zn (a) and MPVMT@Zn (c) symmetrical cells. Corresponding Arrhenius curves and calculated E_a of Zn (b) and MPVMT@Zn (d).

Response Fig. 12 Rate performance of the bare Zn and MPVMT@Zn electrodes at different current densities from 0.5 to 10 mA cm^{-2} with the same capacity of 1 mAh cm^{-2} (a) and corresponding voltage hysteresis (b)

Response Fig. 13 Electric field distribution (a) and Zn ion flux (b) of bare Zn and MPVMT@Zn at 10 mA cm^{-2} .

Comment 7. Please define capacity retention improvement rate in the Supplementary Table 6.

Response: Thank you for your suggestions. The capacity retention improvement rate is defined as follows:

$$\text{Capacity retention improvement rate} = \frac{\text{Cycled Capacity}_{\text{modified Zn}} / \text{Initial Capacity}_{\text{modified Zn}}}{\text{Cycled Capacity}_{\text{bare Zn}} / \text{Initial Capacity}_{\text{bare Zn}}}$$

We have revised our manuscript accordingly.

Revised SI, Page 48, a sentence has been added:

“Capacity retention improvement rate is defined as follows:

$$\text{Capacity retention improvement rate} = \frac{\text{Cycled Capacity}_{\text{modified Zn}} / \text{Initial Capacity}_{\text{modified Zn}}}{\text{Cycled Capacity}_{\text{bare Zn}} / \text{Initial Capacity}_{\text{bare Zn}}} ”$$

Comment 8. EDX Mapping of different elements in Fig. S8 shows nonuniform distribution of elements, how to explain.

Response: We acknowledge the reviewer’s comments and are sorry for the unclear figure and labeling mistake. Now we updated Fig. S8 as shown in **Response Fig. 14**.

Response Fig. 14 SEM image and corresponding elemental maps of MPVMT@Zn.

We have revised our manuscript accordingly.

Revised SI, Page 15, Supplementary Fig. 8 have been revised:

Supplementary Fig. 8 SEM image and corresponding elemental maps of MPVMT@Zn.

Revised manuscript, Page 7, a sentence has been modified:

“The typical thickness of the coatings is around **1.1 μm** measured by scanning electron microscopy (SEM, **Supplementary Fig. 8**).”

Revised SI, Page 16, a sentence has been modified:

“For the MPVMT coating, the thickness (L) is about **1.1 μm**. The contact area (S) is 1.13 cm². The average voltage increase is 26.6 mV. Thus, the calculated value of electronic resistivity for MPVMT layers is **$1.4 \times 10^5 \Omega \text{ cm}$** .”

Revised manuscript, Page 7, a sentence has been modified:

“Additionally, insulating MPVMT layers exhibit an electronic resistivity of approximately **$1.4 \times 10^5 \Omega \text{ cm}$** (**Fig. 2j, Supplementary Note 2**).”

References

- 1 Zheng, J. *et al.* Reversible epitaxial electrodeposition of metals in battery anodes. *Science* **366**, 645-648 (2019).
- 2 Wang, Y. *et al.* MoS₂-Mediated Epitaxial Plating of Zn Metal Anodes. *Adv. Mater.* **35**, e2208171 (2022).
- 3 Ji, J. *et al.* Zinc-Contained Alloy as a Robustly Adhered Interfacial Lattice Locking Layer for Planar and Stable Zinc Electrodeposition. *Adv. Mater.* **35**, e2211961 (2023).
- 4 Tian, H. *et al.* Stable, high-performance, dendrite-free, seawater-based aqueous batteries. *Nat. Commun.* **12**, 237 (2021).
- 5 Yang, X. *et al.* Interfacial Manipulation via In Situ Grown ZnSe Cultivator toward Highly Reversible Zn Metal Anodes. *Adv. Mater.* **33**, e2105951 (2021).
- 6 Yi, Z. *et al.* An Ultrahigh Rate and Stable Zinc Anode by Facet-Matching-Induced Dendrite Regulation. *Adv. Mater.* **34**, e2203835 (2022).
- 7 Li, S. *et al.* Toward Planar and Dendrite-Free Zn Electrodepositions by Regulating Sn-Crystal Textured Surface. *Adv. Mater.* **33**, e2008424 (2021).
- 8 Ouyang, K. *et al.* A New Insight into Ultrastable Zn Metal Batteries Enabled by In Situ Built Multifunctional Metallic Interphase. *Adv. Funct. Mater.* **32**, 2109749 (2021).
- 9 Cao, Q. *et al.* Gradient design of imprinted anode for stable Zn-ion batteries. *Nat. Commun.* **14**, 641 (2023).
- 10 Liang, G. *et al.* Gradient fluorinated alloy to enable highly reversible Zn-metal anode chemistry. *Energy Environ. Sci.* **15**, 1086-1096 (2022).
- 11 Wang, F. *et al.* Production of gas-releasing electrolyte-replenishing Ah-scale zinc metal pouch cells with aqueous gel electrolyte. *Nat. Commun.* **14**, 4211 (2023).
- 12 Wang, Y. *et al.* Sulfolane-containing aqueous electrolyte solutions for producing efficient ampere-hour-level zinc metal battery pouch cells. *Nat. Commun.* **14**, 1828 (2023).
- 13 Wang, W. *et al.* Regulating interfacial reaction through electrolyte chemistry enables gradient interphase for low-temperature zinc metal batteries. *Nat. Commun.* **14**, 5443 (2023).
- 14 Chen, S. *et al.* Coordination modulation of hydrated zinc ions to enhance redox reversibility of zinc batteries. *Nat. Commun.* **14**, 3526 (2023).
- 15 Zhang, N. *et al.* Rechargeable aqueous zinc-manganese dioxide batteries with high energy and power densities. *Nat. Commun.* **8**, 405 (2017).
- 16 Zhou, J. *et al.* Ultrathin Surface Coating of Nitrogen-Doped Graphene Enables Stable Zinc Anodes for Aqueous Zinc-Ion Batteries. *Adv. Mater.* **33**, 2101649 (2021).
- 17 Cao, L. *et al.* Fluorinated interphase enables reversible aqueous zinc battery chemistries. *Nat. Nanotechnol.* **16**, 902 (2021).

REVIEWER COMMENTS

Reviewer #4 (Remarks to the Author):

Authors have resolved most issues and the manuscript could be published after minor revision, please see following comments:

- 1, There is no description about how to obtain transference number? Is it based on the classic Bruce-Vincent method; however, this method has many limitations. Apart from that, high concentration of Zn^{2+} and low concentration of proton may not necessarily mean higher transference number of zinc ions because proton follows a hopping mechanism, which contribute significantly to ionic conductivity.
- 2, The performance between GF A and GF D separator-based cells is different (100 h vs. 300 h), as shorter cycle life detected in thin separator. Please put the response Fig. 2 into the supporting information to let readers know the difference.
- 3, Detailed information about every component in pouch cells should be provided in a table or tables, especially the mass and sizes. These values are important for calculating cell-level energy density.
- 4, How to calculate the N/P ratio of the cell in Fig. 6h, and how to determine the mass of iodine. There are two mass loadings about I2 or AC, one is 20 mg cm^{-2} and another is 16 mg cm^{-2} , it is causing some confusion.

Point-by-point response to the reviewer's comments

Dear Reviewer #4,

Thank you very much for reviewing our manuscript. We greatly appreciate your professional comments and suggestions, which have helped us improve the scientific quality of this work. According to your valuable comments and suggestions, we have carefully revised the manuscript and supplementary information (SI). All details of the revision and our responses to the comments can be found in the following text. We marked the revised parts of the manuscript and SI in red so that the corresponding changes can be clearly seen.

Sincerely yours,

Guangmin Zhou

~~~~~

Guangmin Zhou, Ph.D, Associate Professor;

Tsinghua-Berkeley Shenzhen Institute (TBSI) & Tsinghua Shenzhen International Graduate School, Tsinghua University;

Associate Editor/Scientific Managing Editor, Energy Storage Materials (EnSM);

E-mail: [guangminzhou@sz.tsinghua.edu.cn](mailto:guangminzhou@sz.tsinghua.edu.cn)

**Reviewer #4 (Revision marked with red in the revised manuscript and SI)**

**General comments:** Authors have resolved most issues and the manuscript could be published after minor revision, please see following comments:

**Response:** We greatly appreciate your positive comments on our work and approval of the modifications made to the manuscript.

In addition, there are also issues to be addressed, please check following comments:

**Comment 1.** There is no description about how to obtain transference number? Is it based on the classic Bruce-Vincent method; however, this method has many limitations. Apart from that, high concentration of  $Zn^{2+}$  and low concentration of proton may not necessarily mean higher transference number of zinc ions because proton follows a hopping mechanism, which contribute significantly to ionic conductivity.

**Response:** Thanks for your thoughtful comments. We explained how to obtain transference number in **Supplementary Note 4** based on the classic Bruce-Vincent method. The detail is as follows:

“**Supplementary Note 4.**  $Zn^{2+}$  transference number of MPVMT@Zn and bare Zn were determined in symmetrical cells by recording the impedance before and after CA tests under a polarization of 10 mV for 3600 s based on the classic Bruce-Vincent method. The transference number is calculated as follows:

$$t = \frac{I_S(\Delta V - I_0 R_0)}{I_0(\Delta V - I_S R_S)}$$
$$t_{MPVMT@Zn} = \frac{9.6 \mu A \times (10 mV - 11.3 \mu A \times 635.2 \Omega)}{11.3 \mu A \times (10 mV - 9.6 \mu A \times 669.1 \Omega)} \approx 0.67$$
$$t_{bare Zn} = \frac{4.1 \mu A \times (10 mV - 5.7 \mu A \times 1643 \Omega)}{5.7 \mu A \times (10 mV - 4.1 \mu A \times 1892 \Omega)} \approx 0.20$$

where  $\Delta V$  is the applied voltage,  $I_0$  and  $I_S$  are the initial and steady current,  $R_0$  and  $R_S$  are the initial and steady resistance, respectively.”

This method indeed has some limitations (*Commun. Mater.*, 2022, 3, 31), as it is based on some ideal assumptions. However, the results obtained by the method still throw certain light on the behavior of  $Zn^{2+}$ , and abundant works have used this method to evaluate the  $Zn^{2+}$  transference number (*Nat. Sustain.*, 2023, 6, 325-335; *Nat. Commun.*, 2021, 12, 6606; *Nat. Commun.*, 2022, 13, 5348; *Nat. Commun.*, 2023, 14, 4981; *Energy*

To illustrate the reliability of the method in calculating  $Zn^{2+}$  transference number, it is important to explore the ionic conductivity contributed by protons in the system. Therefore, we have tested the ionic conductivity of  $H_2SO_4$  solutions with a pH value of 4.3 which is approximately equal to the pH of 2 M  $ZnSO_4$  electrolytes, indicating the same concentration of proton between  $H_2SO_4$  solutions (pH=4.3) and 2 M  $ZnSO_4$  electrolytes. As shown in **Response Fig. 1**, the ionic conductivity of  $H_2SO_4$  solutions is calculated to be  $0.11\text{ mS cm}^{-1}$ , much smaller than that of 2 M  $Zn_2SO_4$  solutions ( $10.50\text{ mS cm}^{-1}$ , shown in **Response Fig. 2**), demonstrating that the contribution of ionic conductivity in 2 M  $Zn_2SO_4$  electrolytes mainly comes from  $Zn^{2+}$  and  $SO_4^{2-}$ . Therefore, the steady current  $I_s$  after polarization is mainly contributed by  $Zn^{2+}$ , since most of anions have been blocked by the electrode. It is concluded that although the Bruce-Vincent method has some limitations in the accurate quantitative assessment of  $Zn^{2+}$  transference number, the results acquired by the method are still valuable in revealing  $Zn^{2+}$  transport to some extent.

**Response Fig. 1** a EIS results of the symmetric cells in SS/bare Zn/GF configurations (immersed in pH=4.3  $H_2SO_4$ ). b Calculated ionic conductivity based on the EIS results.

**Response Fig. 2 a** EIS results of the symmetric cells in SS/MPVMT@Zn/GF or SS/bare Zn/GF configurations (immersed in 2 M ZnSO4). **b** Calculated ionic conductivity based on the EIS results.

**We have revised our manuscript accordingly.**

**Revised SI, Page 30, a sentence has been added:**

“**Supplementary Note 4.** Zn2+ transference number of MPVMT@Zn and bare Zn were determined in symmetrical cells by recording the impedance before and after CA tests under a polarization of 10 mV for 3600 s **based on the classic Bruce-Vincent method.**”

**Comment 2.** The performance between GF A and GF D separator-based cells is different (100 h vs. 300 h), as shorter cycle life detected in thin separator. Please put the response Fig. 2 into the supporting information to let readers know the difference.

**Response:** Thanks for your valuable suggestions. We have added the cycling performance of bare Zn and MPVMT@Zn symmetric cells using GF/A separators into the supporting information.

**Revised SI, Page 33, Supplementary Fig. 24 has been revised:**

**Supplementary Fig. 24** Cycling performance of bare Zn and MPVMT@Zn symmetric cells using GF/D separators at 2 mA cm-2/2 mAh cm-2 (a), 10 mA cm-2/1 mAh cm-2 (b), 0.6 mA cm-2/10 mAh cm-2 (c) and 10 mA cm-2/10 mAh cm-2 (d), and GF/A separators at 50 mA cm-2/1 mAh cm-2 (e), respectively. Note that the performance of symmetric cells with thin separators (GF/A) is inferior to that of symmetric cells with thick separators (GF/D).

**Revised SI, Page 3, a sentence has been added:**

“An ampere-hour MPVMT@Zn||MnO2 pouch cells consisted of two 13×15 cm2 MnO2 cathodes (mass loading: 13.5 mg cm-2) clamped in the middle sharing one Ti foil current collector and two 13×15 cm2 Zn foils (50 μm) on both sides separated by 14×16 cm2 glass fiber (GF) separators (GF/D from Whatman was used in this work unless otherwise specified).”

**Comment 3.** Detailed information about every component in pouch cells should be provided in a table or tables, especially the mass and sizes. These values are important for calculating cell-level energy density.

**Response:** We sincerely appreciate your advice. We have added detailed information about every component in pouch cells, as shown in **Response Table 1**. Accordingly, the volumetric energy density ( $E_{\text{volumetric}}$ ) of the pouch cell based on all components except electrolytes, which accounts for the diffusion of the electrolyte within cathodes and separators resulting in little contribution to the overall volume, is calculated as follows:

$$E_{\text{volumetric}} = \frac{1.74 \times 10^6}{150 \times 130 \times (0.05 \times 2 + 0.15 \times 2 + 0.27 \times 2 + 0.02 + 0.113 \times 2)} \approx 75 \text{ Wh L}^{-1}$$

The specific energy density ( $E_{\text{specific}}$ ) of the pouch cell is calculated as follows:

$$E_{\text{specific}} (\text{active materials}) = \frac{1.74 \times 10^3}{(6.9 \times 2 + 2.6 \times 2)} \approx 92 \text{ Wh kg}^{-1},$$

$$\text{and } E_{\text{specific}} (\text{whole cell}) = \frac{1.74 \times 10^3}{(6.9 \times 2 + 2.6 \times 2 + 2.7 \times 2 + 1.7 + 48.8 + 8.6)} \approx 21 \text{ Wh kg}^{-1}$$

**Response Table 1. The parameters of every component in the ampere-hour MPVMT@Zn||MnO2 pouch cell.**

| Components                                                       | Length(mm) | Width(mm) | Thickness(mm) | Weight(g)   |
|------------------------------------------------------------------|----------------|---------------|-------------------|-----------------|
| Anode (Zn foil)                                                  | 150            | 130           | 0.05*2            | 6.9*2           |
| Cathode (MnO 2 )                                      | 150            | 130           | 0.15*2            | 2.6*2           |
| Separator (GF)                                                   | 160            | 140           | 0.27*2            | 2.7*2           |
| Current collector (Ti foil)                                      | 150            | 130           | 0.02              | 1.7             |
| Electrolyte(2 M ZnSO 4 +0.1 M MnSO 4 ) | /              | /             | /                 | 48.8(36 ml) |
| Aluminum-plastic film                                            | 330            | 150           | 0.113*2           | 8.6             |

**We have revised our manuscript accordingly.**

**Revised SI, Page 51, Supplementary Table 8 has been added:**

**Supplementary Table 8. The parameters of every component in the ampere-hour MPVMT@Zn||MnO2 pouch cell.**

| Components                                                       | Length(mm) | Width(mm) | Thickness(mm) | Weight(g)   |
|------------------------------------------------------------------|----------------|---------------|-------------------|-----------------|
| Anode (Zn foil)                                                  | 150            | 130           | 0.05*2            | 6.9*2           |
| Cathode (MnO 2 )                                      | 150            | 130           | 0.15*2            | 2.6*2           |
| Separator (GF)                                                   | 160            | 140           | 0.27*2            | 2.7*2           |
| Current collector (Ti foil)                                      | 150            | 130           | 0.02              | 1.7             |
| Electrolyte(2 M ZnSO 4 +0.1 M MnSO 4 ) | /              | /             | /                 | 48.8(36 ml) |
| Aluminum-plastic film                                            | 330            | 150           | 0.113*2           | 8.6             |

**Revised SI, Page 51, Supplementary Note 5 has been added:**

“**Supplementary Note 5.** The volumetric energy density ( $E_{\text{volumetric}}$ ) of the pouch cell based on all components (listed in **Supplementary Table 8**) except electrolytes, which accounts for the diffusion of the electrolyte within cathodes and separators resulting in little contribution to the overall volume, is calculated as follows:

$$E_{\text{volumetric}} = \frac{1.74 \times 10^6}{150 \times 130 \times (0.05 \times 2 + 0.15 \times 2 + 0.27 \times 2 + 0.02 + 0.113 \times 2)} \approx 75 \text{ Wh L}^{-1}$$

The specific energy density ( $E_{\text{specific}}$ ) of the pouch cell is calculated as follows:

$$E_{\text{specific}} (\text{active materials}) = \frac{1.74 \times 10^3}{(6.9 \times 2 + 2.6 \times 2)} \approx 92 \text{ Wh kg}^{-1},$$

$$\text{and } E_{\text{specific}} (\text{whole cell}) = \frac{1.74 \times 10^3}{(6.9 \times 2 + 2.6 \times 2 + 2.7 \times 2 + 1.7 + 48.8 + 8.6)} \approx 21 \text{ Wh kg}^{-1} ”$$

**Revised manuscript, Page 24, a sentence has been modified:**

“Additionally, we demonstrate a 13×15 cm2 MPVMT@Zn||MnO2 pouch cell that delivers cell capacity of 1.25 Ah and energy density of 75 Wh L-1, which outperforms most of the previously reported ZIBs (**Figs. 6i-k, Supplementary Tables 7 and 8, Supplementary Note 5**)”

**Comment 4.** How to calculate the N/P ratio of the cell in Fig. 6h, and how to determine the mass of iodine. There are two mass loadings about I2 or AC, one is 20 mg cm-2 and another is 16 mg cm-2, it is causing some confusion.

**Response:** We are grateful for your comments and sorry for the confusing description.

1. In **Fig. 6h**, the N/P ratio is calculated as follows:

$$\frac{N}{P} = \frac{10 \mu\text{m} \times 1 \text{ cm}^2 \times 820 \text{ mAh g}_{\text{Zn}}^{-1} \times 7.14 \text{ g cm}^{-3}}{3.01 \text{ mAh}} \approx 1.9$$

**Fig. 6h** Cycling performance of Zn||I2 full cells based on bare Zn and MPVMT@Zn anodes with 2 M ZnSO4 and 0.5 M KI electrolyte at 5 mA cm-2.

2. Both mass loadings (20 mg cm-2 and 16 mg cm-2) correspond to the mass of activated carbon (AC).

When the mass loading of AC is 20 mg cm-2 in coin cells, the corresponding mass of iodine is calculated as follows:

$$\text{Loading mass}_{\text{iodine}} = \frac{3.01 \text{ mAh}}{211 \text{ mAh } g_{I_2}^{-1} \times 1 \text{ cm}^2} = 14 \text{ mg cm}^2$$

When the mass loading of AC is 16 mg cm-2 in pouch cells, the corresponding mass of iodine is calculated as follows:

$$\text{Loading mass}_{\text{iodine}} = \frac{10.4 \text{ mAh}}{211 \text{ mAh } g_{I_2}^{-1} \times 6 \text{ cm}^2} = 8 \text{ mg cm}^2$$

## **REVIEWERS' COMMENTS**

Reviewer #4 (Remarks to the Author):

None. The authors have properly answered all questions.